# EIGENSPACE RESTRUCTURING: A PRINCIPLE OF SPACE AND FREQUENCY IN NEURAL NETWORKS

## ABSTRACT

Understanding the fundamental principles behind the massive success of neural networks is one of the most important open questions in deep learning. However, due to the highly complex nature of the problem, progress has been relatively slow. In this note, through the lens of infinite-width networks, a.k.a. neural kernels, we present one such principle resulting from hierarchical locality. It is well-known that the eigenstructure of infinite-width multilayer perceptrons (MLPs) depends solely on the concept *frequency*, which measures the order of interactions. We show that the topologies from convolutional networks (CNNs) restructure the associated eigenspaces into finer subspaces. In addition to frequency, the new structure also depends on the concept *space* — the distance among interaction terms, defined via the length of a minimum spanning tree containing them. The resulting fine-grained eigenstructure dramatically improves the network's learnability, empowering them to simultaneously model a much richer class of interactions, including long-range-low-frequency interactions, short-range-high-frequency interactions, and various interpolations and extrapolations in-between. Finally, we show that increasing the depth of a CNN can improve the inter/extrapolation resolution and, therefore, the network's learnability.

## 1 INTRODUCTION

Learning in high dimensions is commonly believed to suffer from the curse of dimensionality, in which the number of samples required to solve the problem grows rapidly (often polynomially) with the dimensionality of the input. Nevertheless, modern neural networks often exhibit an astonishing power to tackle a wide range of highly complex and high-dimensional real-world problems, many of which were thought to be out-of-scope of known methods (Krizhevsky et al., 2012; Vaswani et al., 2017; Devlin et al., 2018; Silver et al., 2016; Senior et al., 2020; Kaplan et al., 2020). What are the mathematical principles that govern the astonishing power of neural networks? This question perhaps is the most crucial research question in the theory of deep learning because such principles are also the keys to resolve fundamental questions in the practice of machine learning such as (out-of-distribution) generalization (Zhang et al., 2021), calibration (Ovadia et al., 2019), interpretability (Montavon et al., 2018), robustness (Goodfellow et al., 2014).

Unarguably, there can be more than one of such principles. They are related to one or more of the three basic ingredients of machine learning methods: the data, the model and the inference algorithm. Among them, the models, a.k.a. architectures of neural networks are the most crucial innovation in deep learning that set it apart from classical machine learning methods. More importantly, the current revolution in machine learning is initialized by the (re-)introduction of convolution-based architectures (Krizhevsky et al., 2012; Lecun, 1989), and subsequent breakthroughs are often driven by the discovery or application of novel architectures (Vaswani et al. (2017); Devlin et al. (2018)). As such, identifying and understanding fundamental roles of architectures are of great importance.

In this paper, we take a step forwards by leveraging recent developments in overparameterized networks (Poole et al. (2016); Daniely et al. (2016); Schoenholz et al. (2017); Lee et al. (2018); Matthews et al. (2018); Xiao et al. (2018); Jacot et al. (2018); Du et al. (2018); Novak et al. (2019a); Lee et al. (2019) and many others.) These developments have discovered an important connection between neural networks and kernel machines: the Neural Network Gaussian Process (NNGP) kernels and the neural tangent kernels (NTKs). Under certain scaling limits, the former describes the

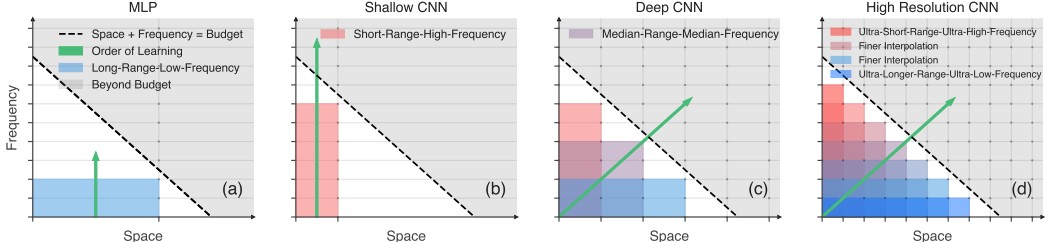

Figure 1: **Architectural Inductive Biases.** An demonstration of learnable functions vs architectures for four families of architectures. Each shaded box indicates the maximum learnable eigenspaces within a given compute budget (**Dashed Line.**) From left to right: (a) MLPs can model **Long-Range-Low-Frequency** interactions; (b) S-CNNs can model **Short-Range-High-Frequency** interactions; (c) Additionally, D-CNNs can also model interactions between **LRLF** and **SRHF**, a.k.a., **Median-Range-Median-Frequency** interactions. (d) Finally, HS-CNNs can additionally model interactions of **Ultra-Short-Range-Ultra-High-Frequency**, **Ultra-Long-Range-Ultra-Low-Frequency**, and **finer interpolations** in-between. The **Green Arrow** indicates the direction of expansion of learnable functions when increasing the compute budget.

distribution of the outputs of a randomly initialized network (a.k.a. *prior*), and the latter can describe the network's gradient descent dynamics. Although recent work (Ghorbani et al., 2019; Yang & Hu, 2020) has identified several limitations of using them in studying the feature learning dynamics of practical networks, we show that they do capture several crucial aspects of the architectural inductive biases.

Our main contribution is an *eigenspace restructuring* theorem. It characterizes a mathematical connection between a network's architecture and its learnability through a trade-off between *space* and *frequency*, providing novel insights behind the mystery power of deep CNNs (more generally, hierarchical locality (Deza et al., 2020; Vasilescu et al., 2021).) By *frequency*, we mean the degree (order) of the eigenfunction and by *space*, we mean the spatial distance among the eigenfunction's interaction terms. We summarize our main contribution below; see Fig. 1.

1. The learning order (see **Green Arrow** in Fig. 1) of eigen-functions is governed by the learning index (LI), the sum of the frequency index (FI) and the spatial index (SI), which can be characterized precisely by the network's topology.

2. There is a trade-off between *space* and *frequency*: within a fixed (compute/data) budget, it is impossible to model generic Long-Range-High-Frequency interactions. MLPs can model **Long-Range-Low-Frequency (LRHF)** interactions but fail to model **Short-Range-High-Frequency (SRHF)**, while shallow CNNs (S-CNNs) are the opposite. Remarkably, deep CNNs (D-CNNs) can simultaneously model both and various interpolating interactions between them (e.g., **Median-Range-Median-Frequency (MRMF)**.)

3. In addition, *high-resolution* CNNs (HS-CNNs, EfficientNet-type of model scaling) further broaden the class of learnable functions to contain (1) extrapolation: **Ultra-Long-Range-Ultra-Low-Frequency** and the **dual** interactions and (2) **finer interpolations** interactions.

4. Finally, we verify the above claims empirically for neural kernel methods and finite-width networks using *SGD + Momentum* for dataset and networks of practical sizes.

## 2 LINEAR AND LINEARIZED MODELS

As a warm up exercise, we briefly go through the training dynamics of linear models. Let $(\mathcal{X}, \mathcal{Y})$ denote the inputs and labels, where $\mathcal{X} \subseteq \mathbb{R}^d$ and $\mathcal{Y} \subseteq \mathbb{R}$. Assume $J : \mathbb{R}^d \to \mathbb{R}^n$ is a feature map and the task is to learn a linear function $f(x, \theta) = J(x)\theta$ to minimize the MSE objective $\frac{1}{2} \sum_{(x,y) \in (\mathcal{X}, \mathcal{Y})} |f(x, \theta) - y|^2$. Let $\mathcal{R}(\mathcal{X}, \theta) = f(\mathcal{X}, \theta) - \mathcal{Y}$ be the residual of the predictions of $\mathcal{X}$. Then the gradient flow dynamics can be written as

$$\frac{d}{dt} \mathcal{R}(\mathcal{X}, \theta) = -J(\mathcal{X}) J^T(\mathcal{X}) \mathcal{R}(\mathcal{X}, \theta) \equiv -\mathcal{K}(\mathcal{X}, \mathcal{X}) \mathcal{R}(\mathcal{X}, \theta) \tag{1}$$

Since the feature kernel $\mathcal{K}(\mathcal{X}, \mathcal{X}) = J(\mathcal{X})J^T(\mathcal{X})$ is constant in time, the above ODE can be solved in closed form. Let $m = |\mathcal{X}|$ the cardinality of $\mathcal{X}$ and $\hat{\mathcal{K}}(j)/u_j$ be the $j$-th eigenvalue/eigenvector of $\mathcal{K}(\mathcal{X}, \mathcal{X})$ in descending order. By initializing $\theta = 0$ at time $t = 0$ and denoting the projection by $\eta_j = u_j^T \mathfrak{R}(\mathcal{X}, 0)$, the dynamics of the residual and the loss can be reduced to

$$\mathfrak{R}(\mathcal{X}, \theta_t) = \sum_{j \in [m]} e^{-\hat{\mathcal{K}}(j)t} \eta_j u_j, \quad \mathcal{L}(\theta_t) = \frac{1}{2} \sum_{j \in [m]} e^{-2\hat{\mathcal{K}}(j)t} \eta_j^2 \qquad (2)$$

Therefore, to make the residual in $u_j$ smaller than some $\epsilon > 0$, the amount of time needed is $t \geq \hat{\mathcal{K}}(j)^{-1} \log \frac{2\epsilon}{\eta_j^2}/2$. The larger $\hat{\mathcal{K}}(j)$ is, the shorter amount of time it takes to learn $u_j$.

Although simple, linear models provide us with the most useful intuition behind the relation between "eigenstructures" and learning dynamics.

## 2.1 LINEARIZED NEURAL NETWORKS: NNGP KERNELS AND NT KERNELS

Let $f(\theta, x)$ be a general function, e.g. $f$ is neural network parameterized by $\theta$. Similarly,

$$\frac{d}{dt}\mathfrak{R}(\mathcal{X}, \theta) = -J(\mathcal{X}; \theta)J^T(\mathcal{X}; \theta)\mathfrak{R}(\mathcal{X}, \theta) \equiv -\mathcal{K}(\mathcal{X}, \mathcal{X}; \theta)\mathfrak{R}(\mathcal{X}, \theta). \qquad (3)$$

However, the kernel $\mathcal{K}(\mathcal{X}, \mathcal{X}; \theta)$ depends on $\theta$ via the Jacobian $J(\mathcal{X}; \theta)$ of $f(\mathcal{X}; \theta)$ and evolves with time. The above system is unsolvable in general. However, under certain parameterization methods (e.g. Sohl-Dickstein et al. (2020)) and when the network is sufficient wide, this kernel does not change much during training and converges to a deterministic kernel called the NTK (Jacot et al., 2018),

$$\mathcal{K}(\mathcal{X}, \mathcal{X}; \theta) \to \Theta(\mathcal{X}, \mathcal{X}) \quad \text{as width} \to \infty. \qquad (4)$$

The residual dynamics becomes a constant coefficient ODE again $\dot{\mathfrak{R}}(\mathcal{X}, \theta) = -\Theta(\mathcal{X}, \mathcal{X})\mathfrak{R}(\mathcal{X}, \theta)$. To solve this system, we need the initial value of $\mathfrak{R}(\mathcal{X}, \theta)$. Since the parameters $\theta$ are often initialized using iid standard Gaussian variables, as the width approach infinity, the logits $f(\mathcal{X}; \theta)$ converge to a Gaussian process (GP), known as the neural network Gaussian process (NNGP). Specifically, $f(\mathcal{X}; \theta) \sim \mathcal{N}(\mathbf{0}; \mathcal{K}(\mathcal{X}, \mathcal{X}))$, where $\mathcal{K}$ is the NNGP kernel. Note that one can also treat infinite-width networks as Bayesian models, a.k.a. Bayesian Neural Networks, and apply Bayesian inference to compute the posteriors. This approach is equivalent to training *only* the network's classification layer (Lee et al., 2019) and the gradient descent dynamics is described by the kernel $\mathcal{K}$.

As such, there are two natural kernels, the NTK $\Theta$ and the NNGP kernel $\mathcal{K}$, associated to infinite-width networks, whose training dynamics are governed by constant coefficient ODEs. To make progress, it is tempting to apply Mercer's Theorem to eigendecompose $\Theta$ and $\mathcal{K}$, e.g.,

$$\mathcal{K}(x, \bar{x}) = \sum \hat{\mathcal{K}}(j)\phi_j(x)\phi_j(\bar{x}) \quad \text{and} \quad \Theta(x, \bar{x}) = \sum \hat{\Theta}(j)\psi_j(x)\psi_j(\bar{x}) \qquad (5)$$

One advantage of applying this decomposition is that it has almost no constraint on the kernels and the inputs. However, this decomposition is too coarse to be useful since it can hardly provide fine-grained information about the eigenstructures. E.g, it is not clear what are the corrections to Eq. (5) when changing the architecture from a 2-layer CNN to a 4-layer CNNs. For this reason, we choose to work on the product space of hyperspheres, which has richer mathematical structures. Our primary goal is to characterize the analytical dependence of the decomposition Eq. (5) on the network's topology in the high-dimensional limit.

## 3 NEURAL COMPUTATIONS ON DAGS

Notations will become heavier starting from this section. In particular, we rely crucially on the directed acyclic graphs (DAGs) and minimal spanning trees (MSTs) to define the spatial complexity of eigenfunctions. In Sec. B, we provide a toy example to help understand the motivation.

For a positive integer $p$, let $\mathbb{S}_{p-1}$ denote the unit sphere in $\mathbb{R}^p$ and $\overline{\mathbb{S}}_{p-1} = \sqrt{p}\mathbb{S}_{p-1}$, the sphere of radius $\sqrt{p}$ in $\mathbb{R}^p$. We introduce the normalized sum (integral)

$$\fint_{x \in X} f(x) \equiv |X|^{-1} \sum_{x \in X} f(x) \quad \left( \fint_{x \in X} f(x) \equiv \mu(X)^{-1} \int_{x \in X} f(x)\mu(dx) \right) \qquad (6)$$

where $X$ is a finite set (a measurable set with a finite positive measure $\mu$).

We find it more convenient to express the computations in neural networks, and in neural kernels via DAGs (Daniely et al., 2016), as both computations are of *recursive* nature. The associated DAG of a network can be thought of as the same network by setting all its widths (or the number of channels for CNNs) to 1. Let $\mathcal{G} = (\mathcal{N}, \mathcal{E})$ denote a DAG, where $\mathcal{N}$ and $\mathcal{E}$ are the nodes and edges, resp. We always assume the graph to have a unique output node $o_{\mathcal{G}}$ and is an ancestor of all other nodes. Denote $\mathcal{N}_0 \subseteq \mathcal{N}$ the set of input nodes (leaves) of $\mathcal{G}$, i.e., the collection nodes without a child. Each node $u \in \mathcal{N}$ is associated with a pointwise function $\phi_u : \mathbb{R} \to \mathbb{R}$, which is normalized in the sense $\mathbb{E}_{z \in \mathcal{N}(0,1)} \phi_u^2(z) = 1$. It induces a function $\phi_u^* : I \equiv [-1,1] \to I$ defined to be $\phi_u^*(t) = \mathbb{E}_{(z_1,z_2) \in \mathcal{N}_t} \phi_u(z_1) \phi_u(z_2)$. Here $\mathcal{N}_t$ denotes a pair of standard Gaussians with correlation $t$. We associate each $u \in \mathcal{N}$ a finite-dimensional Hilbert space $\mathbb{H}_u$, and each $uv \in \mathcal{E}$ a bounded linear operator $\mathcal{L}_{uv} : \mathbb{H}_v \to \mathbb{H}_u$. Let

$$\boldsymbol{\mathcal{X}} \equiv \prod_{u \in \mathcal{N}_0} \boldsymbol{\mathcal{X}}_u \equiv \prod_{u \in \mathcal{N}_0} \overline{\mathbb{S}}_{\dim(\mathbb{H}_u)-1} \subseteq \prod_{u \in \mathcal{N}_0} \mathbb{H}_u \quad \text{and} \quad \boldsymbol{I} = I^{|\mathcal{N}_0|}$$

be the input *tensors* and the input *correlations* to the graph $\mathcal{G}$, resp. We associate two types of computations to a DAG: finite-width neural network computation and kernel computation,

$$\mathcal{N}_{\mathcal{G}} : \boldsymbol{\mathcal{X}} \to \mathbb{H}_{o_{\mathcal{G}}} \quad \text{and} \quad \mathcal{K}_{\mathcal{G}} : \boldsymbol{I} \to I \,, \tag{7}$$

resp. They are defined recursively as follows

$$\mathcal{N}_u(\boldsymbol{x}) = \phi_u \left( \sum_{v : uv \in \mathcal{E}} \mathcal{L}_{uv}(\mathcal{N}_v(\boldsymbol{x})) \right) \qquad \text{if} \quad u \notin \mathcal{N}_0 \quad \text{else} \quad \mathcal{N}_u(\boldsymbol{x}) = \boldsymbol{x}_u \tag{8}$$

$$\mathcal{K}_u(\boldsymbol{t}) = \phi_u^* \left( \fint_{v : uv \in \mathcal{E}} \mathcal{K}_v(\boldsymbol{t}) \right) \qquad \text{if} \quad u \notin \mathcal{N}_0 \quad \text{else} \quad \mathcal{K}_u(\boldsymbol{t}) = \boldsymbol{t}_u \tag{9}$$

where $\boldsymbol{x} \in \boldsymbol{\mathcal{X}}$ and $\boldsymbol{t} \in \boldsymbol{I}$. The outputs of the computations are $\mathcal{N}_{\mathcal{G}}(\boldsymbol{x}) = \mathcal{N}o_{\mathcal{G}}(\boldsymbol{x})$ and $\mathcal{K}_{\mathcal{G}}(\boldsymbol{t}) = \mathcal{K}_{o_{\mathcal{G}}}(\boldsymbol{t})$. Note that $\mathcal{K}_{\mathcal{G}}$ is indeed the NNGP kernel. The NTK can also be written recursively as

$$\Theta_u(\boldsymbol{t}) = \dot{\phi}_u^* \left( \fint_{v : uv \in \mathcal{E}} \mathcal{K}_v(\boldsymbol{t}) \right) \fint_{v : uv \in \mathcal{E}} (\mathcal{K}_v(\boldsymbol{t}) + \Theta_v(\boldsymbol{t})) \quad \text{with} \quad \Theta_{\mathcal{G}} = \Theta_{o_{\mathcal{G}}} . \tag{10}$$

Here, $\Theta_u = 0$ if $u \in \mathcal{N}_0$ and $\dot{\phi}_u^*$ is the derivative of $\phi_u^*$.

## 3.1 Three Examples: MLPs, S-CNNs and D-CNNs.

To unpack the notation, we consider three concrete examples: an $L$-hidden layer MLP, a shallow convolutional network (S-CNN) that contains only one convolutional layer and a deep convolutional network (D-CNN) that contains $(1 + L)$ convolutional layers. The architectures are

**MLP:** $\quad [Input] \to [Dense\text{-}Act]^{\otimes L} \to [Dense]$ (11)

**S-CNN:** $\quad [Input] \to [Conv(p)\text{-}Act] \to [Flatten\text{-}Dense]$ (12)

**D-CNN:** $\quad [Input] \to [Conv(p)\text{-}Act] \to [Conv(k)\text{-}Act]^{\otimes L} \to [Flatten\text{-}Dense\text{-}Act] \to [Dense]$ (13)

where $p/k$ is the filter size of the first/hidden layers and $Act$ means an activation layer. We choose the stride to be the same as the size of the filter for all convolutional layers and choose *flattening* as the readout strategy rather than pooling. See Fig. 2 (a, b, c) for the DAGs associated to a (1+3)-layer CNN (with $p = k = d^{\frac{1}{4}}$), a (1+1)-layer CNN (with $p = k = d^{\frac{1}{2}}$) and a 4-layer MLP.

**MLPs.** Let $\mathcal{G}$ be a linked list with $(L + 2)$ nodes, including the input/output nodes. Let $\mathcal{L}_{uv} \in \mathbb{R}^{n_u \times n_v}$, where $n_{u/v} = \dim(\mathbb{H}_{u/v})$ and the activations of the input/output nodes be the identity function. Then $\mathcal{N}_{\mathcal{G}}$ represents a $L$-hidden-layer MLP. In addition, let $\mathcal{L}_{uv}$ be initialized iid as

$$\mathcal{L}_{uv} = \frac{1}{\sqrt{n_v}} \left( \omega_{uv,ij} \right)_{i \in [n_u], j \in [n_v]} , \quad \omega_{uv,ij} \sim \mathcal{N}(0, 1) . \tag{14}$$

Let $\boldsymbol{t}_{\boldsymbol{x}, \boldsymbol{x}'} = \boldsymbol{x}^T \boldsymbol{x}'/n_u$ for $u \in \mathcal{N}_0$ and $n_v \to \infty$ for all hidden nodes, then the outputs of $\mathcal{N}(\boldsymbol{\mathcal{X}})$ converge weakly to the GP $\mathcal{GP}(0, \mathcal{K}_{\mathcal{G}}(\boldsymbol{t}_{\boldsymbol{x},\boldsymbol{x}'})_{\boldsymbol{x},\boldsymbol{x}' \in \boldsymbol{\mathcal{X}}})$ and $\Theta_{\mathcal{G}}(\boldsymbol{t}_{\boldsymbol{x},\boldsymbol{x}'})_{\boldsymbol{x},\boldsymbol{x}' \in \boldsymbol{\mathcal{X}}}$ is the NTK in the sense

$$\mathbb{E}\mathcal{N}_{\mathcal{G}}(\boldsymbol{x})\mathcal{N}_{\mathcal{G}}(\boldsymbol{x}') \xrightarrow{\text{in prob.}} \mathcal{K}_{\mathcal{G}}(\boldsymbol{t}_{\boldsymbol{x},\boldsymbol{x}'}) \quad \text{and} \quad \langle \nabla \mathcal{N}_{\mathcal{G}}(\boldsymbol{x}), \nabla \mathcal{N}_{\mathcal{G}}(\boldsymbol{x}') \rangle \xrightarrow{\text{in prob.}} \Theta_{\mathcal{G}}(\boldsymbol{t}_{\boldsymbol{x},\boldsymbol{x}'}) . \tag{15}$$

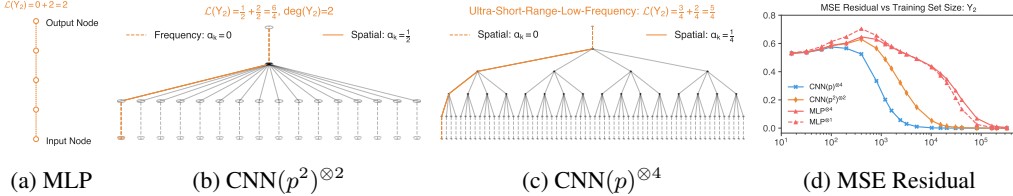

(a) MLP      (b) CNN$(p^2)^{\otimes 2}$      (c) CNN$(p)^{\otimes 4}$      (d) MSE Residual

Figure 2: **Architectures/DAGs. vs Eigenfunctions vs Learning Indices.** Left to right: DAGs associated to (a) a four-layer MLP; (b) CNN$(p^2)^{\otimes 2}$, a "D"-CNN that has two convolutional layer (c) CNN$(p)^{\otimes 4}$, a "HR"-CNN that has four convolutional layers; and (d) MSE (Y-axis) vs training set size (X-axis) for $\mathbf{Y_2}$ obtained by NTK-regression for 4 architectures. Here $\mathbf{Y_2}$ is a linear combination of eigenfunctions of Short-Range-Low-Frequency interactions ($\deg(\mathbf{Y_2}) = 2$); see Sec. D for the expression. The DAGs are generated with $p = 4$. In each DAG, the **Dashed Lines** represent the edges with zero weights. The **Solid Lines** have weights $0$, $\frac{1}{2}$ and $\frac{1}{4}$ in (a), (b) and (c), resp. The **colored path** represents the minimum spanning tree used to compute the spatial indices of $\mathbf{Y_2}$. Under architectures (a), (b) and (c), the spatial indices are $0$, $\frac{1}{2}$ and $\frac{3}{4}$, resp. Each input node represents an input patch of dimension $p^4 = d$, $p^2 = d^{\frac{1}{2}}$ and $p = d^{\frac{1}{4}}$ and the frequency indices are $2$, $2 \times \frac{1}{2}$ and $2 \times \frac{1}{4}$ in (a), (b) and (c), resp.

Indeed, note that $\deg(u) = 1$ for all $u \notin \mathcal{N}_0$. Eq. (9) and Eq. (10) become

$$\mathscr{K}_u(\boldsymbol{t}_{\boldsymbol{x},\boldsymbol{x}'}) = \phi_u^*(\mathscr{K}_v(\boldsymbol{t}_{\boldsymbol{x},\boldsymbol{x}'})) \quad \text{and} \quad \Theta_u(\boldsymbol{t}_{\boldsymbol{x},\boldsymbol{x}'}) = \dot{\phi}_u^*(\mathscr{K}_v(\boldsymbol{t}_{\boldsymbol{x},\boldsymbol{x}'}))(\mathscr{K}_v(\boldsymbol{t}_{\boldsymbol{x},\boldsymbol{x}'}) + \Theta_v(\boldsymbol{t}_{\boldsymbol{x},\boldsymbol{x}'})) \quad (16)$$

which are the recursive formulas for the NNGP kernel and NTK; see e.g. Sec.E in Lee et al. (2019).

**S-CNN.** The input $\mathcal{X} = (\overline{\mathbb{S}}_{p-1})^{1 \times w} \subseteq \mathbb{R}^d$, where $p$ is the patch size, $w$ is the number of patches, $d = pw$ is the dimension of the inputs. Here, the inputs have been *pre-processed* by a patch extractor and then by a normalization operator. In words, the S-CNN has one convolutional layer with filter size $p$, followed by an activation function $\phi$ (e.g., Relu), and finally by a *flatten-dense* readout layer. Mathematically, by letting $n \in \mathbb{N}$ be the number of channels in the hidden layer, the output (i.e., logit) is given by

**Convolution + Activation:** $\quad z_{ij}(\boldsymbol{x}) = \phi\left(p^{-\frac{1}{2}}\sum_{\beta \in [p]} \omega_{1,j,\beta} x_{\beta,i}\right) \quad \text{for} \quad i \in [w], j \in [n]$ (17)

**Flatten + Dense:** $\quad f(\boldsymbol{x}) = (wn)^{-\frac{1}{2}}\sum_{i \in [w], j \in [n]} \omega_{2,ij} z_{ij}(\boldsymbol{x}),$ (18)

where $\omega_{1,i,\beta}$ and $\omega_{2,ij}$ are the parameters of the first and readout layers, resp.

We can associate a DAG $\mathcal{G} = (\mathcal{N}, \mathcal{E})$ to the above S-CNN. Let the input, hidden and output nodes be $\mathcal{N}_0 = [1] \times [w]$, $\mathcal{N}_1 = [w]$ and $\mathcal{N}_2 = \{o_\mathcal{G}\} = \{\emptyset\}$, resp. and $\mathcal{N} = \mathcal{N}_0 \cup \mathcal{N}_1 \cup \mathcal{N}_2$. Moreover, $uv \in \mathcal{E}$ if $u = o_\mathcal{G}$ and $v \in \mathcal{N}_1$ or $u = (i,) \in \mathcal{N}_1$ and $v = (0,i) \in \mathcal{N}_0$. Let $\mathbb{H}_v = \mathbb{R}^p$ for $v \in \mathcal{N}_0$, $\mathbb{H}_u = \mathbb{R}^n$ if $u \in \mathcal{N}_1$ and $\mathbb{H}_{o_\mathcal{G}} = \mathbb{R}$. The associated linear operators are given by

$$\mathcal{L}_{uv} = p^{-\frac{1}{2}}(\omega_{1,j,\beta})_{j \in [n], \beta \in [p]} \in \mathbb{R}^{n \times p} \quad \text{for} \quad (u,v) \in \mathcal{N}_1 \times \mathcal{N}_0 \tag{19}$$

$$\mathcal{L}_{o_\mathcal{G} v} = (wn)^{-\frac{1}{2}}(\omega_{2,ij})_{j \in [n]} \in \mathbb{R}^n \quad \text{if} \quad v = (i,) \in \mathcal{N}_1 \tag{20}$$

Note that the weights are shared in the first layer but not in the readout layer (i.e., the network has no pooling layer). To compute the NNGP kernel and NTK, we initalize all parameters $\omega_{1,i,\beta}$ and $\omega_{2,ij}$ with iid Gaussian $\mathcal{N}(0,1)$. Letting $n \to \infty$ and denoting $\boldsymbol{t}_v = \boldsymbol{x}_v^T \boldsymbol{x}_v'/p$ and $\boldsymbol{t} = (\boldsymbol{t}_v)_{v \in \mathcal{N}_0}$, we have

$$\mathscr{K}_\mathcal{G}(\boldsymbol{t}) = \fint_{v \in \mathcal{N}_0} \phi^*(\boldsymbol{t}_v) \quad \text{and} \quad \Theta_\mathcal{G}(\boldsymbol{t}) = \fint_{v \in \mathcal{N}_0} \phi^*(\boldsymbol{t}_v) + \dot{\phi}^*(\boldsymbol{t}_v) \tag{21}$$

**D-CNN.** The input space is $\mathcal{X} = (\overline{\mathbb{S}}_{p-1})^{k^L \times w} \subseteq \mathbb{R}^{p \times 1 \times k^L \times w}$, where $p$ is the patch size of the input convolutional layer, $k$ is the filter size in *hidden* layers, $L$ is the number of *hidden* convolution

layers and $w$ is the spatial dimension of the penultimate layer. The total dimension of the input is $d = p \cdot k^L \cdot w$, and the number of input nodes is $|\mathcal{N}_0| = k^L \cdot w$. Since the stride is equal to the filter size for all convolutional layers, the *spatial* dimension is reduced by a factor of $p$ in the first layer, a factor of $k$ by each hidden layer, and is reduced to 1 by the *Flatten-Dense* layer. Similar to S-CNNs, one can associate a DAG to a D-CNN. Briefly, the input layer has $k^L \times w$ nodes and is reduced by a factor of $k$ by each convolutional layer. The penultimate and output layers have $w$ and 1 nodes, resp.

## 4 MAIN RESULTS

The goal is to obtain a precise charaterization of the relation between the eigenstructures of $\mathcal{K}$ / $\Theta$ and the DAG associated to the network's architectures in the large input dimension setting. As such we consider a sequence of graphs $\mathcal{G} = \left( \mathcal{G}^{(d)} \right)_{d \in \mathbb{N}}$, where $\mathcal{G}^{(d)} = (\mathcal{N}^{(d)}, \mathcal{E}^{(d)})$. We associate a *finite* set of non-negative numbers $\Lambda_{\mathcal{G}}$ to $\mathcal{G}$, which is called the shape parameters of $\mathcal{G}$,

$$0 \in \Lambda_{\mathcal{G}} \subseteq [0,1] \quad \text{and} \quad |\Lambda_{\mathcal{G}}| < \infty. \tag{22}$$

We need several technical assumptions on $\mathcal{G}$ regarding the asymptotic shapes of $\mathcal{G}^{(d)}$, which are summarized as **Assumption-$\mathcal{G}$** in Sec.G of the appendix. We list two of them which are the most crucial ones. (1) For each *non-input* node $u \in \mathcal{N}^{(d)}$, there is $\alpha_u \in \Lambda_{\mathcal{G}}$ with $\deg(u) \sim d^{\alpha_u}$. The weight associated to the edge $uv \in \mathcal{E}^{(d)}$ is defined to be $\pi_{uv} \equiv \alpha_u$. (2) For each input node $v$, there is $0 < \alpha_v \in \Lambda_{\mathcal{G}}$ so that the input dimension $d_v \sim d^{\alpha_v}$. Here $a \sim b$ means $a/b \in [1/C, C]$ for some $C > 0$ independent of $d$. The main purpose of making these two assumptions is to remove non-leading terms when computing the spectra.

We say $\phi^*$ is semi-admissible if, for all $r \geq 1$, the $r$-th derivative of $\phi^*$ at zero is non-vanishing, i.e., $\phi^{*(r)}(0) > 0$. If, in addition, $\phi^*(0) = 0$ (i.e., the activation is centered), then we say $\phi$ is admissible. An activation $\phi$ is (semi-)admissible if $\phi^*$ is (semi-)admissible. Note that if $\phi^*$ is (semi-)admissible, then $\dot{\phi}^*$ is semi-admissible.

**Assumption-$\phi$.** We make the following assumptions on the activations. (a.) If $u \in \mathcal{N}_0^{(d)}$, $\phi_u$ is the identity function. (b.) If $u \notin \mathcal{N}_0^{(d)} \cup \{o_{\mathcal{G}}\}$, $\phi_u$ is admissible. (c.) If $u = o_{\mathcal{G}}$, $\phi_u$ semi-admissible.

Next, we introduce the key concept which defines the *spatial distance* among nodes. It is the length of the minimum spanning tree (MST) of the nodes.

**Definition 1** (Spatial Index of Nodes). *Let $n \subseteq \mathcal{N}^{(d)}$. The spatial index of $n$ is defined to be*

$$\mathcal{S}(n) = \min_{n \subseteq \mathcal{T} \leq \mathcal{G}^{(d)}} \sum_{uv \in \mathcal{E}(\mathcal{T})} \pi_{uv} = \min_{n \subseteq \mathcal{T} \leq \mathcal{G}^{(d)}} \sum_{uv \in \mathcal{E}(\mathcal{T})} \deg(u; \mathcal{T}) \alpha_u \tag{23}$$

*where $n \subseteq \mathcal{T} \leq \mathcal{G}^{(d)}$ means $\mathcal{T}$ is a sub-graph containing $n$ and $\deg(u; \mathcal{T})$ is the degree of $u$ in $\mathcal{T}$. By default, $\mathcal{S}(n) = 0$ if $n$ contains only one or zero node.*

Let $\boldsymbol{t^r} : I^{|\mathcal{N}_0^{(d)}|} \to I$ be a monomial, where $\boldsymbol{r} : \mathcal{N}_0^{(d)} \to \mathbb{N}^{|\mathcal{N}_0^{(d)}|}$. We use $n(\boldsymbol{r}) = \{v \in \mathcal{N}_0^{(d)} : r_v \neq 0\}$ to denote the support of $\boldsymbol{r}$ and $n(\boldsymbol{r}; o_{\mathcal{G}}) = n(\boldsymbol{r}) \cup \{o_{\mathcal{G}}\}$.

**Definition 2** (Spatial, Frequency and Learning Indices of $\boldsymbol{t^r}$). *We say $\boldsymbol{r} \in \mathbb{N}^{|\mathcal{N}_0^{(d)}|}$ is $\mathcal{G}^{(d)}$-learnable, or learnable for short, if there is a common ancestor node $u$ of $n(\boldsymbol{r})$ such that $\phi_u$ is semi-admissible. We use $\mathbb{A}(\mathcal{G}^{(d)}) \equiv \{\boldsymbol{r} \in \mathbb{N}^{\mathcal{N}_0^{(d)}} : \boldsymbol{r} \text{ is learnable}\}$. For $\boldsymbol{r} \in \mathbb{A}(\mathcal{G}^{(d)})$, the spatial index, frequency index and the learning index are defined to be,*

$$\mathcal{S}(\boldsymbol{r}) := \mathcal{S}(n(\boldsymbol{r}; o_{\mathcal{G}})), \quad \mathcal{F}(\boldsymbol{r}) := \sum_{v \in \mathcal{N}_0^{(d)}} r_v \alpha_v \quad \text{and} \quad \mathcal{L}(\boldsymbol{r}) := \mathcal{S}(\boldsymbol{r}) + \mathcal{F}(\boldsymbol{r}), \tag{24}$$

*resp. If $\boldsymbol{r} \notin \mathbb{A}(\mathcal{G}^{(d)})$, we set $\mathcal{S}(\boldsymbol{r}) = \mathcal{F}(\boldsymbol{r}) = \mathcal{L}(\boldsymbol{r}) = +\infty$. Let $\mathscr{L}(\mathcal{G}^{(d)})$ denote the sequence of learning indices in non-descending order, i.e.*

$$\mathscr{L}(\mathcal{G}^{(d)}) \equiv \left( \mathcal{L}(\boldsymbol{r}) : \boldsymbol{r} \in \mathbb{A}(\mathcal{G}^{(d)}) \right) \equiv (\cdots \leq r_j \leq r_{j+1} \leq \dots) \tag{25}$$

Finally, for each $u \in \mathcal{N}_0^{(d)}$, let $\{\overline{Y}_{r,l}\}_{l \in [N(d_u,r)], r \in \mathbb{N}}$ be the family of normalized spherical harmonics in $\overline{\mathbb{S}}_{d_u-1}$, where $N(d_u, r)$ is the number of degree $r$ spherical harmonics in $\mathbb{S}_{d_u-1}$. Define

$$\overline{Y}_{r,l}(\xi) = \prod_{u \in \mathcal{N}_0^{(d)}} \overline{Y}_{r_u, l_u}(\xi_u), \quad l = (l_u)_{u \in \mathcal{N}_0^{(d)}} \in [N(d,r)] \equiv \prod_{u \in \mathcal{N}_0^{(d)}} [N(d_u, r_u)] \quad (26)$$

for $\xi = (\xi_u) \in \mathcal{X}$. The following is our main theorem. It describes a connection between the architecture of a network and the eigenstructure of its inducing kernels.

**Theorem 1 (Eigenspace Restructuring).** *Assume **Assumption-$\mathcal{G}$** and **Assumption-$\phi$**. We have the following eigen-decomposition for $\mathcal{K} = \mathcal{K}_{\mathcal{G}^{(d)}}$ or $\Theta_{\mathcal{G}^{(d)}}$. For $\xi, \eta \in \mathcal{X}$*

$$\mathcal{K}(\xi, \eta) = \sum_{r \in \mathbb{N}^{|\mathcal{N}_0^{(d)}|}} \lambda_{\mathcal{K}}(r) \sum_{l \in N(d,r)} \overline{Y}_{r,l}(\xi) \overline{Y}_{r,l}(\eta), \quad \text{where} \quad \lambda_{\mathcal{K}}(r) \sim d^{-\mathcal{L}(r)} \text{ if } r \neq 0. \quad (27)$$

Coupling with the observation in Eq. (2), Theorem 1 implies that within $t \sim d^r$ amount of time for gradient flow, when $d$ is sufficiently large, only the eigenfunctions $\overline{Y}_{r,l}$ with $\mathcal{L}(r) \leq r$ can be learned. By leveraging an analytical result from Mei et al. (2021a) (Sec. 3 Theorem 4), which says under certain regularity conditions on the eigenstructure, the corresponding kernel regression acts as a projection, Theorem 1 establishes a connection between architectures and generalization bounds of NNGP kernel/NTK.

Let $\sigma$ be the uniform (product) probability measure on $\mathcal{X}$ and denote $L^p(\mathcal{X}) \equiv L^p(\mathcal{X}, \sigma)$. For $X \subseteq \mathcal{X}$ and $r \notin \mathcal{L}(\mathcal{G}^{(d)})$, define the regressor and the projection operator to be

$$R_X(f)(x) = \mathcal{K}(x, X)\mathcal{K}(X, X)^{-1} f(X) \text{ and } P_{>r}(f) = \sum_{r:\mathcal{L}(r)>r} \sum_{l \in N(d,r)} \langle f, \overline{Y}_{r,l} \rangle_{L^2(\mathcal{X})} \overline{Y}_{r,l}.$$

**Theorem 2.** *Let $\mathcal{G} = \{\mathcal{G}^{(d)}\}_d$, where each $\mathcal{G}^{(d)}$ is a DAG associated to the D-CNN in Eq. (13). Let $r \notin \mathcal{L}(\mathcal{G}^{(d)})$ be fixed. Let $f \in L^2(\mathcal{X})$ with $\mathbb{E}_\sigma f = 0$. Then for $\epsilon > 0$,*

$$\left| \|R_X(f) - f\|_{L^2(\mathcal{X})}^2 - \|P_{>r}(f)\|_{L^2(\mathcal{X})}^2 \right| = c_{d,\epsilon} \|f\|_{L^{2+\epsilon}(\mathcal{X})}^2, \quad (28)$$

*where $c_{d,\epsilon} \to 0$ in probability as $d \to \infty$ over $X \sim \sigma^{[d^r]}$.*

In words, with $[d^r]$ many training samples where $r \notin \mathcal{L}(\mathcal{G}^{(d)})$, the NNGP kernel and the NTK are able to learn all $\overline{Y}_{r,l}$ with $\mathcal{L}(r) < r$ but not any eigenfunctions with $\mathcal{L}(r) > r$. In Sec. C, we show that the number of training samples can be reduced by a factor of $w$ if the readout layer *Fattening* is replaced by the global average pooling.

## 5 INTERPRETATION OF THE MAIN RESULTS

We say $r$ is the budget index if (1) (**Finite Training Set**) the training set size $m \sim d^r$, or (2) (**Finite Compute Time**) the training set $X = \mathcal{X}$ and the total number of training steps/time $t \sim d^r$.

**MLPs and LRLF Interactions Fig. 1 (a).** Each $\mathcal{G}^{(d)}$ is a linked list. Let $v$ be the input node. Clearly, we have $d_v = d$, i.e., $\alpha_v = 1$ and $\alpha_u = 0$ (since $\deg(u) = 1$) for all other nodes $u$. As such

$$\mathcal{A}(\mathcal{G}^{(d)}) = \mathbb{N} \backslash \{0\}, \quad \mathcal{F}(r) = |r|, \quad \mathcal{S}(r) = 0, \quad \mathcal{L}(r) = |r| \text{ and } \mathcal{L}_{\mathcal{G}^{(d)}} = (|r| : r \in \mathbb{N} \backslash \{0\}). \quad (29)$$

There isn't any *spatial* structure since $\mathcal{S}(r) = 0$. Given a budget index $r$, the kernels can only learn $\text{span}\{\overline{Y}_{r,l} : r < r, r \in \mathbb{N} \backslash \{0\}\}$. Therefore, MLPs are good at modeling LRLF interactions.

**S-CNNs and SRHF Interactions Fig. 1 (b).** For one-hidden layer convolutional networks, we have $d = p \times k^0 \times w$, (i.e. $L = 0$). The number of input nodes is $w \equiv d^{\alpha_w}$ and for each input node $v$ we have $d_v = p \equiv d^{\alpha_p}$. We have $\alpha_p + \alpha_w = 1$. Since the activation function of the last layer is the identity function, there is no non-linear interactions between different patches and $\mathcal{A}(\mathcal{G}^{(d)}) = \{r \in \mathbb{N}^{|\mathcal{N}_0^{(d)}|} \backslash \{0\}, |n(r)| = 1\}$. Therefore for $r \in \mathcal{A}(\mathcal{G}^{(d)})$,

$$\mathcal{S}(r) = \alpha_w, \quad \mathcal{F}(r) = |r|\alpha_p, \quad \mathcal{L}_{\mathcal{G}} = \{\alpha_w + |r|\alpha_p, r \in \mathcal{A}(\mathcal{G}^{(d)})\} \quad (30)$$

Given a budget index $r$, the learnable $\boldsymbol{r}$ are the ones $\boldsymbol{r} \in \mathscr{A}(\mathcal{G}^{(d)})$ with

$$r > \alpha_w + |\boldsymbol{r}|\alpha_p = 1 + (|\boldsymbol{r}| - 1)\alpha_p \Rightarrow |\boldsymbol{r}| < 1 + (r-1)/\alpha_p. \tag{31}$$

When $\alpha_p = 1$ (and thus $\alpha_w = 0$), this S-CNN is essentially a shallow MLP and we have $|\boldsymbol{r}| < r$. On the other hand, if $\alpha_p$ is small (say $\alpha_p = 0.1$), this network can model interactions with much higher frequencies, at the cost of giving up long-range interactions. Therefore, there is a trade-off between space and frequency, and S-CNNs with small patch size are good at modeling SRHF interactions.

**D-CNNs and Interterpolation Fig. 1 (c).** Recall that $d = p \times k^L \times w$. Let $p = d^{\alpha_p}, k = d^{\alpha_k}$ and $w = d^{\alpha_w}$. Then we have the constraint $\alpha_p + L\alpha_k + \alpha_w = 1$. The learnable terms are the ones with

$$r > \mathcal{S}(\boldsymbol{r}) + \mathcal{F}(\boldsymbol{r}) = \mathcal{S}(\boldsymbol{r}) + |\boldsymbol{r}|\alpha_p. \tag{32}$$

D-CNNs can simultaneously model interpolations (including the two ends) between LRLF and SRHF, i.e. MRMF interactions. Consider two extreme cases. (i.) $|n(\boldsymbol{r})| = 1$, i.e. there is an input node $v$ s.t. $\boldsymbol{r}_v = |\boldsymbol{r}|$ and thus the non-linear interaction happens within one patch. In this case, the length of the MST reaches its infimum $\mathcal{S}(\boldsymbol{r}) = \alpha_w + L\alpha_k = 1 - \alpha_p$ and Eq. (32) implies $|\boldsymbol{r}| < (r-1)/\alpha_p + 1$, which is exactly Eq. (31). Thus D-CNNs can model SRHF interactions; see Fig. 6 $\mathbf{Y_5^*}$. (ii.) $|n(\boldsymbol{r})| = |\boldsymbol{r}|$, i.e. $\boldsymbol{r}_v = 0$ or 1 for all $v \in \mathcal{N}_0^{(d)}$. Then $\mathcal{S}(\boldsymbol{r}) = |\boldsymbol{r}|(1 - \alpha_p)$ and Eq. (32) implies $|\boldsymbol{r}| < r$, which is the constraint for MLPs. In particular, D-CNNs can model LRLF interactions; see $\mathbf{Y_5}$. By varying $|\boldsymbol{r}|$ and then varying $|n(\boldsymbol{r})|$ in $[1, |\boldsymbol{r}|]$, D-CNNs can model various interpolating interactions between LRLF and SRHF.

**HR-CNNs Extrapolations and Finer Interpolations Fig. 1 (d).** The resolution of the learning indices $\mathcal{L}(\mathcal{G}^{(d)})$ can be improved by decreasing $\alpha_k, \alpha_p$ and increasing $L$ accordingly. E.g., changing $\alpha_p \rightarrow \frac{\alpha_p}{2}, \alpha_k \rightarrow \alpha_k/2$ and doubling the number of convolutional layers accordingly, then the resolution of the range of spatial/frequency/learning indices is doubled. This empowers the network to model finer-grained interpolating modes, and extrapolating modes. E.g., the last equation in Eq. (31) becomes $|\boldsymbol{r}| < 1 + 2(r-1)/\alpha_p$ (almost doubles the upper bound of $|\boldsymbol{r}|$) and the network can additionally model **Ultra-Short-Range-Ultra-High-Frequency** interactions (see $\mathbf{Y_5^*}$ in Fig. 3) without sacrificing its expressivity, which isn't the case for S-CNNs due to the space-frequency trade-off. We refer to such networks as high-resolution CNNs (HR-CNNs). For practical networks, e.g. ResNet (He et al., 2016), the filters/patches are already quite small and there isn't much room to reduce them. Equivalently, one increases the resolution of the input images instead, i.e. increasing $d$. From this point of view, HR-CNNs and, therefore, our theorems justify the additional performance gain from EfficientNet-type (Tan & Le, 2019) of model scaling.

For more details regarding computing the learning index, see Sec. D.

## 6 EXPERIMENTS

There are many practical consequences due to Theorem 1 and Theorem 2. We focus on two of them which are about *the impact of architectures to learning / generalization*:

1. **Order of Learning (Fig. 1 Green Arrow.)** The order of learning is restructured from frequency-based (MLPs) to space-and-frequency-based (CNNs).

2. **Learnability (Fig. 1 Cells under the budget line.)** With the same budget index, MLP-Learnable $\subsetneq$ D-CNN-Learnable $\subsetneq$ HR-CNN-Learnable. Moreover, the set differences between these learnable sets are captured as in Sec.5.

Overall, we see excellent agreements between predictions from our theorems and experimental results from both practical-size networks and kernel methods using NNGP/NT kernels, even when $d = 256$ is moderate-size. We detail the setup, results, corrections, etc. for the experiments below.

**Setup.** Set $d = p^4$ and the input $\boldsymbol{\mathcal{X}} = (\overline{\mathbb{S}}_{p-1})^{p^3} \subseteq \mathbb{R}^{p^4}$, where $p \in \mathbb{N}$. Note that $\alpha_p = 1/4$. The task is learning a function $Y \in L^2(\boldsymbol{\mathcal{X}})$ by minimizing the MSE, where

$$Y = \mathbf{Y_1} + \mathbf{Y_2} + \mathbf{Y_3} + \mathbf{Y_4} + \mathbf{Y_5} + \mathbf{Y_5^*} + \mathbf{Y_6} + \mathbf{Y_7} \tag{33}$$

and each eigenfunction $Y_i$ is normalized so that $\|Y_i\|_2^2 = \|Y\|_2^2/8 = 1$. The exact expressions of the functions can be found in Sec.D.

We optimize finite-width networks by *SGD+Momentum* and infinite-width networks (NNGP and NTK) by kernel regression. We investigate three types of architectures: (1) Dense$^{\otimes 4}$, a four hidden layer MLP; (2) Conv$(p^2)^{\otimes 2}$, a "deep" CNN with filter size/stride $k = p^2$, and (3) Conv$(p)^{\otimes 4}$, a "HS"-CNN with filter size/stride $k = p$. See Fig. 2 and Fig. 6 for a visualization of the associated DAGs, and Sec. F for the code of the architectures. There is an activation $\phi$ in each *hidden* layer, which is chosen so that $\phi^*$ is the Gaussian kernel. For the CNNs, the readout layer(s) is *Flatten-Dense-Act-Dense*. We carefully chose the eigenfunctions $\{Y_i\}$ so that they cover a wide range of space-frequency combinations $(\mathcal{S}(Y_i), \mathcal{F}(Y_i))$ w.r.t. Conv$(p)^{\otimes 4}$. Under Conv$(p)^{\otimes 4}$, the corresponding learning indices are $\mathcal{L}(Y_i) = \mathcal{S}(Y_i) + \mathcal{F}(Y_i) = 3\alpha_p + i\alpha_p = (3+i)/4$. For the learning indices of $Y_i$ under Conv$(p^2)^{\otimes 2}$ or Dense$^{\otimes 4}$, see the legends in Fig.3. The purpose of doing so is to create a "separation of learning" under Conv$(p)^{\otimes 4}$, since in the large $p$ limit, learning $Y_i$ requires $d^{(3+i)/4+\epsilon}$ examples/SGD steps. The relation between architectures and learning indices can be "visualized" in Fig. 2 for $\mathbf{Y_2}$, which is short-range-low-frequency ( $\deg(\mathbf{Y_2}) = 2$). Fig. 2 (d) plots the test residual of $\mathbf{Y_2}$ vs training set size for NTK regression, as one example to showcase the relation among architectures, learning indices and generalization. See Fig. 6 in the appendix for other eigenfunctions.

In the experiments, the width/number of channels is set to 512 for all networks. We sample $m_t = 32 \times 10240$ ($m_v = 10240$) data points randomly from $\mathcal{X}$ as training (test) set with $p = 4$ and $d = 256$. The SGD training configurations (batch size (=10240), learning rate (=1.), momentum (=0.9) etc.) are identical across architectures. To compute the kernels, we rely crucially on *NeuralTangents* (Novak et al., 2020) which is based on *JAX* (Bradbury et al., 2018).

In Fig.3, for each eigenfunction $Y_i$, we plot $\frac{1}{2}\mathbb{E}|\hat{Y}_i(x,t) - Y_i(x)|_2^2$ against $t$, where $\hat{Y}_i(x,t)$ is the projection of the prediction onto $Y_i$ and $t$ is either the training steps (SGD) or training set size (kernels). The expectation is taken over the test set. The budget index $r = \log(m_t)/\log(d) \approx 2.28$. As $d = 256$ ($p = 4$) is far from the asymptotic limit, we expect $r = 2.28$ being a *soft* cut-off between learnable and non-learnable indices. Although the theorems assume $d, p \to \infty$, they do provide good predictions even when $d$ and $p$ are far from $\infty$. We summarize several key observations below.

1. **Dense$^{\otimes 4}$** (1nd Row.) This architecture can capture all low-frequency interactions ($\deg = 1, 2, \mathbf{Y_1}, \mathbf{Y_2}, \mathbf{Y_3}, \mathbf{Y_5}$) but fail to learn $\deg \geq 3$ interactions, as expected. For MLPs, making the network deeper won't improve its learnability much; see Fig. 7.

2. **Conv$(p^2)^{\otimes 2}$** (2nd Row.) Learning curves of $\mathbf{Y_2}/\mathbf{Y_3}$ are separated from $\mathbf{Y_5}$ because the spatial indices of them are different. Higher-frequency ($\deg = 3, 4$) shorter-range interactions ($\mathbf{Y_4}, \mathbf{Y_6}$) become (partially) learnable, as $\mathcal{L}(\mathbf{Y_4}) = \frac{8}{4}, \mathcal{L}(\mathbf{Y_6}) = \frac{10}{4} < r \approx 2.28$.

3. **Conv$(p)^{\otimes 4}$** (3rd Row). We see $\mathcal{L}(Y_i)$ capture the order of learning very/reasonably well in the kernel/SGD setting. To test the ability of Conv$(p)^{\otimes 4}$ in modeling ultra-short-range-ultra-high-frequency interactions, we trace the learning progress of $\mathbf{Y_5^*}$ ($\deg(\mathbf{Y_5^*}) = 5, \mathcal{L}(\mathbf{Y_5^*}) = \frac{8}{4}$.) As expected, while other architectures completely fail to make progress, the NTK/NNGP of Conv$(p)^{\otimes 4}$ makes good progress and the SGD even completes the learning process. Interestingly, $\mathbf{Y_5}$[1] is learned faster than $\mathbf{Y_5^*}$ in the kernel setting but slower in the SGD setting (even slower than $\mathcal{L}(\mathbf{Y_6}) = \frac{10}{4}$), which is unexpected. We suspect it might be due to certain "implicit" effect of SGD. Further investigation is needed to understand it.

## 7 CONCLUSION

We establish a precise relation among networks' architectures, eigenstructures of the inducing kernels, and generalization of the corresponding kernel machines. We show that deep convolutional networks restructure the eigenspaces of the inducing kernels, which empowers them to learn a dramatically broader class of functions, covering a wide range of space-frequency combinations. We believe our framework can be extended to study architectural inductive biases for other families of topologies, such as RNNs, GNNs, and self-attention. However, we have not covered the learning

---

[1]Ultra-Long-Range-Low-Frequency under Conv$(p)^{\otimes 4}$, with $\mathcal{L}(\mathbf{Y_5}) = 8/4 = \mathcal{L}(\mathbf{Y_5^*})$

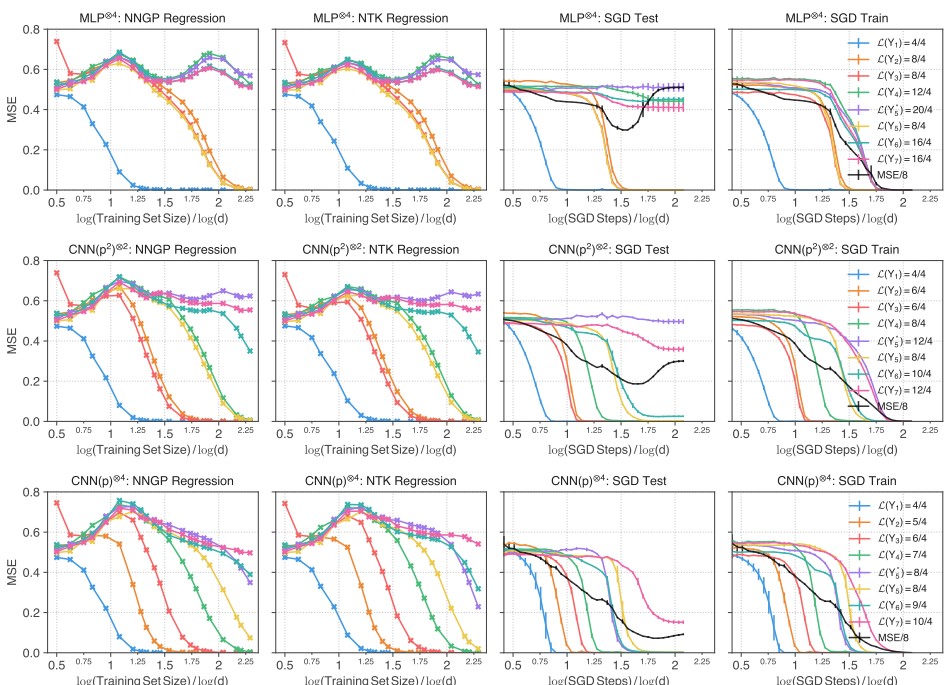

Figure 3: **Learning Dynamics vs Architectures vs Learning Indices.** We plot the learning/training dynamics of each eigenfunction $Y_i$. From top to bottom: a 4-layer MLP, a 2-layer CNN and a 4-layer CNN. From left to right: residual MSE (per eigenfunction) of NNGP/NTK regression, test/training MSE of SGD. The learning indices of $Y_i$ in each architecture is shown in the legends.

dynamics of SGD. In addition, it is of great interest and importance to study the combined effect of SGD and architectures in the future.

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

## A    ADDITIONAL RELATED WORK

Bietti (2021) studies the approximation and learning properties of CNN kernels via the lens of RKHS. Impressively, they demonstrate that a 2-layer CNN kernel can reach 88.3% validation accuracy on CIFAR-10, matching the performance of a 10-layer Myrtle kernel Shankar et al. (2020). Malach & Shalev-Shwartz (2020) and Li et al. (2020) study the algorithmic benefits of shallow CNNs and show that they outperform MLPs in certain tasks. Xiao & Pennington (2021) and Favero et al. (2021) study the benefits of locality in S-CNNs and argue that locality is the key ingredient to defeat the curse of dimensionality. Mei et al. (2021b) and several papers mentioned above study the benefits of pooling in (S-)CNNs in terms of data efficiency. Their conclusion is similar to that of Theorem 4 (b): pooling improves data efficiency by a factor of the pooling size. In addition, we show that (Theorem 4 (a)) pooling does not improve training efficiency for D-CNNs, extending a result from Xiao & Pennington (2021) which concerns S-CNNs. Finally, Scetbon & Harchaoui (2020) also study the eigenstructures of certain CNN kernels without pooling. Their kernels can be considered as a particular case of the NNGP kernels, where the associated networks have only one convolutional layer and multiple dense layers. The key contribution that sets the current work apart from existing work is a precise mathematical characterization of the fundamental role of architectures in (infinite-width) networks through a space-frequency analysis.

## B    A TOY EXAMPLE AND MOTIVATION

In this section, we provide a toy example to help understand the motivation and ideas of the paper. Let's consider learning the following polynomials in $\mathbb{S}_{d-1} \equiv \{x \in \mathbb{R}^d : \|x\|_2 = 1\}$ using (in)finite-width neural networks and for concreteness we have set $d = 10$:

$$f_1(x) = x_9, \ f_2(x) = x_0 x_1, \ f_3(x) = x_0 x_8, \ f_4(x) = x_6 x_7 (x_6^2 - x_7^2), \ f_5(x) = x_2 x_3 x_5 \quad (34)$$

Which architectures (e.g. MLPs, CNNs) can efficiently learn $f_i$ or the sum of $f_i$? More precisely, (1) if we have sufficiently amount of training data, how much time (compute) is required to learn $f_i$ for a given architecture? (2) Alternatively, if we have sufficiently amount of compute, how much data is needed to learn $f_i$? To answer these questions, one crucial step is to provide a meaningful definition of "learning complexity" of a function $f_i$ under an architecture $\mathcal{M}$. Denote this complexity associated to compute and to data by $\mathcal{C}_C(f_i; \mathcal{M})$ and $\mathcal{C}_D(f_i; \mathcal{M})$, resp. With such the complexity properly defined, the questions are reduced to solving the min-max problem $\min_{\mathcal{M}} \max_i \{\mathcal{C}_{C/D}(f_i; \mathcal{M})\}$, if the task is, e.g, to learn the sum of $f_i$.

Let's focus on the complexity. For infinite-width MLPs, aka, inner product kernels, it is well-known that they have the inductive biases (Yang & Salman, 2020; Ghorbani et al., 2020) (known as the frequency biases) that the model prioritizes learning low-frequency modes (i.e., low degree polynomials) over high-frequency modes. In addition, the models require $\sim d^r$ many data points to learn *any* degree $r$ polynomials in $\mathbb{R}^d$. The frequency biases of MLPs are the consequence of the fact that the eigenspaces of inner product kernels are structured based only on frequencies. Specific to our example, for MLP, the order of learning is $f_1/f_2, f_3/f_5/f_4$ and it requires about $10/10^2, 10^2/10^4$ many data points to learn the functions. Clearly, the model is very inefficient in learning $f_4$, the high-frequency modes.

To improve the learning efficiency, we must take the modality of the task into account, which is overlooked by MLPs. We observe that: (1) although of high frequency, $f_4$ depends only on two consecutive terms $x_6$ and $x_7$, which are spatially close; (2) in contrast, $f_3(x) = x_0 x_8$ is of low frequency but the spatial distance between the two interaction terms $x_0$ and $x_8$ are "far" from each other; (3) the function $f_5(x) = x_2 x_3 x_5$ is somewhere in-between: the order of interaction is 3 (lower than that of $f_4$) and the spatial distance (not yet defined) among interaction terms is conceptually "closer" than that of $f_3(x) = x_0 x_8$, but "farther" than that of $f_5$. Using the terminologies from the introduction, the functions $f_2/f_3/f_5/f_4$ model interactions of types: Short-Range-Low-Frequency/Long-Range-Low-Frequency/Median-Range-Median-Frequency/Short-Range-High-Frequency. By *Range* we mean the distance among interaction terms and by *Frequency* we mean the order(=degree) of interactions. Clearly, the MLPs are inefficient since they totally ignore the "spatial structure" of the functions. As such, a good architecture must balance the "spatial structure" and the "frequency structure" of the functions. For the same reason, a good complexity measure must (1) be able to capture both the *frequency* of the functions and the *spatial distance* among interaction terms; (2)

be able to precisely characterize the data and the computation efficiency of learning and their dependence on architectures. The learning index mentioned in the introduction satisfies these two conditions. It is the sum of the frequency index and the spatial index. The former measures the order (=degree=frequency) of interactions, which depends on how the network partitions the input into patches. The latter measures the spatial distance among the interaction terms, which depends on how the network organizes these patches hierarchically. Later we show that, in the high-dimensional setting, the learning index provides a sharp characterization for the learnability of eigenfunctions, and certain CNNs can perfectly balance the learning of $f_3$ and $f_4$, i.e., informally

$$\mathcal{C}_{C/D}(f_3; \text{CNN}) \approx \mathcal{C}_{C/D}(f_4; \text{CNN}) \approx \mathcal{C}_{C/D}(f_{2/3}; \text{MLP}) << \mathcal{C}_{C/D}(f_3; \text{MLP}) \tag{35}$$

See Fig. 3 in the experiment section for more details.

## C  GLOBAL AVERAGE POOLING (GAP) VS FLATTENING

In this section, we compare two readout strategies for CNNs: GAP and Flatten

### C.1  INFINITE-TRAINING DATA.

First, let's state a theorem regarding training efficiency in the infinite-training-data-finite-training-time regime, which follows directly from the same arguments in Sec. 2. The theorem requires knowing only the eigenvalues of the kernels.

Let $\mathcal{F} : L^2(\mathcal{X}) \to L^2(\mathcal{X})$ be the solution operator to the kernel descent $\dot{h} = -\mathcal{K}(h - f)$ with initial value $h_{t=0} = \mathbf{0}$, i.e. $h_t \equiv \mathcal{F}_t(f) \equiv (\mathbf{Id} - e^{-\mathcal{K}t})f$. Then we have for $t \sim d^r$, $\mathcal{F}_t \approx \boldsymbol{P}_{<r}$.

**Theorem 3.** *Assume* **Assumption-$\mathcal{G}$** *and* **Assumption-$\phi$** *and* $\mathcal{K} = \mathfrak{K}_{\mathcal{G}^{(d)}}$ *or* $\Theta_{\mathcal{G}^{(d)}}$. *Let* $r \notin \mathcal{L}(\mathcal{G}^{(d)})$ *and* $t \sim d^r$. *Then for* $0 < \epsilon < \inf\{|r - \bar{r}| : \bar{r} \in \mathcal{L}(\mathcal{G}^{(d)})\}$ *and* $f \in L^2(\mathcal{X})$ *with* $\mathbb{E}_{\boldsymbol{\sigma}} f = 0$ *we have*

$$\|\mathcal{F}_t(\boldsymbol{P}_{<r}f) - \boldsymbol{P}_{<r}f\|_2^2 \lesssim e^{-d^\epsilon}\|\boldsymbol{P}_{<r}f\|_2^2 \quad \text{and} \quad \|\mathcal{F}_t(\boldsymbol{P}_{>r}f) - \boldsymbol{P}_{>r}f\|_2^2 \gtrsim e^{-d^{-\epsilon}}\|\boldsymbol{P}_{>r}f\|_2^2 \tag{36}$$

*for d sufficiently large.*

In words, in the infinite-training-data-finite-training-time regime, within $t \sim d^r$ amount of time only the eigenfunctions $\overline{\boldsymbol{Y}}_{\boldsymbol{r},\boldsymbol{l}}$ with $\mathcal{L}(\boldsymbol{r}) < r$ are learnable.

### C.2  GAP VS FLATTENING.

Let a CNN with and without a global average pooling (GAP) be defined as

$$\textbf{CNN+GAP} \qquad [Input] \to [Conv(p)\text{-}Act] \to [Conv(k)\text{-}Act]^{\otimes L} \to [GAP] \to [Dense] \tag{37}$$

$$\textbf{CNN+Flatten} \qquad [Input] \to [Conv(p)\text{-}Act] \to [Conv(k)\text{-}Act]^{\otimes L} \to [Flatten] \to [Dense] \tag{38}$$

resp. Note that there isn't any activation after the GAP/Flatten layer. The DAGs associated to *CNN+GAP* and *CNN+Flatten* are identical, and the associated kernel and network computations in each layer are also identical but the last layer; see Sec. K for more details. Our last theorem states that the GAP improves the data efficiency by a factor of $w$, the window size of the pooling, but does not improve the training efficiency in the infinite-training-data-finite-training-time regime. The former is due to the dimension reduction effect of GAP in the eigenspaces and the latter is due to the fact that the GAP layer does not change the eigenvalues of the associated eigenspaces.

Let $\mathcal{K}_{\text{Sym}} = \mathfrak{K}_{\text{Sym}}$ or $\Theta_{\text{Sym}}$ be the NNGP kernel or NTK associated to Eq. (37) and $L^p_{\text{Sym}}(\mathcal{X}) \leq L^p(\mathcal{X})$ be the subspace of "translation-invariant" functions, whose co-dimension is $w$. Let $\mathcal{F}^{\text{Sym}}$, $\boldsymbol{P}^{\text{Sym}}$ and $\boldsymbol{R}^{\text{Sym}}$ be the solution operator, projection operator and regressor associated to $\mathcal{K}_{\text{Sym}}$, resp.

**Theorem 4** (Informal). *For the architectures defined as in by Eq. (37), we have*

(a) *Theorem 3 holds with* $L^2(\mathcal{X})$, $\mathcal{F}$ *and* $\boldsymbol{P}$ *replaced by* $L^2_{Sym}(\mathcal{X})$, $\mathcal{F}^{Sym}$ *and* $\boldsymbol{P}^{Sym}$, *resp.*

(b) *Eq. (28) holds with* $L^p(\mathcal{X})$, $\boldsymbol{R}$ *and* $\boldsymbol{P}$ *replaced by* $L^p_{Sym}(\mathcal{X})$, $\boldsymbol{R}^{Sym}$ *and* $\boldsymbol{P}^{Sym}$, *resp and with* $X \sim \boldsymbol{\sigma}^{[d^{r-\alpha_w}]}$ *under the assumptions that all activations in the hidden layers are poly-admissible.*

**Remark 1.** *Several remarks are in order.*

1. *In terms of order of learning, our results say that (infinite-width) neural networks progressively learn more complex functions, where complexity is defined to be the learning index $\mathcal{L}(\boldsymbol{r})$. This is consistent with the empirical observation from Nakkiran et al. (2019). Regardless of architectures (MLPs vs CNNs), linear functions have the smallest learning index[2] $\mathcal{L}(\boldsymbol{r}) = 1$ and are always learned first, which was first proved in Hu et al. (2020) for MLPs. After learning the linear functions, the learning dynamics of MLPs and CNNs diverge. MLPs will start to learn quadratic functions ($\mathcal{L}(\boldsymbol{r}) = 2$), then cubic functions ($\mathcal{L}(\boldsymbol{r}) = 3$) and so on. While for CNNs, $\mathcal{L}(\boldsymbol{r})$ can be fractional numbers (e.g. $\mathcal{L}(\boldsymbol{r}) = 5/4, 6/4$) and it is possible for the network to learn higher order functions before lower order functions. See Sec.D and Sec. 6 for more details.*

2. *By NTK-style convergent arguments (e.g., Du et al. (2019)), the above kernel regression results could be extended to the over-parameterized network setting as long as the learning rate of gradient descent is small and the number of channels is sufficiently large (polynomially in $d$ (Huang et al., 2020)).*

3. *Theorem 4 states that CNN+GAP is better than CNN+Flatten in terms of data efficiency but not training efficiency. In particular, when the dataset size is sufficiently large, CNN+GAP and CNN+Flatten perform equally well. Therefore, in the large (infinite) data set regime, CNN+Flatten may be preferable since the associated function class is less restricted.*

## C.3 Experimental Results: GAP vs Flatten.

We compare the SGD learning dynamics of two convolutional architectures: $\text{Conv}(p)^{\otimes 3}$-Flatten and $\text{Conv}(p)^{\otimes 3}$-GAP. Both networks have three convolutional layers with filter size and stride equal to $p$. The spatial dimension is reduced to $p$ after the convolutional layers. The only difference is that the architecture $\text{Conv}(p)^{\otimes 3}$-Flatten ( $\text{Conv}(p)^{\otimes 3}$-GAP) uses a Flatten-Dense ( GAP-Dense) to map the penultimate layer to the logit layer.

The experimental setup is almost the same as that of Sec. 6 except the eigenfunctions $\{Y_i\}$ are chosen to be in the RKHS of the NNGP kernel/NTK of $\text{Conv}(p)^{\otimes 3}$-GAP and thus of $\text{Conv}(p)^{\otimes 3}$-Flatten, i.e., they are shifting-invariant (the invariant group is of order $p$). Moreover, we still have $\mathscr{L}(Y_i) = (i + 3)/4$. For each $Y_i$, we plot the validation MSE of the residual vs SGD steps in Fig. 4. Overall, the predictions from Theorem 4 gives excellent agreement with the empirical result. With training set size $m_t = 32 \times 10240$, the residuals of $Y_i$ for GAP and Flatten are almost indistinguishable from each other for $i \leq 6$ (recall that $\mathscr{L}(\mathbf{Y_6}) = 9/4 = 2.25 < r \approx 2.28$). However, when $i = 7$ the dataset size ($r \approx 2.28$) is relatively small compared to the learning index ($\mathscr{L}(\mathbf{Y_7}) = 2.5$), GAP outperforms Flatten in learning $\mathbf{Y_7}$

To test the robustness of the prediction from Theorem 4 (a) on more practical datasets and models, we perform an additional experiment on ImageNet (Deng et al., 2009) using ResNet (He et al., 2016). We compare the performance of the original ResNet50, denoted by ResNet50-GAP, and a modified version ResNet50-Flatten, in which the GAP readout layer is replaced by Flatten. We use the ImageNet codebase from FLAX[3](Heek et al., 2020). In order to see how the performance difference between ResNet50-GAP and ResNet50-Flatten evolves as the training set size increases, we make a scaling plot, namely, we vary the training set sizes[4] $m_i = [m \times 2^{-i/2}]$ for $i = 0, \ldots, 11$, where $m = 1281167$ is the total number of images in the training set of ImageNet. The networks are trained for 150 epochs with batch size 128. We plot the validation accuracy and loss (averaged over 3 runs) as a function of training set size $m_i$ in Fig. 5. Overall, we see that the performance gap between ResNet50-GAP and ResNet50-Flatten shrink substantially as the training set size increases. E.g, using 1/8 of the training set (i.e. $i = 6$), the top 1 accuracy between the two is 19.3% (57.7% GAP vs 38.4% Flatten). However, with the whole training set (i.e., $m_0$), this gap is reduced to 2% (76.5% GAP vs 74.5% Flatten). To demonstrate the robustness of this trend, we additionally generate the same plots for ResNet34 and ResNet101; see Fig. 8 in Sec. E.

---

[2]Ignoring the constant functions.

[3]https://github.com/google/flax/blob/main/examples/imagenet/README.md

[4]Standard data-augmentation is applied for each $i$; see input_pipeline.py.

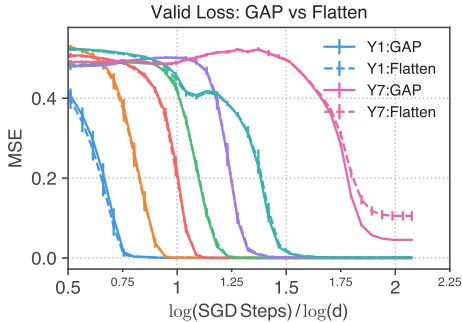

Figure 4: **Learning Dynamics: GAP vs Flatten.** We plot the validation MSE of the residual of each $Y_i$ (left → right: $i = 1 \to 7$) for GAP (Solid lines) and Flatten (Dashed lines). The mean/std in each curve is obtained by 5 random initializations. Clearly, the residual dynamics of GAP and Flatten are almost indistinguishable for $Y_i$ with $i \leq 6$. However, GAP outperforms Flatten when learning $\mathbf{Y_7}$.

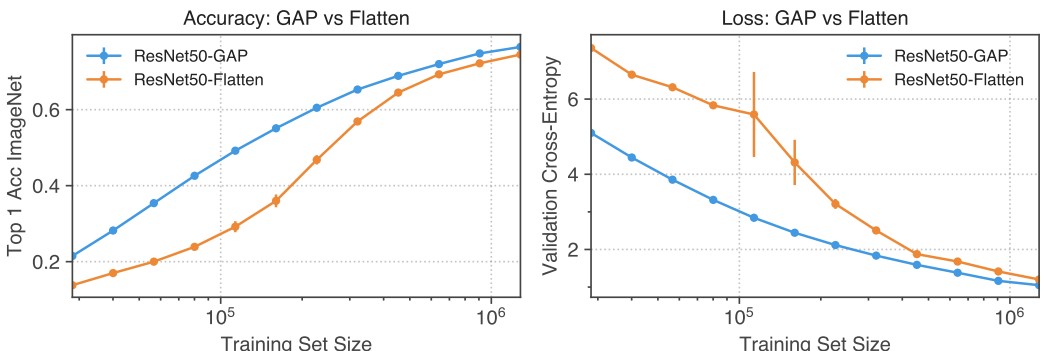

Figure 5: **ResNet50-GAP vs ResNet50-Flatten.** As the training set size increases the performance (accuracy and loss) gap between the two shrinks.

# D  DAGS, EIGENFUNCTIONS, SPATIAL INDEX, AND FREQUENCY INDEX.

In this section, we provide more details regarding the DAGs and the eigenfunctions used in the experiments, and how the spatial, frequency and learning indices are computed.

Let $(\overline{\mathbb{S}}_{p-1})^{p^3} \subseteq \mathbb{R}^{p^4}$ be the input space, where $d = p^4$ is the input dimension. We use $\boldsymbol{x} \equiv (\boldsymbol{x_k})_{\boldsymbol{k} \in [p]^4} \in (\overline{\mathbb{S}}_{p-1})^{p^3}$ to denote one input (an image), where $\boldsymbol{k} = [\boldsymbol{k}_1, \boldsymbol{k}_2, \boldsymbol{k}_3, \boldsymbol{k}_4] \in [p]^4$. In addition, we treat $[p]^4$ as a group (i.e. with circular boundaries) and let $\boldsymbol{e}_1 = [1, 0, 0, 0]$, $\boldsymbol{e}_2 = [0, 1, 0, 0]$, $\boldsymbol{e}_3 = [0, 0, 1, 0]$ and $\boldsymbol{e}_4 = [0, 0, 0, 1]$ be a set of generator/basis of the group. Note that each input $\boldsymbol{x_k}$ is partitioned into $p^3$ many patches: $\{\boldsymbol{x}_{\boldsymbol{k}_1, \boldsymbol{k}_2, \boldsymbol{k}_3, :} : \boldsymbol{k}_1, \boldsymbol{k}_2, \boldsymbol{k}_3 \in [p]\}$.

The eigenfunctions used in the experiments and the associated space/frequency indices (will be explained momentarily) are given as follow

| Eigenfunction | degree | Space/Freq Index | | |
| --- | --- | --- | --- | --- |
| | | MLP | CNN$(p^2)^{\otimes 2}$ | CNN$(p)^{\otimes 4}$ |
| $\mathbf{Y_1}(\boldsymbol{x}) = \sum_{\boldsymbol{k} \in [p-1]^4} c_{\boldsymbol{k}}^{(1)} \boldsymbol{x_k}$ | 1 | 0/1 | $\frac{1}{2}/\frac{1}{2}$ | $\frac{3}{4}/\frac{1}{4}$ |
| $\mathbf{Y_2}(\boldsymbol{x}) = \sum_{\boldsymbol{k} \in [p-1]^4} c_{\boldsymbol{k}}^{(2)} \boldsymbol{x_k}\boldsymbol{x_{k+e_4}}$ | 2 | 0/2 | $\frac{1}{2}/\frac{2}{2}$ | $\frac{3}{4}/\frac{2}{4}$ |
| $\mathbf{Y_3}(\boldsymbol{x}) = \sum_{\boldsymbol{k} \in [p-1]^4} c_{\boldsymbol{k}}^{(3)} \boldsymbol{x_{k+e_3}}\boldsymbol{x_{k+e_4}}$ | 2 | 0/2 | $\frac{1}{2}/\frac{2}{2}$ | $\frac{4}{4}/\frac{2}{4}$ |
| $\mathbf{Y_4}(\boldsymbol{x}) = \sum_{\boldsymbol{k} \in [p-1]^4} c_{\boldsymbol{k}}^{(4)} \boldsymbol{x_{k+e_3+e_4}}\boldsymbol{x_{k+e_4}}\boldsymbol{x_k}$ | 3 | 0/3 | $\frac{1}{2}/\frac{3}{2}$ | $\frac{4}{4}/\frac{3}{4}$ |
| $\mathbf{Y_5^*}(\boldsymbol{x}) = \sum_{\boldsymbol{k} \in [p-1]^4} c_{\boldsymbol{k}}^{(5^*)} \boldsymbol{x_k}\boldsymbol{x_{k+e_4}}\boldsymbol{x_{k+2e_4}}(\boldsymbol{x_k^2} - \boldsymbol{x_{k+e_4}^2})$ | 5 | 0/5 | $\frac{1}{2}/\frac{5}{2}$ | $\frac{3}{4}/\frac{5}{4}$ |
| $\mathbf{Y_5}(\boldsymbol{x}) = \sum_{\boldsymbol{k} \in [p-1]^4} c_{\boldsymbol{k}}^{(5)} \boldsymbol{x_k}\boldsymbol{x_{k+e_1}}$ | 2 | 0/2 | $\frac{2}{2}/\frac{2}{2}$ | $\frac{6}{4}/\frac{2}{4}$ |
| $\mathbf{Y_6}(\boldsymbol{x}) = \sum_{\boldsymbol{k} \in [p-1]^4} c_{\boldsymbol{k}}^{(6)} \boldsymbol{x_k}\boldsymbol{x_{k+e_3}}(3\boldsymbol{x_{k-e_3}^2} - \boldsymbol{x_k^2})$ | 4 | 0/4 | $\frac{1}{2}/\frac{4}{2}$ | $\frac{5}{4}/\frac{4}{4}$ |
| $\mathbf{Y_7}(\boldsymbol{x}) = \sum_{\boldsymbol{k} \in [p-1]^4} c_{\boldsymbol{k}}^{(7)} \boldsymbol{x_{k-e_3+e_4}}\boldsymbol{x_{k+e_2}}(3\boldsymbol{x_k^2} - \boldsymbol{x_{k-e_3+e_4}^2})$ | 4 | 0/4 | $\frac{1}{2}/\frac{4}{2}$ | $\frac{6}{4}/\frac{4}{4}$ |

Each (eigen)function $Y_i$ is a linear combination of basis eigenfunction of the same type, in the sense they have the same eigenvalue and the same spatial/frequency/learning indices. The coefficients $c_{\boldsymbol{k}}^{(i)}$ are first sampled from standard Gaussians iid and then multiplied by an $i$-dependent constant so that $Y_i$ has unit norm[5]. In the experiments, $p$ is chosen to be $4$ and the target is defined to be sum of them

$$Y = \sum Y_i \tag{39}$$

Since they are orthogonal to each other, $\|Y\|_2^2/8 = \|Y_i\|_2^2 = 1$ where the $L^2$-norm is taken over the uniform distribution on $(\overline{\mathbb{S}}_{p-1})^{p^3}$. In the experiment when we compare Flatten and GAP ( Fig. 4), we set all $c_{\boldsymbol{k}}^{(i)}$ to be the same (note that we also remove $\mathbf{Y_5}$ from the target function), so that the functions are learnable by convolutional networks with a GAP readout layer.

We compare three architectures. Fig. 6 Column (a) MLP$^{\otimes 4}$, a (four-layer) MLP, the most coarse architecture used in the paper. Fig. 6 Column (b), CNN$(p^2)^{\otimes 2}$, a "D"-CNN that contains two convolutional layers with filter size/stride equal to $p^2$. Fig. 6 Column (c), CNN$(p)^{\otimes 4}$, a "HR"-CNN, the finest architecture used in the experiments, that contains four convolutional layers with filter size/stride equal to $p$. In all experiments except the one in Fig. 4, we use Flatten as the readout layer for the convolutional networks and add a *Act-Dense* layer after Flatten to improve the expressivity of the function class. However, in the Flatten vs GAP experiments, Fig. 4, we have only one dense layer after GAP/Flatten.

We show how to compute the frequency index, the spatial index and the learning index through three examples. We focus on $\mathbf{Y_2}$ / $\mathbf{Y_3}$ / $\mathbf{Y_5^*}$ which have degree 2 / 2 / 5, resp. The indices of other eigenfunctions can be computed using the same approach. We use **Dashed Lines** to represent either an edge connecting an input node to a node in the first-hidden layer or an edge associated to a dense layer. In either case, the corresponding output node of the edge has degree $O(1)$ and thus the weights (of the DAGs) of such edges are always 0. Only **Solid Lines** are relevant in computing the *spatial index*. Since each $Y_i$ is a linear combination of basis eigenfunctions of the same type, we only need to compute the indices of one component. We use the $\boldsymbol{k} = 0$ component, which corresponds to the

---

[5]In our experiments, they are normalized over the test set.

colored path in each DAG. Recall that $p = 4$, $d = p^4 = 256$ and $m_t = 32 \times 10240 \sim d^{2.28}$ (training set size), i.e. the budget index is roughly $r = 2.28$.

(a.) **MLP** Column (a) Fig. 6. The NTK and NNGP kernels are inner product kernels and the associated DAGs are linked lists. The corresponding DAG has only one input node whose dimension is equal to $d = p^4$. The spatial index is always 0 since the degree of each hidden node is 1 (since $1 = d^0$) and the frequency index is equal to the degree of the eigenfunctions. Thus $\mathscr{L}(\mathbf{Y_2}) = \mathscr{L}(\mathbf{Y_3}) = 2$ and $\mathscr{L}(\mathbf{Y_5^*}) = 5$. Changing the number of layers won't change the learning indices. In sum, learning $\mathbf{Y_2}/\mathbf{Y_3}/\mathbf{Y_5^*}$ using infinite-width MLP requires $d^{2^+}/d^{2^+}/d^{5^+}$ many samples /SGD steps. Clearly, $\mathbf{Y_5^*}$ is completely unlearnable as $r = 2.28 << 5$. In the MSE plot $\mathbf{Y_5^*}$ (5-th row in Fig. 6), the **Red Lines** does not make any progress

(b.) **CNN**$(p^2)^{\otimes 2}$ Column (b) Fig. 6. The input image is partitioned into $p^2$ patches and each patch has dimension $p^2$. The second layer of the DAG has $p^2$ many nodes, each node represents one pixel (with many channels) in the first hidden layer of a finite-width ConvNet. After one more convolutional layer with filter size/stride $p^2$, the number of node (pixel) is reduced to one. The remaining part of the DAG is essentially a linked list (**Dashed Line**) with length equal to 1, which corresponds to the *Act-Dense* layer. The frequency index $\mathscr{F}(\mathbf{Y_2}) = 2\frac{1}{2} = 1$. This is because the degree of $\mathbf{Y_2}$ is 2 and the input dimension of a node is $p^2 = d^{1/2}$. The spatial index is equal to $1/2$, since the minimum tree containing $\boldsymbol{x_k x_{k+e_4}}$ has only one non-zero edge (**Solid Lines**) whose weight is equal to 1/2 (since the degree of the output node is $p^2 = d^{1/2}$); see the **colored paths** in Fig. 6 Column (b). Therefore the learning index of $\mathscr{L}(\mathbf{Y_2}) = 1 + 1/2 = 3/2$. Similarly $\mathscr{L}(\mathbf{Y_3}) = 1 + 1/2 = 3/2$, as the term $\boldsymbol{x_{k+e_3} x_{k+e_4}}$ are lying in the same patch of size $p^2$ for all $\boldsymbol{k}$, and $\mathscr{L}(\mathbf{Y_5^*}) = 5/2 + 1/2 = 3$. In sum, learning $\mathbf{Y_2}/\mathbf{Y_3}/\mathbf{Y_5^*}$ using infinite-width CNN$(p^2)^{\otimes 2}$ requires $d^{1.5^+}/d^{1.5^+}/d^{3^+}$ many samples /SGD steps. While neither infinite-width CNN$(p^2)^{\otimes 2}$ nor MLP$^{\otimes 4}$ distinguishes $\mathbf{Y_2}$ from $\mathbf{Y_3}$, CNN$(p^2)^{\otimes 2}$ does improve the learning efficiency: $d^{2^+} \to d^{1.5^+}$. Note that $\mathbf{Y_5^*}$ is still unlearnable as $r = 2.28 < 3 = \mathscr{L}(\mathbf{Y_5^*})$. In the MSE plot $\mathbf{Y_5^*}$ (5-th row in Fig. 6), the **Orange Line** does not make any progress.

(c.) **CNN**$(p)^{\otimes 4}$ Column (c) Fig. 6. The input image is partitioned into $p^3$ patches and each patch has dimension $p$. The second/third/fourth/output layer of the DAG has $p^3/p^2/p/1$ many nodes. The frequency indices are: $\mathscr{F}(\mathbf{Y_2}) = \mathscr{F}(\mathbf{Y_3}) = 2\frac{1}{4} = 1/2$ and $\mathscr{F}(\mathbf{Y_5^*}) = 5\frac{1}{4} = 5/4$. This is because the size of input nodes is reduced to $p = d^{1/4}$. Unlike the above cases, the spatial indices become different. The two interacting terms in $\boldsymbol{x_k x_{k+e_4}}$ and the three interacting terms in $\boldsymbol{x_{k+e_4} x_{k+2e_4}}(\boldsymbol{x_k^2} - \boldsymbol{x_{k+e_4}^2})$ are in the same input node while the two interacting terms in $\boldsymbol{x_{k+e_3} x_{k+e_4}}$ and are in two different input nodes. As a consequence, the minimum spanning tree (MST) that contains $\boldsymbol{x_k}$ and $\boldsymbol{x_{k+e_4}}$ and the one contains $\boldsymbol{x_k}$, $\boldsymbol{x_{k+e_4}}$ and $\boldsymbol{x_{k+2e_4}}$ are the same. They have **3 solid lines**. However, the MST containing $\boldsymbol{x_{k+e_3}}$ and $\boldsymbol{x_{k+e_4}}$ has **4 solid lines**. Therefore $\mathcal{S}(\mathbf{Y_2}) = \mathcal{S}(\mathbf{Y_5^*}) = 3 \times \frac{1}{4}$ and $\mathcal{S}(\mathbf{Y_3}) = 4 \times \frac{1}{4}$. As such, $\mathscr{L}(\mathbf{Y_2}) = \frac{5}{4}$, $\mathscr{L}(\mathbf{Y_3}) = \frac{6}{4}$ and $\mathscr{L}(\mathbf{Y_5^*}) = \frac{8}{4}$. In sum, learning $\mathbf{Y_2}/\mathbf{Y_3}/\mathbf{Y_5^*}$ using infinite-width CNN$(p)^{\otimes 4}$ requires $d^{\mathbf{1.25^+}}/d^{\mathbf{1.5^+}}/d^{\mathbf{2^+}}$ many samples /SGD steps, resp. Now $\mathscr{L}(\mathbf{Y_5^*}) = 2. < 2.28$ and in the MSE plot $\mathbf{Y_5^*}$ (5-th row in Fig. 6), the **Blue Line** does make significant progress (the MSE is reduced to $\sim 0.2$.)

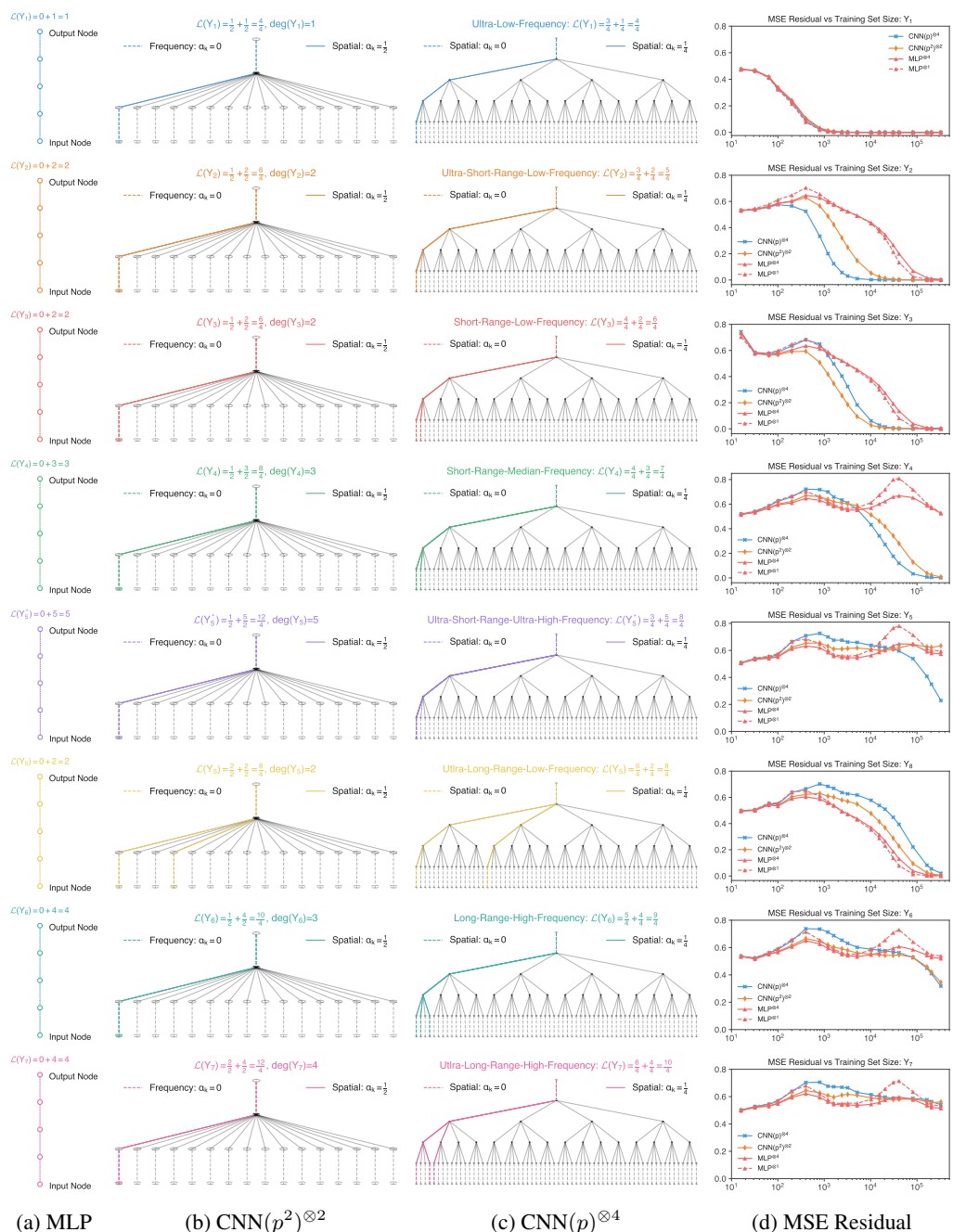

(a) MLP  (b) CNN$(p^2)^{\otimes 2}$  (c) CNN$(p)^{\otimes 4}$  (d) MSE Residual

Figure 6: **Eigenfunction vs Learning Index vs Architecture/DAG.** Rows: eigenfunctions $Y_i$ with various space-frequency combinations. Columns: DAGs associated to (1) a four-layer MLP; (b) CNN$(p^2)^{\otimes 2}$, a"D"-CNN; (c) CNN$(p)^{\otimes 4}$ a "HR"-CNN. Column (d) is the MSE, as a function of training set size (X-axis), of the residual of the corresponding eigenfunction $Y_i$ obtained by NTK-regression. The colored path in each DAG corresponds to the minimum spanning tree that contains all interacting terms in the $\boldsymbol{k} = \boldsymbol{0}$ component of $Y_i$.

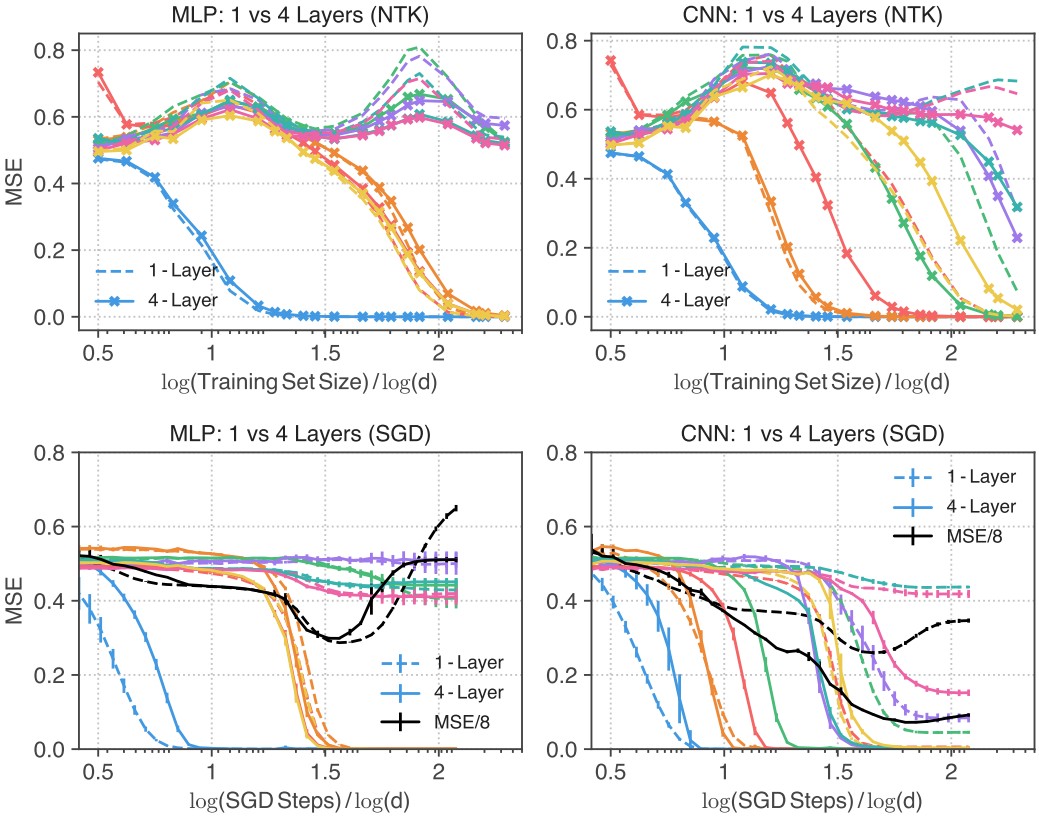

Figure 7: **MLPs do not benefits from having more layers.** We plot the learning dynamics vs training set size / SGD steps for each eigenfunction $Y_i$. Top: NTK regression and bottom: SGD + Momentum. Left: MLP; right: CNN. **Dashed lines / Solid lines** correspond to one-hidden/four-hidden layer networks. For both finite-width SGD training and infinite-width kernel regression, having more layers does not essentially improve performance of a MLP. This is in stark contrast to CNNs (right). By having more layers, the eigenstuctures of the kernels are refined.

# E   FIGURE ZOO

## E.1   MLPS: DEPTH ≠ HIERARCHY

We compare a one hidden layer MLP and a four hidden layer MLP in Fig. 7. Unlike CNNs, increasing the number of layers does not improve the performance of MLPs much for both NTK regression and SGD. This is consistent with a theoretical result from Bietti & Bach (2020), which says the NTKs of Relu MLPs are essentially the same for any depth.

## E.2   IMAGENET PLOTS

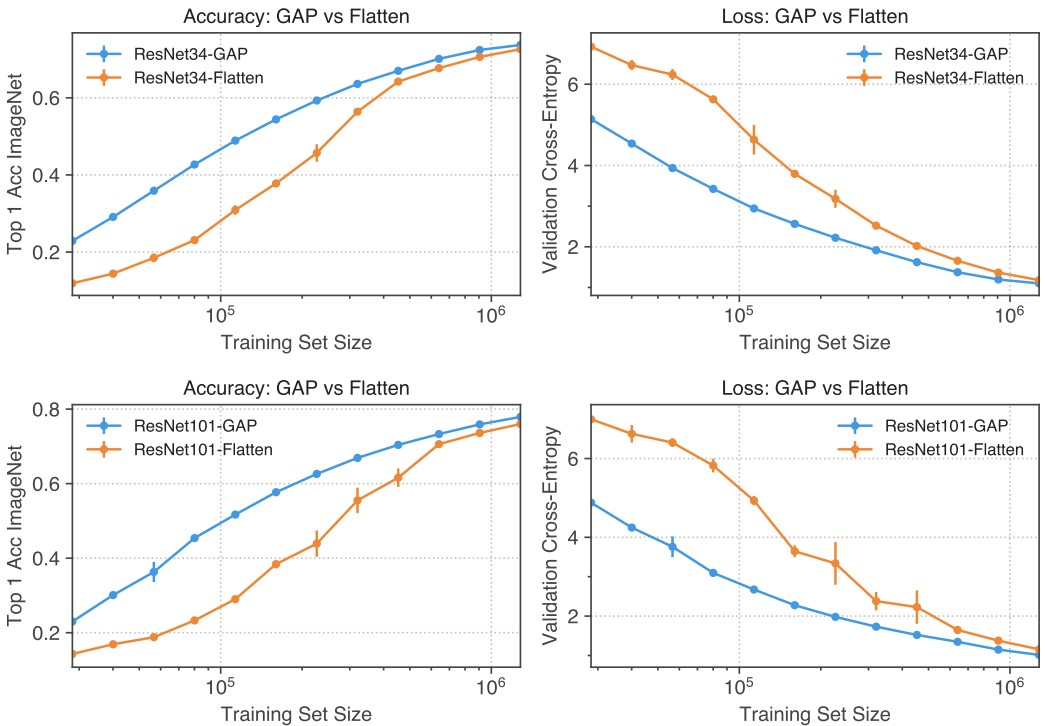

Figure 8: **ResNet-GAP vs ResNet-Flatten.** As the training set size increases the performance (accuracy and loss) gap between the two shrinks substantially. Top/bottom ResNet34/ResNet101

## F ARCHITECTURE SPECIFICATIONS

In this section, we provide the details of the architectures used in the experiments.

## G ASSUMPTIONS

**Assumption-$\mathcal{G}$.** Let $\mathcal{G} = (\mathcal{G}^{(d)})_d$. There are absolute constants $c, C > 0$ such that

(a.) For each *non-input* node $u \in \mathcal{N}^{(d)}$, there is $\alpha_u \in \Lambda_{\mathcal{G}}$ such that

$$cd^{\alpha_u} \leq \deg(u) \leq Cd^{\alpha_u} . \tag{40}$$

For each edge $uv \in \mathcal{E}^{(d)}$, its weight is defined to be $\omega_{uv} \equiv \alpha_u$.

(b.) For each *input* node $v$, there are $d_v \in \mathbb{N}$ and $0 < \alpha_v \in \Lambda_{\mathcal{G}}$ such that

$$cd^{\alpha_v} \leq d_v \leq Cd^{\alpha_v} \quad \text{and} \quad \sum_{v \in \mathcal{N}_0^{(d)}} \alpha_v = 1. \tag{41}$$

(c.) Let $\mathcal{N}_1^{(d)} \equiv \{u : \exists v \in \mathcal{N}_0^{(d)} \text{ s.t. } uv \in \mathcal{E}^{(d)}\}$ be the collection of nodes in the *first* hidden layer. We assume that for every $u \in \mathcal{N}_1^{(d)}$, $\alpha_u = 0$ and all children of $u$ are input nodes.

(d.) For every $v \in \mathcal{N}^{(d)}$, $|\{u : uv \in \mathcal{E}^{(d)}\}| \leq C$. Moreover, the number of *layers* is uniformly bounded, namely, for any node $u$, any path from $u$ to $o_{\mathcal{G}}$ contains at most $C$ edges.

The first two assumptions help to create spectral gaps between eigenspaces. When $d$ is not large, "finite-width" effect is no longer negligible and we expect that the spectra decay more smoothly. Assumption (c.) says there is no "skip" connections from the input layer to other layers except to the first hidden layer.

```python
from neural_tangents import stax

def MLP(width=2048, depth=1, W_std=0.5, activation=stax.Rbf()):
    layers = []
    for _ in range(depth):
        layers += [stax.Dense(width, W_std=W_std), activation]
    layers += [stax.Dense(1)]
    return stax.serial(*layers)

def CNN(width=512, ksize=4, depth=1, W_std=0.5, activation=stax.Rbf(), readout=stax.Flatten(),
        act_after_readout=True):
    layers = []
    conv_op = stax.Conv(width, (ksize, 1), strides=(ksize, 1), W_std=W_std,  padding='VALID')
    for _ in range(depth):
        layers += [conv_op, activation]
    layers += [readout]
    if act_after_readout:
        layers += [stax.Dense(width * 4, W_std=W_std), activation]
    layers += [stax.Dense(1)]
    return stax.serial(*layers)

p = 4 # input shape: (-1, p**4, 1, 1)
S_MLP = MLP(depth=1) # Shallow MLP
D_MLP = MLP(depth=4) # Deep MLP

S_CNN = CNN(ksize=p, depth=1, act_after_readout=False) # Shallow CNN
# One layer CNN with an additional activation-dense layer
S_CNN_plus_Act = CNN(ksize=p, depth=1, act_after_readout=True)
D_CNN = CNN(ksize=p**2, depth=2) # Deep CNN
HR_CNN = CNN(ksize=p, depth=4) # High-resolution CNN
# High-resolution CNN with global average pooling readout
HR_CNN_GAP = CNN(ksize=p, depth=4, readout=stax.GlobalAvgPool(), act_after_readout=False)
# High-resolution CNN with flattening as readout
HR_CNN_Flatten = CNN(ksize=p, depth=4, act_after_readout=False)
```

Listing 1: **Definitions of MLPs, S-CNN, D-CNN and HR-CNN used in the experiments**. The architectures used in the experiments of Fig. 3 are defined as follows. $\text{Dense}^{\otimes 4}$ = D_MLP, $\text{Conv}(p^2)^{\otimes 2}$ = D_CNN, $\text{Conv}(p)^{\otimes 4}$ = HR_CNN. The one layer MLP and one layer CNN used in Fig. 7 are S_MLP and S_CNN_plus_Act, resp.

## H  PROOF OF THE EIGENSPACE RESTRUCTURING THEOREM.

**Lemma 1.** *Let* $r \in \mathcal{N}_0^{(d)}$. *Then*

$$\mathcal{K}_{\mathcal{G}}(0), \quad \Theta_{\mathcal{G}}(0) = 0 \tag{42}$$

*and for* $r \neq 0$, *for* $d$ *large enough,*

$$\mathcal{K}_{\mathcal{G}}^{(r)}(0), \quad \Theta_{\mathcal{G}}^{(r)}(0) \sim d^{-\mathcal{S}(r)} \quad and \quad \|\mathcal{K}_{\mathcal{G}}^{(r)}\|_{\infty}, \ \|\Theta_{\mathcal{G}}^{(r)}\|_{\infty} \lesssim d^{-\mathcal{S}(r)} \tag{43}$$

*Proof.* Recall that the recursion formulas for $\mathcal{K}$ and $\Theta$ are

$$\mathcal{K}_u(t) = \phi_u^* \left( \fint_{v:uv\in\mathcal{E}} \mathcal{K}_v(t) \right) \tag{44}$$

$$\Theta_u(t) = \dot{\phi}_u^* \left( \fint_{v:uv\in\mathcal{E}} \mathcal{K}_v(t) \right) \fint_{v:uv\in\mathcal{E}} (\mathcal{K}_v(t) + \Theta_v(t)) \tag{45}$$

For convenience, define $\overline{\mathcal{K}}, \overline{\Theta}, \dot{\mathcal{K}}$ as follows

$$\overline{\mathcal{K}}_u(t) = \fint_{v:uv\in\mathcal{E}} \mathcal{K}_v(t) \tag{46}$$

$$\overline{\Theta}_u(t) = \fint_{v:uv\in\mathcal{E}} \Theta_v(t) \tag{47}$$

$$\dot{\mathcal{K}}_u(t) = \dot{\phi}_u^* \circ \overline{\mathcal{K}}_u(t) . \tag{48}$$

Note that

$$\mathcal{K}_u(t) = \phi_u^* \circ \overline{\mathcal{K}}_u(t) \quad \text{and} \quad \Theta(t) = \dot{\mathcal{K}}_u(t)(\overline{\mathcal{K}}_u(t) + \overline{\Theta}_u(t)) .$$

The equalities $\mathcal{K}_u(\mathbf{0}) = \Theta_u(\mathbf{0}) = 0$ follow easily from recursion and the fact $\phi_u^*(0) = 0$ for all $u \in \mathcal{N}^{(d)}$.

We induct on the tuple $(h, |\boldsymbol{r}|)$, where $h \geq 0$ is the number of *hidden* layers in $\mathcal{G}^{(d)}$ and $|\boldsymbol{r}|$ is the total degree of $\boldsymbol{r}$. We begin with the proof of the NNGP kernel $\mathcal{K}$.

**Base Case I:** $|\boldsymbol{r}| = 1$ and $h \geq 0$ is any integer. Let $u \in \mathcal{N}_0$ be such that $\boldsymbol{r} = \boldsymbol{e}_u$, where $\{\boldsymbol{e}_u\}_{u\in\mathcal{N}_0^{(d)}}$ is the standard basis. Then

$$\partial_{\boldsymbol{t}_u} \mathcal{K}_{\mathcal{G}}(\boldsymbol{t}) = \sum_{\text{path}\in\mathcal{P}(u\to o_{\mathcal{G}})} \prod_{v\in\text{path}} \deg(v)^{-1} \dot{\phi}_v^* \circ \overline{\mathcal{K}}_v(\boldsymbol{t}) \sim \sum_{\text{path}\in\mathcal{P}(u\to o_{\mathcal{G}})} \prod_{v\in\text{path}} d^{-\alpha_v} \dot{\phi}_v^* \circ \overline{\mathcal{K}}_v(\boldsymbol{t}) \tag{49}$$

Here $\mathcal{P}(u \to u')$ represents the set of paths from $u$ to $u'$. By **Assumption-$\mathcal{G}$** and **Assumption-$\phi$**, $|\mathcal{P}(u \to o_{\mathcal{G}})|$ is uniformly bounded and $\dot{\phi}_v^*(0) > 0$ for all hidden nodes $v$. Therefore

$$\partial_{\boldsymbol{t}_u} \mathcal{K}_{\mathcal{G}}(\mathbf{0}) \sim \sum_{\text{path}\in\mathcal{P}(u\to o_{\mathcal{G}})} d^{-\sum_{v\in\text{path}} \alpha_v} \dot{\phi}_v^* \circ \overline{\mathcal{K}}_v(\mathbf{0})$$

$$= \sum_{\text{path}\in\mathcal{P}(u\to o_{\mathcal{G}})} d^{-\sum_{v\in\text{path}} \alpha_v} \dot{\phi}_v^*(0)$$

$$\sim \max_{\text{path}\in\mathcal{P}(u\to o_{\mathcal{G}})} d^{-\sum_{v\in\text{path}} \alpha_v}$$

$$= d^{-\mathcal{S}(\boldsymbol{e}_u)} = d^{-\mathcal{S}(\boldsymbol{r})}$$

In the above, we have used $\overline{\mathcal{K}}_v(\mathbf{0}) = 0$ (which is due to $\phi^*(0) = 0$.) The second estimate $\|\partial_{t_u} \mathcal{K}_{\mathcal{G}}\|_\infty \lesssim d^{-\mathcal{S}(\boldsymbol{r})}$ follows from $|\dot{\phi}_v^* \circ \overline{\mathcal{K}}_v(\boldsymbol{t})| \lesssim 1$.

**Base Case II:** $h = 0$ and $|\boldsymbol{r}| \geq 1$ is any number. Note that $\mathcal{G}^{(d)}$ has no hidden layer and all input nodes are linked to the output node $o_{\mathcal{G}}$. The case when the activation $\phi_{o_{\mathcal{G}}}$ is the identity function is obvious and we assume $\phi_{o_{\mathcal{G}}}$ is admissible.

$$\partial_{\boldsymbol{t}}^{\boldsymbol{r}} \mathcal{K}_{\mathcal{G}}(\boldsymbol{t}) = \deg(o_{\mathcal{G}})^{-|\boldsymbol{r}|} \phi_{o_{\mathcal{G}}}^{*}{}^{(|\boldsymbol{r}|)} \left( \fint_{u\in\mathcal{N}_0^{(d)}} \boldsymbol{t}_u \right) .$$

This implies Eq. (43) since $\mathcal{S}(\boldsymbol{r}) = 0$, $\deg(o_{\mathcal{G}}) \lesssim 1$ by **Assumption-$\mathcal{G}$** and $\phi_{o_{\mathcal{G}}}^{*}{}^{(|\boldsymbol{r}|)}(0) > 0$ by **Assumption-$\phi$**.

**Induction:** $|\boldsymbol{r}| \geq 2$, $h \geq 1$ and $\boldsymbol{r} \in \mathcal{A}(\mathcal{G}^{(d)})$. We only prove the first estimate in Eq. (43) since the other one can be proved similarly. WLOG, we assume $\phi_{o_{\mathcal{G}}}$ is not the identity function and hence is

semi-admissible. Let $u \in \mathcal{N}_0^{(d)}$ be such that $\boldsymbol{r}_u \geq 1$ and denote $\bar{\boldsymbol{r}} = \boldsymbol{r} - \boldsymbol{e}_u$. Then

$$\partial_{\boldsymbol{t}}^{\boldsymbol{r}} \mathcal{K}_{\mathcal{G}}(\boldsymbol{t})\Big|_{\boldsymbol{t}=\boldsymbol{0}} = \partial_{\boldsymbol{t}}^{\bar{\boldsymbol{r}}}(\partial_{\boldsymbol{t}}^{\boldsymbol{e}_u}\mathcal{K}_{\mathcal{G}}(\boldsymbol{t}))\Big|_{\boldsymbol{t}=\boldsymbol{0}} = \sum_{\substack{o_{\mathcal{G}}v\in\mathcal{E} \\ \partial_{\boldsymbol{t}}^{\boldsymbol{e}_u}\mathcal{K}_v \neq 0}} \deg(o_{\mathcal{G}})^{-1}\partial_{\boldsymbol{t}}^{\bar{\boldsymbol{r}}}\left(\dot{\mathcal{K}}_{o_{\mathcal{G}}}(\boldsymbol{t})\partial_{\boldsymbol{t}}^{\boldsymbol{e}_u}\mathcal{K}_v(\boldsymbol{t})\right)\Big|_{\boldsymbol{t}=\boldsymbol{0}} \quad (50)$$

$$= \sum_{\substack{o_{\mathcal{G}}v\in\mathcal{E} \\ \partial_{\boldsymbol{t}}^{\boldsymbol{e}_u}\mathcal{K}_v \neq 0}} \sum_{\bar{\boldsymbol{r}}_1+\bar{\boldsymbol{r}}_2=\bar{\boldsymbol{r}}} \deg(o_{\mathcal{G}})^{-1}\left(\partial_{\boldsymbol{t}}^{\bar{\boldsymbol{r}}_1}\dot{\mathcal{K}}_{o_{\mathcal{G}}}(\boldsymbol{t})\,\partial_{\boldsymbol{t}}^{\bar{\boldsymbol{r}}_2+\boldsymbol{e}_u}\mathcal{K}_v(\boldsymbol{t})\right)\Big|_{\boldsymbol{t}=\boldsymbol{0}} \quad (51)$$

$$\sim \sum_{\substack{o_{\mathcal{G}}v\in\mathcal{E} \\ \partial_{\boldsymbol{t}}^{\boldsymbol{e}_u}\mathcal{K}_v \neq 0}} \sum_{\bar{\boldsymbol{r}}_1+\bar{\boldsymbol{r}}_2=\bar{\boldsymbol{r}}} \deg(o_{\mathcal{G}})^{-1}d^{-\mathcal{S}(\bar{\boldsymbol{r}}_1)}d^{-\mathcal{S}(n(\bar{\boldsymbol{r}}_2+\boldsymbol{e}_u;v))} \quad (52)$$

$$\sim \sum_{\substack{o_{\mathcal{G}}v\in\mathcal{E} \\ \partial_{\boldsymbol{t}}^{\boldsymbol{e}_u}\mathcal{K}_v \neq 0}} \sum_{\bar{\boldsymbol{r}}_1+\bar{\boldsymbol{r}}_2=\bar{\boldsymbol{r}}} d^{-\mathcal{S}(\bar{\boldsymbol{r}}_1)}d^{-\left(\alpha_{o_{\mathcal{G}}}+\mathcal{S}(n(\bar{\boldsymbol{r}}_2+\boldsymbol{e}_u;v))\right)} \quad (53)$$

$$\sim \sup_{\substack{o_{\mathcal{G}}v\in\mathcal{E} \\ \partial_{\boldsymbol{t}}^{\boldsymbol{e}_u}\mathcal{K}_v \neq 0}} \sup_{\bar{\boldsymbol{r}}_1+\bar{\boldsymbol{r}}_2=\bar{\boldsymbol{r}}} d^{-\left(\mathcal{S}(\bar{\boldsymbol{r}}_1)+\alpha_{o_{\mathcal{G}}}+\mathcal{S}(n(\bar{\boldsymbol{r}}_2+\boldsymbol{e}_u;v))\right)} \quad (54)$$

We have applied induction twice in Eq. (52): one to obtain the estimate $\partial_{\boldsymbol{t}}^{\bar{\boldsymbol{r}}_1}\dot{\mathcal{K}}_{o_{\mathcal{G}}}^*(\boldsymbol{0}) \sim d^{-\mathcal{S}(\bar{\boldsymbol{r}}_1)}$ (with $|\bar{\boldsymbol{r}}_1| < |\boldsymbol{r}|$ and $\dot{\phi}_{o_{\mathcal{G}}}^*$ semi-admissible) and one to $\partial_{\boldsymbol{t}}^{\bar{\boldsymbol{r}}_2+\boldsymbol{e}_u}\mathcal{K}_v(\boldsymbol{t}) \sim d^{-\mathcal{S}(n(\bar{\boldsymbol{r}}_2+\boldsymbol{e}_u;v))}$, in which the subgraph with $v$ as the output node has depth at most $(h-1)$. The last line follows from that both the cardinality of the tuple $(\bar{\boldsymbol{r}}_1, \bar{\boldsymbol{r}}_2)$ with $\bar{\boldsymbol{r}}_1, \bar{\boldsymbol{r}}_2 \geq \boldsymbol{0}$ and $\bar{\boldsymbol{r}}_1 + \bar{\boldsymbol{r}}_2 = \bar{\boldsymbol{r}}$ and the cardinality of $v \in \mathcal{N}_0^{(d)}$ with $o_{\mathcal{G}}v \in \mathcal{E}$ and $\partial_{\boldsymbol{t}}^{\boldsymbol{e}_u}\mathcal{K}_v \neq 0$ are finite and independent of $d$. From the definition of MST, it is clear that for all $(\bar{\boldsymbol{r}}_1, \bar{\boldsymbol{r}}_2)$

$$\mathcal{S}(\bar{\boldsymbol{r}}_1) + \alpha_{o_{\mathcal{G}}} + \mathcal{S}(n(\bar{\boldsymbol{r}}_2+\boldsymbol{e}_u;v)) \geq \mathcal{S}(\bar{\boldsymbol{r}}_1) + \mathcal{S}(\bar{\boldsymbol{r}}_2+\boldsymbol{e}_u) \geq \mathcal{S}(\boldsymbol{r}) \quad (55)$$

It remains to show that there exists at least one pair $(\bar{\boldsymbol{r}}_1, \bar{\boldsymbol{r}}_2)$ such that the above can be an equality. Let $\mathcal{T} \subseteq \mathcal{G}^{(d)}$ be a MST containing all nodes in $n(\boldsymbol{r})$. If $o_{\mathcal{G}}$ has only one child $v$ in $\mathcal{T}$, then we choose $\bar{\boldsymbol{r}}_1 = \boldsymbol{0}$ and notice that

$$\mathcal{S}(\bar{\boldsymbol{r}}_1) + \alpha_{o_{\mathcal{G}}} + \mathcal{S}(n(\bar{\boldsymbol{r}}_2+\boldsymbol{e}_u;v)) = 0 + \mathcal{S}(\boldsymbol{r}_2+\boldsymbol{e}_u) = \mathcal{S}(\boldsymbol{r}) \quad (56)$$

since $\mathcal{S}(\bar{\boldsymbol{r}}_1) = 0$ and $\bar{\boldsymbol{r}}_2 + \boldsymbol{e}_u = \boldsymbol{r}$. Else, $\mathcal{T}$ contains at least two children and therefore at least two disjoint branches split from $o_{\mathcal{G}}$. Let $\mathcal{T}_u \subseteq \mathcal{T}$ be the branch that contains $u$ and choose $\bar{\boldsymbol{r}}_2 \leq \bar{\boldsymbol{r}}$ be such that all the nodes of $(\bar{\boldsymbol{r}}_2 + \boldsymbol{e}_u)$ are contained in $\mathcal{T}_u$ and all the nodes of $\bar{\boldsymbol{r}}_1 \equiv \boldsymbol{r} - (\bar{\boldsymbol{r}}_2 + \boldsymbol{e}_u)$ are contained in $\mathcal{T}\backslash\mathcal{T}_u$. Clearly

$$\mathcal{S}(\boldsymbol{r}) = \mathcal{S}(\bar{\boldsymbol{r}}_1) + \mathcal{S}(\bar{\boldsymbol{r}}_2+\boldsymbol{e}_u) = \mathcal{S}(\bar{\boldsymbol{r}}_1) + \mathcal{S}(n(\bar{\boldsymbol{r}}_2+\boldsymbol{e}_u;v)) + \alpha_{o_{\mathcal{G}}}, \quad (57)$$

where $v$ is the unique child of $o_{\mathcal{G}}$ in $\mathcal{T}_u$.

This completes the proof of the NNGP kernel $\mathcal{K}$. As the proof of the NTK part is quite similar, we will be brief and focus only on the induction step.

**Induction Step of $\Theta$:** $|\boldsymbol{r}| \geq 2$, $h \geq 1$ and $\boldsymbol{r} \in \mathcal{A}(\mathcal{G}^{(d)})$. Recall that the formula of $\Theta$ is

$$\Theta_u(\boldsymbol{t}) = \dot{\phi}_u^*\left(\fint_{v:uv\in\mathcal{E}} \mathcal{K}_v(\boldsymbol{t})\right)\fint_{v:uv\in\mathcal{E}} (\mathcal{K}_v(\boldsymbol{t}) + \Theta_v(\boldsymbol{t})) \quad (58)$$

For $\boldsymbol{r} \in \mathcal{N}_0^{(d)}$,

$$\partial_{\boldsymbol{t}}^{\boldsymbol{r}}\Theta_{o_{\mathcal{G}}}(\boldsymbol{t}) = \sum_{\bar{\boldsymbol{r}}_1+\bar{\boldsymbol{r}}_2=\boldsymbol{r}} \partial_{\boldsymbol{t}}^{\bar{\boldsymbol{r}}_1}\dot{\phi}_{o_{\mathcal{G}}}^*\left(\fint_{v:o_{\mathcal{G}}v\in\mathcal{E}} \mathcal{K}_v(\boldsymbol{t})\right)\partial_{\boldsymbol{t}}^{\bar{\boldsymbol{r}}_2}\fint_{v:o_{\mathcal{G}}v\in\mathcal{E}} (\mathcal{K}_v(\boldsymbol{t}) + \Theta_v(\boldsymbol{t})) \quad (59)$$

Note that $\dot{\phi}_{o_{\mathcal{G}}}^*$ is semi-admissible. We apply the result of $\mathcal{K}$ to conclude that

$$\partial_{\boldsymbol{t}}^{\bar{\boldsymbol{r}}_1}\dot{\phi}_{o_{\mathcal{G}}}^*\left(\fint_{v:o_{\mathcal{G}}v\in\mathcal{E}} \mathcal{K}_v(\boldsymbol{t})\right)\Bigg|_{\boldsymbol{t}=\boldsymbol{0}} \sim d^{-\mathcal{S}(\boldsymbol{r}_1)} \quad (60)$$

$$\left\|\partial_{\boldsymbol{t}}^{\bar{\boldsymbol{r}}_1}\dot{\phi}_{o_{\mathcal{G}}}^*\left(\fint_{v:o_{\mathcal{G}}v\in\mathcal{E}} \mathcal{K}_v(\boldsymbol{t})\right)\right\|_\infty \lesssim d^{-\mathcal{S}(\bar{\boldsymbol{r}}_1)} \quad (61)$$

and the inductive step to conclude that

$$\partial_t^{\bar{r}_2} \fint_{v:o_{\mathcal{G}}v\in\mathcal{E}} (\mathcal{K}_v(t) + \Theta_v(t))\Big|_{t=0} \sim \sum_{\substack{v:o_{\mathcal{G}}v\in\mathcal{E} \\ \partial_t^{\bar{r}_2}(\mathcal{K}_v+\Theta_v)\not\equiv 0}} d^{-\mathscr{S}(\bar{r}_2)} \quad \text{if} \quad \bar{r}_2 \neq \mathbf{0} \quad \text{else} \quad 0 \tag{62}$$

$$\partial_t^{\bar{r}_2} \fint_{v:o_{\mathcal{G}}v\in\mathcal{E}} (\mathcal{K}_v(t) + \Theta_v(t)) \lesssim \sum_{\substack{v:o_{\mathcal{G}}v\in\mathcal{E} \\ \partial_t^{\bar{r}_2}(\mathcal{K}_v+\Theta_v)\not\equiv 0}} d^{-\mathscr{S}(\bar{r}_2)}. \tag{63}$$

Note that

$$|\{v : o_{\mathcal{G}}v \in \mathcal{E}^{(d)} \quad \text{and} \quad \partial_t^{\bar{r}_2}(\mathcal{K}_v + \Theta_v) \not\equiv 0\}| \lesssim 1.$$

Thus

$$\|\partial_t^r \Theta_{o_{\mathcal{G}}}(t)\|_\infty \lesssim \sum_{\bar{r}_1+\bar{r}_2=r} d^{-\mathscr{S}(\bar{r}_1)-\mathscr{S}(\bar{r}_2)} \lesssim d^{-\mathscr{S}(r)}. \tag{64}$$

To control the lower bound, let $\mathcal{T}$ be a MST containing $n(r)$. If $\deg(o_{\mathcal{G}};\mathcal{T}) = 1$, then we can choose $\bar{r}_1 = 0$ and $\bar{r}_2 = r \neq \mathbf{0}$. Notice that there is at least one child node $v$ of $o_{\mathcal{G}}$ with $\partial_t^{\bar{r}_2}(\mathcal{K}_v + \Theta_v) \not\equiv 0$. Therefore

$$\sum_{v:\partial_t^{\bar{r}_2}(\mathcal{K}_v+\Theta_v)\not\equiv 0} d^{-\mathscr{S}(\bar{r}_2)} \gtrsim d^{-\mathscr{S}(\bar{r}_2)} = d^{-\mathscr{S}(r)} \tag{65}$$

Combining with

$$\dot{\phi}^*_{o_{\mathcal{G}}}\left(\fint_{v:o_{\mathcal{G}}v\in\mathcal{E}} \mathcal{K}_v(\mathbf{0})\right) = \dot{\phi}^*_{o_{\mathcal{G}}}(0) > 0 \tag{66}$$

we have

$$\partial_t^r \Theta_{o_{\mathcal{G}}}(\mathbf{0}) \gtrsim d^{-\mathscr{S}(r)}$$

It remains to handle the $\deg(o_{\mathcal{G}};\mathcal{T}) > 1$ case. We choose $(\bar{r}_1, \bar{r}_2)$ such that one branch of $\mathcal{T}$ is the MST that contains $n(\bar{r}_2)$ and the $o_{\mathcal{G}}$ and the remaining branch(es) is a MST that contains $n(\bar{r}_1)$ and the $o_{\mathcal{G}}$. Then

$$\partial_t^r \Theta_{o_{\mathcal{G}}}(\mathbf{0}) \gtrsim \partial_t^{\bar{r}_1} \dot{\phi}^*_{o_{\mathcal{G}}}\left(\fint_{v:o_{\mathcal{G}}v\in\mathcal{E}} \mathcal{K}_v(t)\right) \partial_t^{\bar{r}_2} \fint_{v:o_{\mathcal{G}}v\in\mathcal{E}} (\mathcal{K}_v(t) + \Theta_v(t))\Big|_{t=0}$$

$$\gtrsim d^{-\mathscr{S}(r_1)-\mathscr{S}(r_2)} = d^{-\mathscr{S}(r)}$$

$\square$

## H.1 Legendre Polynomials, Spherical Harmonics and their Tensor Products.

Our notation follows closely from (Frye & Efthimiou, 2012).

**Legendre Polynomials.** Let $d_{\text{in}} \in \mathbb{N}^*$ and $\omega_{d_{\text{in}}}$ be the measure defined on the interval $I = [-1, 1]$

$$\omega_{d_{\text{in}}}(t) = (1 - t^2)^{(d_{\text{in}}-3)/2} \tag{67}$$

The Legendre polynomials[6] $\{P_r(t) : r \in \mathbb{N}\}$ is an orthogonal basis for the Hilbert space $L^2(I, \omega_{d_{\text{in}}})$, i.e.

$$\int_I P_r(t)P_{r'}(t)\omega_{d_{\text{in}}}(t)dt = 0 \quad \text{if} \quad r \neq r' \quad \text{else} \quad N(d_{\text{in}}, r)^{-1}\left(\frac{|\mathbb{S}_{d_{\text{in}}-1}|}{|\mathbb{S}_{d_{\text{in}}-2}|}\right) \tag{68}$$

Here $P_r(t)$ is a degree $r$ polynomials with $P_r(1) = 1$ that satisfies the formula below, $N(d_{\text{in}}, r)$ is the cardinality of degree $r$ spherical harmonics in $\mathbb{R}^{d_{\text{in}}}$ and $|\mathbb{S}_{d_{\text{in}}-1}|$ is the measure of $\mathbb{S}_{d_{\text{in}}-1}$.

---

[6]More accurate, this should be called Gegenbauer Polynomials. However, we decide to stick to the terminology in (Frye & Efthimiou, 2012)

**Lemma 2** (Rodrigues Formula. Proposition 4.19 (Frye & Efthimiou, 2012)).

$$P_r(t) = c_r \omega_{d_{in}}^{-1}(t) \left(\frac{d}{dt}\right)^r (1 - t^2)^{r+(d_{in}-3)/2} , \tag{69}$$

*where*

$$c_r = \frac{(-1)^r}{2^r (r + (d_{in} - 3)/2)_r} \tag{70}$$

In the above lemma, $(x)_l$ denotes the falling factorial

$$(x)_l \equiv x(x-1)\cdots(x-l+1) \tag{71}$$
$$(x)_0 \equiv 1 \tag{72}$$

**Spherical Harmonics.** Let $d\mathbb{S}_{d_{in}-1}$ define the (un-normalized) uniform measure on the unit sphere $\mathbb{S}_{d_{in}-1}$. Then

$$|\mathbb{S}_{d_{in}-1}| \equiv \int_{\mathbb{S}_{d_{in}-1}} d\mathbb{S}_{d_{in}-1} = \frac{2\pi^{d_{in}/2}}{\Gamma(\frac{d_{in}}{2})} . \tag{73}$$

The normalized measure on this sphere is defined to be

$$d\sigma_{d_{in}} = \frac{1}{|\mathbb{S}_{d_{in}-1}|} d\mathbb{S}_{d_{in}-1} \quad \text{and} \quad \int_{\mathbb{S}_{d_{in}-1}} d\sigma_{d_{in}} = 1 . \tag{74}$$

The spherical harmonics $\{Y_{r,l}\}_{r,l}$ in $\mathbb{R}^{d_{in}}$ are homogeneous harmonic polynomials that form an orthonormal basis in $L^2(\mathbb{S}_{d_{in}-1}, \sigma_{d_{in}})$

$$\int_{\xi \in \mathbb{S}_{d_{in}-1}} Y_{r,l}(\xi) Y_{r',l'}(\xi) d\sigma_{d_{in}} = \delta_{(r,l)=(r',l')} . \tag{75}$$

Here $Y_{r,l}$ denotes the $l$-th spherical harmonic whose degree is $r$, where $r \in \mathbb{N}$, $l \in [N(d_{in}, r)]$ and

$$N(d_{in}, r) = \frac{2r + d_{in} - 2}{r} \binom{d_{in} + r - 3}{r - 1} \sim (d_{in})^r / r! \quad \text{as} \quad d_{in} \to \infty . \tag{76}$$

The Legendre polynomials and spherical harmonics are related through the addition theorem.

**Lemma 3** (Addition Theorem. Theorem 4.11 (Frye & Efthimiou, 2012)).

$$P_r(\xi^T \eta) = \frac{1}{N(d_{in}, r)} \sum_{l \in [N(d_{in}, r)]} Y_{r,l}(\xi) Y_{r,l}(\eta), \quad \xi, \eta \in \mathbb{S}_{d_{in}-1} . \tag{77}$$

**Tensor Products.** Let $p = |\mathcal{N}_0^{(d)}|$, $\boldsymbol{d} = (d_u)_{|u \in \mathcal{N}_0^{(d)}|}$, $\boldsymbol{r} \in \mathbb{N}^{|\mathcal{N}_0^{(d)}|} \simeq \mathbb{N}^p$, $I^{|\mathcal{N}_0^{(d)}|} \simeq I^p = [-1, 1]^p$ and $\boldsymbol{\omega} = \bigotimes_{u \in \mathcal{N}_0^{(d)}} \omega_{d_u}^p$ be the product measure on $I^p$. Then the (product of) Legendre polynomials

$$\boldsymbol{P_r(t)} = \prod_{u \in \mathcal{N}_0^{(d)}} P_{r_u}(t_u), \quad \boldsymbol{t} = (t_u)_{u \in \mathcal{N}_0^{(d)}} \in I^p , \tag{78}$$

which form an orthogonal basis for the Hilbert space $L^2(I^p, \boldsymbol{\omega}) = \bigotimes_{u \in \mathcal{N}_0^{(d)}} L^2(I, \omega_{d_u})$. Similarly, the tensor product of spherical harmonics

$$\boldsymbol{Y_{r,l}} = \prod_{u \in \mathcal{N}_0^{(d)}} Y_{r_u, l_u}, \quad \boldsymbol{l} = (l_u)_{u \in \mathcal{N}_0^{(d)}} \in [\boldsymbol{N(d, r)}] \equiv \prod_{u \in \mathcal{N}_0^{(d)}} [N(d_u, r)] \tag{79}$$

form an orthonormal basis for the product space

$$L^2(\boldsymbol{\mathcal{X}}, \boldsymbol{\sigma}) \equiv \bigotimes_{u \in \mathcal{N}_0^{(d)}} L^2(\mathbb{S}_{d_u-1}, \sigma_{d_u}) \tag{80}$$

Elements in the set $\{\boldsymbol{Y_{r,l}}\}_{l \in [\boldsymbol{N(d,r)}]}$ are called degree (order) $\boldsymbol{r}$ spherical harmonics in $L^2(\boldsymbol{\mathcal{X}}, \boldsymbol{\sigma})$ and also degree $r$ spherical harmonics if $|\boldsymbol{r}| = r \in \mathbb{N}$.

**Theorem 5.** *We have the following, for* $\mathcal{K} = \mathcal{K}_{\mathcal{G}^{(d)}}$ *or* $\mathcal{K} = \Theta_{\mathcal{G}^{(d)}}$

$$\mathcal{K}(t) = \sum_{r \in \mathbb{N}_0^{\mathcal{N}_0^{(d)}}} \hat{\mathcal{K}}(r) P_r(t) \quad with \quad \hat{\mathcal{K}}(r) \sim d^{-\mathcal{S}(r)} \quad if \quad r \neq 0 \quad else \quad 0. \tag{81}$$

Note that Theorem 1 follows from this theorem and the addition theorem.

*Proof of Theorem 1.* Assume $r \neq 0$. Indeed, setting

$$\boldsymbol{\xi} = (\boldsymbol{\xi}_u)_{u \in \mathcal{N}_0^{(d)}} \in \mathcal{X}, \quad \boldsymbol{\eta} = (\boldsymbol{\eta}_u)_{u \in \mathcal{N}_0^{(d)}} \in \mathcal{X} \text{ and } \quad \boldsymbol{t} = (\boldsymbol{t}_u)_{u \in \mathcal{N}_0^{(d)}} = (\boldsymbol{\xi}_u^T \boldsymbol{\eta}_u / \boldsymbol{d}_u)_{u \in \mathcal{N}_0^{(d)}},$$

we have

$$\boldsymbol{P_r}(\boldsymbol{t}) = \prod_{u \in \mathcal{N}_0^{(d)}} P_{r_u}(t_u) = \prod_{u \in \mathcal{N}_0^{(d)}} N(d_u, r_u)^{-1} \sum_{l_u \in N(d_u, r_u)} Y_{r_u, l_u}(\boldsymbol{\xi}_u / \sqrt{d_u}) Y_{r_u, l_u}(\boldsymbol{\eta}_u / \sqrt{d_u})$$

$$\tag{82}$$

$$= \boldsymbol{N}(\boldsymbol{d}, \boldsymbol{r})^{-1} \sum_{l \in [\boldsymbol{N}(\boldsymbol{d}, \boldsymbol{r})]} \overline{\boldsymbol{Y}}_{\boldsymbol{r}, l}(\boldsymbol{\xi}) \overline{\boldsymbol{Y}}_{\boldsymbol{r}, l}(\boldsymbol{\eta}). \tag{83}$$

Then Theorem 1 follows by noticing

$$\mathcal{K}(\boldsymbol{r}) \boldsymbol{N}(\boldsymbol{d}, \boldsymbol{r})^{-1} \sim d^{-\mathcal{S}(\boldsymbol{r})} d^{-\sum_{u \in \mathcal{N}_0^{(d)}} r_u \alpha_u} = d^{-\mathcal{L}(\boldsymbol{r})} \tag{84}$$

$\square$

*Proof of Theorem 5.* From the orthogonality,

$$\hat{\mathcal{K}}(\boldsymbol{r}) = \langle \mathcal{K}, \boldsymbol{P_r} \rangle / \|\boldsymbol{P_r}\|_{L^2(I^p, \boldsymbol{\omega})}^2 \tag{85}$$

We begin with the denominator. Note that

$$\|\boldsymbol{P_r}\|_{L^2(I^p, \boldsymbol{\sigma})}^2 = \prod_{u \in \mathcal{N}_0^{(d)}} \|P_{r_u}\|_{L^2(I, \omega_{d_u})}^2 = \boldsymbol{N}(\boldsymbol{d}; \boldsymbol{r})^{-1} \prod_{u \in \mathcal{N}_0^{(d)}} (|\mathbb{S}_{d_u - 1}| / |\mathbb{S}_{d_u - 2}|) \tag{86}$$

By applying Lemma 2, integration by parts and continuity of $\mathcal{K}^{(r)}$ on the boundary $\partial I^p$

$$\langle \mathcal{K}, \boldsymbol{P_r} \rangle_{L^2(I^p, \boldsymbol{\omega})} = c_r \int_{I^p} \mathcal{K}(\boldsymbol{t}) \left( \frac{d}{d\boldsymbol{t}} \right)^r \left( 1 - \boldsymbol{t}^2 \right)^{r + (d-3)/2} d\boldsymbol{t} \tag{87}$$

$$= (-1)^r c_r \int_{I^p} \mathcal{K}^{(r)}(\boldsymbol{t}) \left( 1 - \boldsymbol{t}^2 \right)^{r + (d-3)/2} d\boldsymbol{t} \tag{88}$$

$$= (-1)^r c_r \left( \mathcal{M}(\mathcal{K}, \boldsymbol{d}) + \epsilon(\mathcal{K}, \boldsymbol{d}) \right) \tag{89}$$

where $\mathcal{K}^{(r)}$ is the $r$ derivative of $\mathcal{K}$, the coefficient $c_r$ is given by Lemma 2

$$c_r = \prod_{u \in \mathcal{N}_0^{(d)}} c_{r_u} = \prod_{u \in \mathcal{N}_0^{(d)}} \frac{(-1)^{r_u}}{2^{r_u}(r_u + (d-3)/2)_{r_u}} \sim \prod_{u \in \mathcal{N}_0^{(d)}} (-1)^{r_u} d_u^{-r_u} = (-1)^r d^{-r} \tag{90}$$

and the major and error terms are given by

$$\mathcal{M}(\mathcal{K}, \boldsymbol{d}) \equiv \mathcal{K}^{(r)}(0) \int_{I^p} \left( 1 - \boldsymbol{t}^2 \right)^{r + (d-3)/2} d\boldsymbol{t} = \mathcal{K}^{(r)}(0) \prod_{u \in \mathcal{N}_0^{(d)}} \frac{|\mathbb{S}_{2r_u + d_u - 1}|}{|\mathbb{S}_{2r_u + d_u - 2}|} \tag{91}$$

$$\epsilon(\mathcal{K}, \boldsymbol{d}) \equiv \int_{I^p} \left( \mathcal{K}^{(r)}(\boldsymbol{t}) - \mathcal{K}^{(r)}(0) \right) \left( 1 - \boldsymbol{t}^2 \right)^{r + (d-3)/2} d\boldsymbol{t} \tag{92}$$

The mean value theorem gives

$$|\mathcal{K}^{(r)}(\boldsymbol{t}) - \mathcal{K}^{(r)}(0)| \leq \sum_{u \in \mathcal{N}_0^{(d)}} \|\mathcal{K}^{(r + e_u)}\|_{L^\infty(I^p)} |\boldsymbol{t}_u| \tag{93}$$

and the error term $|\epsilon(\mathcal{K}, \boldsymbol{d})|$ is bounded above by

$$
\int_{I^p} \left(1 - \boldsymbol{t}^2\right)^{\boldsymbol{r}+(\boldsymbol{d}-3)/2} d\boldsymbol{t} \sum_{u \in \mathcal{N}_0^{(d)}} \|\mathcal{K}^{(\boldsymbol{r}+\boldsymbol{e}_u)}\|_{L^\infty(I^p)} \left( \frac{\int_I |\boldsymbol{t}_u| \left(1 - \boldsymbol{t}_u^2\right)^{\boldsymbol{r}_u+(\boldsymbol{d}_u-3)/2} d\boldsymbol{t}_u}{\int_I \left(1 - \boldsymbol{t}_u^2\right)^{\boldsymbol{r}_u+(\boldsymbol{r}_u-3)/2} d\boldsymbol{t}_u} \right) \quad (94)
$$

$$
\sim \prod_{u \in \mathcal{N}_0^{(d)}} \frac{|\mathbb{S}_{2\boldsymbol{r}_u+\boldsymbol{d}_u-1}|}{|\mathbb{S}_{2\boldsymbol{r}_u+\boldsymbol{d}_u-2}|} \sum_{u \in \mathcal{N}_0^{(d)}} \|\mathcal{K}^{(\boldsymbol{r}+\boldsymbol{e}_u)}\|_{L^\infty(I^p)} \boldsymbol{d}_u^{-1} \left( \frac{|\mathbb{S}_{2\boldsymbol{r}_u+\boldsymbol{d}_u-1}|}{|\mathbb{S}_{2\boldsymbol{r}_u+\boldsymbol{d}_u-2}|} \right)^{-1}. \quad (95)
$$

Since for any $\alpha \in \mathbb{N}$, as $\boldsymbol{d}_u \to \infty$,

$$
\frac{|\mathbb{S}_{\alpha+\boldsymbol{d}_u-1}|}{|\mathbb{S}_{\alpha+\boldsymbol{d}_u-2}|} = \pi^{\frac{1}{2}} \Gamma((\alpha + \boldsymbol{d}_u - 1)/2)/\Gamma((\alpha + \boldsymbol{d}_u)/2) \sim \pi^{\frac{1}{2}} (\boldsymbol{d}_u/2)^{-\frac{1}{2}} \sim (\boldsymbol{d}_u)^{-\frac{1}{2}}, \quad (96)
$$

we have

$$
|\epsilon(\mathcal{K}, \boldsymbol{d})| \lesssim \sum_{u \in \mathcal{N}_0^{(d)}} \|\mathcal{K}^{(\boldsymbol{r}+\boldsymbol{e}_u)}\|_{L^\infty(I^p)} \boldsymbol{d}_u^{-\frac{1}{2}} \prod_{u \in \mathcal{N}_0^{(d)}} \frac{|\mathbb{S}_{2\boldsymbol{r}_u+\boldsymbol{d}_u-1}|}{|\mathbb{S}_{2\boldsymbol{r}_u+\boldsymbol{d}_u-2}|}. \quad (97)
$$

We claim that (which will be proved later)

$$
\sum_{u \in \mathcal{N}_0^{(d)}} \|\mathcal{K}^{(\boldsymbol{r}+\boldsymbol{e}_u)}\|_{L^\infty(I^p)} \lesssim d^{-8(\boldsymbol{r})} \quad (98)
$$

which implies

$$
\langle \mathcal{K}, \boldsymbol{P}_{\boldsymbol{r}} \rangle_{L^2(I^p, \omega_{d_{\text{in}}}^p)} = c_{\boldsymbol{r}} \left( \mathcal{K}^{(\boldsymbol{r})}(0) + \mathcal{O}\left( d^{-8(\boldsymbol{r})} (\min_{u \in \mathcal{N}_0^{(d)}} \boldsymbol{d}_u)^{-\frac{1}{2}} \right) \right) \prod_{u \in \mathcal{N}_0^{(d)}} \frac{|\mathbb{S}_{2\boldsymbol{r}_u+\boldsymbol{d}_u-1}|}{|\mathbb{S}_{2\boldsymbol{r}_u+\boldsymbol{d}_u-2}|} \quad (99)
$$

Plugging back to Eq. (85), we have

$$
\hat{\mathcal{K}}(\boldsymbol{r}) = (-1)^{\boldsymbol{r}} c_{\boldsymbol{r}} \boldsymbol{N}(\boldsymbol{d}, \boldsymbol{r}) \left( \mathcal{K}^{(\boldsymbol{r})}(\boldsymbol{0}) + \mathcal{O}\left( d^{-8(\boldsymbol{r})} (\min_{u \in \mathcal{N}_0^{(d)}} \boldsymbol{d}_u)^{-\frac{1}{2}} \right) \right) \left( \prod_{u \in \mathcal{N}_0^{(d)}} \frac{|\mathbb{S}_{2\boldsymbol{r}_u+\boldsymbol{d}_u-1}|}{|\mathbb{S}_{2\boldsymbol{r}_u+\boldsymbol{d}_u-2}|} \left( \frac{|\mathbb{S}_{\boldsymbol{d}_u-1}|}{|\mathbb{S}_{\boldsymbol{d}_u-2}|} \right)^{-1} \right) \quad (100)
$$

Since, for $\boldsymbol{r}$ fixed and as $\boldsymbol{d}_u \to \infty$ for all $u \in \mathcal{N}_0^{(d)}$

$$
\frac{c_{\boldsymbol{r}}}{(-1)^{\boldsymbol{r}} \boldsymbol{d}^{-\boldsymbol{r}}} \to 1 \quad \text{and} \quad \frac{\boldsymbol{N}(\boldsymbol{d}, \boldsymbol{r})}{\boldsymbol{d}^{\boldsymbol{r}}/\boldsymbol{r}!} \to 1 \quad \text{and} \quad \left( \prod_{u \in \mathcal{N}_0^{(d)}} \frac{|\mathbb{S}_{2\boldsymbol{r}_u+\boldsymbol{d}_u-1}|}{|\mathbb{S}_{2\boldsymbol{r}_u+\boldsymbol{d}_u-2}|} \left( \frac{|\mathbb{S}_{\boldsymbol{d}_u-1}|}{|\mathbb{S}_{\boldsymbol{d}_u-2}|} \right)^{-1} \right) \to 1 \quad (101)
$$

and thus

$$
\hat{\mathcal{K}}(\boldsymbol{r}) \sim \boldsymbol{r}!^{-1} \left( \mathcal{K}^{(\boldsymbol{r})}(\boldsymbol{0}) + \mathcal{O}\left( d^{-8(\boldsymbol{r})} (\min_{u \in \mathcal{N}_0^{(d)}} \boldsymbol{d}_u)^{-\frac{1}{2}} \right) \right) \quad (102)
$$

It remains to verify Eq. (98). By Lemma 1, we only need to show that

$$
\sum_{u \in \mathcal{N}_0^{(d)}} d^{-8(\boldsymbol{r}+\boldsymbol{e}_u)} \lesssim d^{-8(\boldsymbol{r})} \quad (103)
$$

We prove this by induction on the number of hidden layers of $\mathcal{G}^{(d)}$. The base case is obvious. Now suppose the depth of $\mathcal{G}^{(d)}$ is $h$. Let $\mathcal{C}(\boldsymbol{r})$ be the set of children of $o_\mathcal{G}$ that are ancestors of at least one node of $n(\boldsymbol{r})$. We split $\mathcal{N}_0^{(d)}$ into two disjoint sets

$$
\mathcal{Q}(\boldsymbol{r}) \equiv \{u \in \mathcal{N}_0^{(d)} : \exists v \in \mathcal{C}(\boldsymbol{r}) \text{ s.t. } \mathcal{P}(u \to v) \neq \emptyset\} \quad \text{and} \quad \mathcal{N}_0^{(d)} \backslash \mathcal{Q}_0(\boldsymbol{r}).
$$

For $u \notin \mathcal{Q}(\boldsymbol{r})$, we have $\mathcal{S}(\boldsymbol{r} + \boldsymbol{e}_u) = \mathcal{S}(\boldsymbol{r}) + \mathcal{S}(\boldsymbol{e}_u)$ and hence

$$\sum_{u \notin \mathcal{Q}(\boldsymbol{r})} d^{-\mathcal{S}(\boldsymbol{r}+\boldsymbol{e}_u)} = \sum_{u \notin \mathcal{Q}(\boldsymbol{r})} d^{-\mathcal{S}(\boldsymbol{r})-\mathcal{S}(\boldsymbol{e}_u)} = d^{-\mathcal{S}(\boldsymbol{r})} \sum_{u \notin \mathcal{Q}(\boldsymbol{r})} d^{-\mathcal{S}(\boldsymbol{e}_u)} \lesssim d^{-\mathcal{S}(\boldsymbol{r})} \,. \tag{104}$$

In the last inequality above, we have used

$$\sum_{u \notin \mathcal{Q}(\boldsymbol{r})} d^{-\mathcal{S}(\boldsymbol{e}_u)} \leq \sum_{u \in \mathcal{N}_0^{(d)}} d^{-\mathcal{S}(\boldsymbol{e}_u)} \sim 1 \,. \tag{105}$$

To estimate the remaining, we use induction. Note that $|\mathcal{C}(\boldsymbol{r})|$ is finite and independent of $d$. Then

$$\sum_{u \in \mathcal{Q}(\boldsymbol{r})} d^{-\mathcal{S}(\boldsymbol{r}+\boldsymbol{e}_u)} \leq \sum_{v \in \mathcal{C}(\boldsymbol{r})} \sum_{u \in \mathcal{Q}(\boldsymbol{r})} d^{-\alpha_{o_{\mathcal{G}}}-\mathcal{S}(n(\boldsymbol{r}+\boldsymbol{e}_u;v))} \tag{106}$$

$$= d^{-\alpha_{o_{\mathcal{G}}}} \sum_{v \in \mathcal{C}(\boldsymbol{r})} \sum_{u \in \mathcal{Q}(\boldsymbol{r})} d^{-\mathcal{S}(n(\boldsymbol{r}+\boldsymbol{e}_u;v))} \tag{107}$$

$$\lesssim |\mathcal{C}(\boldsymbol{r})| d^{-\alpha_{o_{\mathcal{G}}}} \max_{v \in \mathcal{C}(\boldsymbol{r})} d^{-\mathcal{S}(n(\boldsymbol{r};v))} \sim d^{-\mathcal{S}(\boldsymbol{r})} \tag{108}$$

We have used induction on the sub-graph with $v$ as the output node. $\qquad \square$

## I  PROOF OF THEOREM 2

Let $\mathcal{G}^{(d)}$ be a DAG associated to the convolutional networks whose filter sizes in the $l$-th layer is $k_l = [d^{\alpha_l}]$, for $0 \leq l \leq L+1$, in which we treat the *flatten-dense* readout layer as a convolution with filter size $[d^{\alpha_{L+1}}]$. Note that we have set $\alpha_p = \alpha_0$ and $\alpha_w = \alpha_{L+1}$. We also assume an *activation* layer after the *flatten-dense* layer, which does not essentially alter the topology of the DAG.

We need the following dimension counting lemma.

**Lemma 4.** *Let $r \in \mathcal{L}(\mathcal{G}^{(d)})$. Then*

$$\dim\left(\mathrm{span}\left\{\overline{\boldsymbol{Y}}_{\boldsymbol{r},\boldsymbol{l}} : \quad \mathcal{L}(\boldsymbol{r}) = r, \ \boldsymbol{l} \in \boldsymbol{N}(\boldsymbol{d},\boldsymbol{r})\right\}\right) \sim d^r \tag{109}$$

To prove Theorem 2, we only need to verify the assumptions of Theorem 4 in Mei et al. (2021a). For convenience, we briefly recap the assumptions and results from Mei et al. (2021a) in Sec.L.

It is convenient to group the eigenspaces together according to the learning indices $\mathcal{L}(\mathcal{G}^{(d)})$. Recall that $\mathcal{L}(\mathcal{G}^{(d)}) = (r_1 \leq r_2 \leq r_3 \dots)$. Let

$$E_i = \mathrm{span}\{\overline{\boldsymbol{Y}}_{\boldsymbol{r},\boldsymbol{l}} : \mathcal{L}(\boldsymbol{r}) = r_i\} \tag{110}$$

Then by Theorem 1 and Lemma 4,

$$\dim(E_i) \sim d^{r_i} \quad \text{and} \quad \lambda(g) \sim d^{-r_i} \quad \forall g \in E_i, \ g \neq \boldsymbol{0} \,, \tag{111}$$

where $\lambda(g)$ denote the eigenvalue of $g$. We proceed to verify Assumptions 4 and 5 in Sec. L. They follow directly from Theorem 1, Lemma 4 and the hypercontractivity of spherical harmonics Beckner (1992).

### I.1  VERIFYING **Assumption 4**

We need the following.

**Proposition 1.** *For $0 < s \in \mathbb{R}$, let $D_s = span\{\overline{\boldsymbol{Y}}_{\boldsymbol{r},\boldsymbol{l}} : |\mathcal{L}(\boldsymbol{r})| < s\}$. Then for $\boldsymbol{f} \in D_s$,*

$$\|\boldsymbol{f}\|_q^2 \leq (q-1)^{s/\alpha_0} \|\boldsymbol{f}\|_2^2 \tag{112}$$

*Proof of Proposition 1.* The lemma follows from the tensorization of hypercontractivity. Let $f = \sum_{k \geq 0} Y_k \in L^2(\mathbb{S}_n)$ where $Y_k$ is a degree $k$ spherical harmonics in $\mathbb{S}_n$. Define the Poisson semi-group operator

$$P_\epsilon f(x) = \sum_{k \geq 0} \epsilon^k Y_k(x) \tag{113}$$

Then we have the hypercontractivity inequality (Beckner, 1992), for $1 \leq p \leq q$ and $\epsilon \leq \sqrt{\frac{p-1}{q-1}}$

$$\|P_\epsilon f\|_{L^q(\mathbb{S}_n)} \leq \|f\|_{L^p(\mathbb{S}_n)} \tag{114}$$

One can then tensorize (Beckner, 1975) it to obtain the same bound in the tensor space.

**Lemma 5** (Corollary 11 Montanaro (2012)). *Let $\boldsymbol{f} : (\mathbb{S}_n)^k \to \mathbb{R}$. If $1 \leq p \leq q$ and $\epsilon \leq \sqrt{\frac{p-1}{q-1}}$, then*

$$\|P_\epsilon^{\otimes k} \boldsymbol{f}\|_{L^q((\mathbb{S}_n)^k)} \leq \|\boldsymbol{f}\|_{L^p((\mathbb{S}_n)^k)} . \tag{115}$$

Let $\boldsymbol{f} = \sum_{\boldsymbol{r},\boldsymbol{l}} a_{\boldsymbol{r},\boldsymbol{l}} \overline{\boldsymbol{Y}}_{\boldsymbol{r},\boldsymbol{l}} \in D_s$. Choosing $\epsilon = \sqrt{\frac{1}{q-1}}$ and $p = 2$ in the above lemma, we have

$$\|\boldsymbol{f}\|_q^2 = \|\sum_{\boldsymbol{r},\boldsymbol{l}} a_{\boldsymbol{r},\boldsymbol{l}} \overline{\boldsymbol{Y}}_{\boldsymbol{r},\boldsymbol{l}}\|_q^2 \tag{116}$$

$$= \|P_\epsilon^{\otimes |\mathcal{N}_0^{(d)}|} \sum_{\boldsymbol{r},\boldsymbol{l}} a_{\boldsymbol{r},\boldsymbol{l}} \epsilon^{-\boldsymbol{r}} \overline{\boldsymbol{Y}}_{\boldsymbol{r},\boldsymbol{l}}\|_q^2 \tag{117}$$

$$\leq \|\sum_{\boldsymbol{r},\boldsymbol{l}} a_{\boldsymbol{r},\boldsymbol{l}} \epsilon^{-\boldsymbol{r}} \overline{\boldsymbol{Y}}_{\boldsymbol{r},\boldsymbol{l}}\|_2^2 \tag{118}$$

$$= \sum_{\boldsymbol{r},\boldsymbol{l}} a_{\boldsymbol{r},\boldsymbol{l}}^2 \epsilon^{-2\boldsymbol{r}} \|\overline{\boldsymbol{Y}}_{\boldsymbol{r},\boldsymbol{l}}\|_2^2 \tag{119}$$

$$\leq \epsilon^{-2\max|\boldsymbol{r}|} \sum_{\boldsymbol{r},\boldsymbol{l}} a_{\boldsymbol{r},\boldsymbol{l}}^2 \|\overline{\boldsymbol{Y}}_{\boldsymbol{r},\boldsymbol{l}}\|_2^2 \tag{120}$$

$$= (q-1)^{\max|\boldsymbol{r}|} \|\boldsymbol{f}\|_2^2 \leq (q-1)^{s/\alpha_0} \|\boldsymbol{f}\|_2^2 \tag{121}$$

$\square$

Since $r \notin \mathcal{L}(\mathcal{G}^{(d)})$, there is a $j$ such that $r_j < r < r_{j+1}$. Let $n(d) = d^r$ and

$$m(d) = \dim\left(\text{span}\{\overline{\boldsymbol{Y}}_{\boldsymbol{r},\boldsymbol{l}} : \mathcal{L}(\boldsymbol{r}) \leq r_j\}\right) = \dim(\text{span} \bigcup_{i \leq j} E_i) \tag{122}$$

Clearly, $m(d) \sim d^{r_j}$. We list all eigenvalues of $\mathcal{K}$ in non-ascending order as $\{\lambda_{d,i}\}$. In particular, we have

$$\lambda_{d,m(d)} \sim d^{-r_j} > d^{-r} > d^{-r_{j+1}} \sim \lambda_{d,m(d)+1} . \tag{123}$$

**Assumption 4 (a).** We choose $u(d)$ to be

$$u(d) = \dim\left(\text{span} \bigcup_{i:r_i \leq 2r+100} E_i\right) . \tag{124}$$

**Assumption 4 (a)** follows from Proposition 1.

**Assumptions 4 (b).** Let $s = \inf\{\bar{r} \in \mathcal{L}(\mathcal{G}^{(d)}) : \bar{r} > 2r + 100\}$. For $l > 1$, we have

$$\sum_{j=u(d)+1} \lambda_{d,j}^l \sim \sum_{r_i:r_i \geq s} (d^{-r_i})^l \dim(E_i) \sim d^{-s(l-1)} \tag{125}$$

which also holds for $l = 1$ since

$$\sum_{j=u(d)+1} \lambda_{d,j} \sim 1 \tag{126}$$

Thus

$$\frac{(\sum_{j=u(d)+1} \lambda_{d,j}^l)^2}{\sum_{j=u(d)+1} \lambda_{d,j}^{2l}} \sim \frac{d^{-2s(l-1)}}{d^{-s(2l-1)}} = d^s > d^{2r+100} > n(d)^{2+\delta} \sim d^{(2+\delta)r}. \qquad (127)$$

as long as $\delta < 100/r$.

**Assumption 4 (c).** Denote

$$\mathcal{K}_{d,>m(d)}(\boldsymbol{\xi}, \boldsymbol{\eta}) = \sum_{\boldsymbol{r},\boldsymbol{l}:\mathcal{L}(\boldsymbol{r})>r} \lambda_{\mathcal{K}}(\boldsymbol{r}) \overline{\boldsymbol{Y}}_{\boldsymbol{r},\boldsymbol{l}}(\boldsymbol{\xi}) \overline{\boldsymbol{Y}}_{\boldsymbol{r},\boldsymbol{l}}(\boldsymbol{\eta}) \qquad (128)$$

We have

$$\mathbb{E}_{\boldsymbol{\eta}} \mathcal{K}_{d,>m(d)}(\boldsymbol{\xi}, \boldsymbol{\eta})^2 = \mathbb{E}_{\boldsymbol{\eta}} \sum_{\boldsymbol{r},\boldsymbol{l}:\mathcal{L}(\boldsymbol{r})>r} \lambda_{\mathcal{K}}(\boldsymbol{r})^2 |\overline{\boldsymbol{Y}}_{\boldsymbol{r},\boldsymbol{l}}(\boldsymbol{\xi}) \overline{\boldsymbol{Y}}_{\boldsymbol{r},\boldsymbol{l}}(\boldsymbol{\eta})|^2 \qquad (129)$$

$$= \sum_{\boldsymbol{r},\boldsymbol{l}:\mathcal{L}(\boldsymbol{r})>r} \lambda_{\mathcal{K}}(\boldsymbol{r})^2 |\overline{\boldsymbol{Y}}_{\boldsymbol{r},\boldsymbol{l}}(\boldsymbol{\xi})|^2 \qquad (130)$$

$$= \sum_{\boldsymbol{r}:\mathcal{L}(\boldsymbol{r})>r} \lambda_{\mathcal{K}}(\boldsymbol{r})^2 \sum_{\boldsymbol{l}} |\overline{\boldsymbol{Y}}_{\boldsymbol{r},\boldsymbol{l}}(\boldsymbol{\xi})|^2 \qquad (131)$$

$$= \sum_{\boldsymbol{r}:\mathcal{L}(\boldsymbol{r})>r} \lambda_{\mathcal{K}}(\boldsymbol{r})^2 \boldsymbol{N}(d,\boldsymbol{r}) \boldsymbol{P_r}(\boldsymbol{1}) = \sum_{\boldsymbol{r}:\mathcal{L}(\boldsymbol{r})>r} \lambda_{\mathcal{K}}(\boldsymbol{r})^2 \boldsymbol{N}(d,\boldsymbol{r}). \qquad (132)$$

Similarly,

$$\mathbb{E}_{\boldsymbol{\xi},\boldsymbol{\eta}} \sum_{\boldsymbol{r},\boldsymbol{l}:\mathcal{L}(\boldsymbol{r})>r} \lambda_{\mathcal{K}}(\boldsymbol{r})^2 |\overline{\boldsymbol{Y}}_{\boldsymbol{r},\boldsymbol{l}}(\boldsymbol{\xi}) \overline{\boldsymbol{Y}}_{\boldsymbol{r},\boldsymbol{l}}(\boldsymbol{\eta})|^2 = \sum_{\boldsymbol{r}:\mathcal{L}(\boldsymbol{r})>r} \lambda_{\mathcal{K}}(\boldsymbol{r})^2 \boldsymbol{N}(d,\boldsymbol{r}). \qquad (133)$$

Thus

$$\mathbb{E}_{\boldsymbol{\eta}} \sum_{\boldsymbol{r},\boldsymbol{l}:\mathcal{L}(\boldsymbol{r})>r} \lambda_{\mathcal{K}}(\boldsymbol{r})^2 |\overline{\boldsymbol{Y}}_{\boldsymbol{r},\boldsymbol{l}}(\boldsymbol{\xi}) \overline{\boldsymbol{Y}}_{\boldsymbol{r},\boldsymbol{l}}(\boldsymbol{\eta})|^2 - \mathbb{E}_{\boldsymbol{\xi},\boldsymbol{\eta}} \sum_{\boldsymbol{r},\boldsymbol{l}:\mathcal{L}(\boldsymbol{r})>r} \lambda_{\mathcal{K}}(\boldsymbol{r})^2 |\overline{\boldsymbol{Y}}_{\boldsymbol{r},\boldsymbol{l}}(\boldsymbol{\xi}) \overline{\boldsymbol{Y}}_{\boldsymbol{r},\boldsymbol{l}}(\boldsymbol{\eta})|^2 = 0. \quad (134)$$

For the diagonal terms,

$$\mathcal{K}_{d,>m(d)}(\boldsymbol{\xi}, \boldsymbol{\xi}) = \sum_{\boldsymbol{r},\boldsymbol{l}:\mathcal{L}(\boldsymbol{r})>r} \lambda_{\mathcal{K}}(\boldsymbol{r}) |\overline{\boldsymbol{Y}}_{\boldsymbol{r},\boldsymbol{l}}(\boldsymbol{\xi})|^2 = \sum_{\boldsymbol{r}:\mathcal{L}(\boldsymbol{r})>r} \lambda_{\mathcal{K}}(\boldsymbol{r}) \boldsymbol{N}(d,\boldsymbol{r}) = \mathbb{E}_{\boldsymbol{\xi}} \mathcal{K}_{d,>m(d)}(\boldsymbol{\xi}, \boldsymbol{\xi})$$

$$(135)$$

which is deterministic.

## I.2 VERIFYING **Assumption 5**.

Recall that $n(d) \sim d^r$ and $r_j < r < r_{j+1}$. **Assumption 5(a)** follows from Eq. (111). Indeed, for $l > 1$

$$\lambda_{d,m(d)+1}^{-l} \sum_{k=m(d)+1} \lambda_{d,k}^l \sim (d^{-r_{j+1}})^{-l} \sum_{i:r_i \geq r_{j+1}} \dim(E_i) d^{-lr_i} \qquad (136)$$

$$\sim d^{lr_{j+1}} \sum_{i:r_i \geq r_{j+1}} d^{r_i} d^{-lr_i} \qquad (137)$$

$$= d^{r_{j+1}} > n(d)^{1+\delta} \qquad (138)$$

for some $\delta > 0$ since $r < r_{j+1}$. Similarly, for $l = 1$, since

$$\sum_{k=m(d)+1} \lambda_{d,k} \sim 1 \qquad (139)$$

we have

$$(d^{-r_{j+1}})^{-1} \sum_{k=m(d)+1} \lambda_{d,k} \sim d^{r_{j+1}} > n(d)^{1+\delta}. \qquad (140)$$

**Assumption 5(b)** follows from $r > r_j$.
**Assumption 5(c).** Note that

$$\lambda_{d,m(d)}^{-1} \sum_{k=m(d)+1} \lambda_{d,k} \sim \lambda_{d,m(d)}^{-1} \sim d^{r_j} < n(d)^{(1-\delta)} \sim d^{r(1-\delta)} \qquad (141)$$

for some $\delta > 0$ since $r_j < r$.

## J  PROOF OF LEMMA 4

*Proof.* The lemma can be proved by induction. **Base case:** $L = 0$. The network is a S-CNN. WLOG, assume $\alpha_0 \neq 0$ and $\alpha_1 \neq 0$. For $r \in \mathcal{L}(\mathcal{G}^{(d)})$, we know that $r$ can be written as a combination of $\alpha_0$ and $\alpha_1$, i.e. $r = k_0\alpha_0 + k_1\alpha_1$ for some $k_0, k_1 \geq 0$. We say a tuple $(k_0, k_1)$ is $r$-feasible if in addition, there exists $\boldsymbol{r}$ with $\mathcal{S}(\boldsymbol{r}) = k_1\alpha_1$ and $\mathcal{F}(\boldsymbol{r}) = k_0\alpha_0$. Consider the set of all $r$-feasible tuple

$$F(r) \equiv \{(k_0, k_1) : r\text{-feasible}\}. \tag{142}$$

Clearly, $F(r)$ is finite. It suffices to prove that for each $r$-feasible tuple $(k_0, k_1)$,

$$\dim\left(\text{span}\left\{\overline{\boldsymbol{Y}}_{\boldsymbol{r},\boldsymbol{l}} : \mathcal{F}(\boldsymbol{r}) = k_0\alpha_0 \quad \mathcal{S}(\boldsymbol{r}) = k_1\alpha_1,\ \boldsymbol{l} \in \boldsymbol{N}(\boldsymbol{d}, \boldsymbol{r})\right\}\right) \sim d^r \tag{143}$$

Note that there are $\sim (d^{\alpha_1})^{k_1}$ many ways to choose $k_1$ nodes in the penultimate layer. Then the dimension of the above set is about

$$(d^{\alpha_1})^{k_1} \sum_{\substack{(k_{0,1}, \ldots, k_{0,k_1}) \\ k_{0,1} + \cdots + k_{0,k_1} = k_0}} \prod_{j=1}^{k_1} N(d^{\alpha_0}, k_{0,j}) \sim (d^{\alpha_1})^{k_1} \sum_{\substack{(k_{0,1}, \ldots, k_{0,k_1}) \\ k_{0,1} + \cdots + k_{0,k_1} = k_0}} \prod_{j=1}^{k_1} (d^{\alpha_0})^{k_{0,j}} \tag{144}$$

$$\sim d^{k_0\alpha_0 + k_1\alpha_1} = d^r, \tag{145}$$

since the cardinality of the set

$$\{(k_{0,1}, \ldots, k_{0,k_1}) : k_{0,1} + \cdots + k_{0,k_1} = k_0, \quad k_{0,j} \geq 1 \quad k_{0,j} \in \mathbb{N}\}$$

is finite.

**Induction step:** $L \geq 1$. For $\boldsymbol{r}$ with $\mathcal{L}(\boldsymbol{r}) = r$, let $k$ be the number of children of a MST of $n(\boldsymbol{r}; o_{\mathcal{G}})$. Clearly, $k \in [1, [r/\alpha_{L+1}]]$. Then we can classify $\overline{\boldsymbol{Y}}_{\boldsymbol{r},\boldsymbol{l}}$ into at most $[r/\alpha_{L+1}]$ bins: $\{\Omega_k\}_{k=1,\ldots,[r/\alpha_{L+1}]}$ depending on the number of children of $o_{\mathcal{G}}$ in a MST. Let $\Omega_k$ be non-empty. We only need to prove the number of $\overline{\boldsymbol{Y}}_{\boldsymbol{r},\boldsymbol{l}}$ in $\Omega_k$ is $d^r$. Note that there are $\sim (d^{\alpha_{L+1}})^k$ many ways to choose $k$ children from $o_{\mathcal{G}}$. Let $\{u_j\}_{j=1,\ldots,k}$ be one fixed choice and $\{\mathcal{G}_j\}$ be the subgraphs with $\{u_j\}$ as the output nodes. Next, we partition $(r - k\alpha_{L+1})$ into $k$ components,

$$r - k\alpha_{L+1} = r_1 + \cdots + r_k$$

so that each $r_j$ is a combination of $\{\alpha_j\}_{0 \leq j \leq L}$. The cardinality of such partition is also finite. We fix one of such partition $(r_1, \ldots, r_k)$ so that each $r_j$ is a learning index of $\mathcal{G}_j$. We can apply induction to each $(\mathcal{G}_j, r_j)$ to conclude that the cardinality of $\overline{\boldsymbol{Y}}_{\boldsymbol{r}_j, \boldsymbol{l}_j}$ with $\mathcal{L}_{\mathcal{G}_j}(\boldsymbol{r}_j) = r_j$ is $\sim d^{r_j}$, where $\mathcal{L}_{\mathcal{G}_j}(\boldsymbol{r}_j)$ is the learning index of $\boldsymbol{r}_j$ of $\mathcal{G}_j$. Therefore, we have

$$\dim\left(\text{span}\left\{\overline{\boldsymbol{Y}}_{\boldsymbol{r},\boldsymbol{l}} : \quad \mathcal{L}(\boldsymbol{r}) = r,\ \boldsymbol{l} \in \boldsymbol{N}(\boldsymbol{d}, \boldsymbol{r})\right\}\right) \sim (d^{\alpha_{L+1}})^k \prod_{j \in [k]} d^{r_j} = d^r. \tag{146}$$

$\square$

## K  CNN-GAP: CNNs WITH GLOBAL AVERAGE POOLING

Consider convolutional networks whose readout layer is a global average pooling (GAP) and a flattening layer (namely, without pooling), resp.

**CNN + GAP:** $\qquad [Conv(p)\text{-}Act] \rightarrow [Conv(k)\text{-}Act]^{\otimes L} \rightarrow [GAP] \rightarrow [Dense] \tag{147}$

**CNN:** $\qquad [Conv(p)\text{-}Act] \rightarrow [Conv(k)\text{-}Act]^{\otimes L} \rightarrow [Flatten] \rightarrow [Dense] \tag{148}$

Concretely, the input space is $\mathcal{X} = (\overline{\mathbb{S}}_{p-1})^{k^L \times w} \subseteq \mathbb{R}^{p \times 1 \times k^L \times w}$, where $p$ is the patch size of the input convolutional layer, $k$ is the filter size in *hidden* layers, $L$ is the number of *hidden* convolution layers and $w$ is the spatial dimension of the penultimate layer. The total dimension of the input is $d = p \cdot k^L \cdot w$, and the number of input nodes is $|\mathcal{N}_0| = k^L \cdot w$. Since the stride is equal to the filter size for all convolutional layers, the *spatial* dimension is reduced by a factor of $p$ in the first layer, a factor of $k$ by each hidden layer. The penultimate layer (before pooling/flattening)

has spatial dimension $w$ and is reduced to 1 by the *GAP-dense* layer or the *Flatten-dense* layer. The DAGs associated to these two architectures are identical which is denoted by $\mathcal{G}$. However, the kernel/neural network computations are slightly different. If the penultimate layer has $n$ many channels and $f_{pen} : \mathcal{X} \to \mathbb{R}^{n \times w}$ is the mapping from the input layer to the penultimate layer, then the outputs of the *CNN-GAP* and *CNN-Flatten* are

$$f_{\text{CNN-GAP}}(\boldsymbol{x}) = n^{-\frac{1}{2}} \sum_{j \in [n]} \omega_j \fint_{i \in [w]} f_{pen}(\boldsymbol{x})_{j,i} \tag{149}$$

$$f_{\text{CNN-Flatten}}(\boldsymbol{x}) = (nw)^{-\frac{1}{2}} \sum_{j \in [n], i \in [w]} \omega_{ji} f_{pen}(\boldsymbol{x})_{j,i}\,, \tag{150}$$

resp., where $w_j$ and $w_{ji}$ are parameters of the last layer and are usually initialized with standard iid Guassian $w_j, w_{ji} \sim \mathcal{N}(0,1)$. Let $\mathcal{N}_{-1} \subseteq \mathcal{N}$ be the nodes in the penultimate layer, then $|\mathcal{N}_{-1}| = w$. Let $\boldsymbol{\xi} = (\boldsymbol{\xi}_v)_{v \in \mathcal{N}_{-1}} \in \mathcal{X}$, where $\boldsymbol{\xi}_v \in (\overline{\mathbb{S}}_{p-1})^{k^L}$. Thus, each $\boldsymbol{\xi}_v$ contains $k^L$ many input nodes $\{\boldsymbol{\xi}_{u,i}\}_{i \in [k^L]}$. Define

$$\boldsymbol{t}_{uv} = (\langle \boldsymbol{\xi}_{u,i}, \boldsymbol{\eta}_{v,i} \rangle / p)_{i \in [k^L]} \in [-1,1]^{k^L}. \tag{151}$$

Recall that in the case without pooling

$$\mathcal{K}_u(\boldsymbol{t}) = \phi^*(\fint_{uv \in \mathcal{E}} \mathcal{K}_v(\boldsymbol{t})), \quad \mathcal{K}_{\mathcal{G}} = \fint_{o_{\mathcal{G}} v \in \mathcal{E}} \mathcal{K}_v(\boldsymbol{t}) = \fint_{v \in \mathcal{N}_{-1}} \mathcal{K}_v(\boldsymbol{t}) \tag{152}$$

where $\boldsymbol{t} \in [-1,1]^{k^L \times w}$, which is usually obtained by $\boldsymbol{t} = (\langle \boldsymbol{\xi}_{u,i}, \boldsymbol{\eta}_{u,i} \rangle / p)_{u \in \mathcal{N}_{-1}, i \in [k^L]}$. Indeed, for each $v \in \mathcal{N}_{-1}$, $\mathcal{K}_v$ depends only on the *diagonal* terms $\boldsymbol{t}_{vv} = (\langle \boldsymbol{\xi}_{v,i}, \boldsymbol{\eta}_{v,i} \rangle / p)_{i \in [k^L]} \in [-1,1]^{k^L}$. We can find a function

$$\mathcal{K}_{pen} : [-1,1]^{k^L} \to [-1,1] \quad \text{s.t.} \quad \mathcal{K}_v(\boldsymbol{t}) = \mathcal{K}_{pen}(\boldsymbol{t}_{vv}) \quad \forall v \in \mathcal{N}_{-1} \tag{153}$$

Therefore, without pooling the NNGP kernel is

$$\mathcal{K}_{\text{CNN}}(\boldsymbol{t}) = \fint_{v \in \mathcal{N}_{-1}} \mathcal{K}_{pen}(\boldsymbol{t}_{vv}) = \frac{1}{w} \sum_{v \in \mathcal{N}_{-1}} \mathcal{K}_{pen}(\boldsymbol{t}_{vv}) \tag{154}$$

Note that the kernel does not depend on any *off-diagonal* terms $\boldsymbol{t}_{uv}$ with $u \neq v$ because there isn't weight-sharing in the last layer. Let $\boldsymbol{d}_{pen} = (p, p, \ldots, p) \in \mathbb{N}^{k^L}$. Then $\|\boldsymbol{d}_{pen}\|_1 = pk^L$ is the effective dimension of the input to any node $u \in \mathcal{N}_{-1}$. Assume $k = d^{\alpha_k}$, $p = d^{\alpha_p}$ and $w = d^{\alpha_w}$ and $\alpha_k, \alpha_p, \alpha_w > 0$. Applying Theorem 1 to $\mathcal{K}_{pen}$, we have

$$\mathcal{K}_{\text{CNN}}(\boldsymbol{t}) = \sum_{\boldsymbol{r} \in \mathbb{N}^{k^L}} \frac{1}{w} \lambda_{\mathcal{K}_{pen}}(\boldsymbol{r}) \sum_{v \in \mathcal{N}_{-1}} \sum_{\boldsymbol{l} \in \boldsymbol{N}(\boldsymbol{d}_{pen}, \boldsymbol{r})} \overline{\boldsymbol{Y}}_{\boldsymbol{r}, \boldsymbol{l}}(\boldsymbol{\xi}_v), \overline{\boldsymbol{Y}}_{\boldsymbol{r}, \boldsymbol{l}}(\boldsymbol{\eta}_v) \tag{155}$$

Clearly, the eigenfunctions are $\{\overline{\boldsymbol{Y}}_{\boldsymbol{r}, \boldsymbol{l}}(\boldsymbol{\xi}_v)\}_{\boldsymbol{r}, \boldsymbol{l}, v}$ and the corresponding eigenvalues are $\{\frac{1}{w} \lambda_{\mathcal{K}_{pen}}(\boldsymbol{r})\}_{\boldsymbol{r}, \boldsymbol{l}, v}$. Note that

$$\frac{1}{w} \lambda_{\mathcal{K}_{pen}}(\boldsymbol{r}) = \lambda_{\mathcal{K}_{\mathcal{G}}}(\boldsymbol{r}) \sim d^{-\mathcal{L}(\boldsymbol{r})} \tag{156}$$

Here and in what follows, we also consider $\boldsymbol{r} \in \mathbb{N}^{k^L}$ as an element in $\mathbb{N}^{wk^L}$

When the readout layer is a GAP, the weights of the penultimate layer are shared across different spatial locations, namely, all nodes in $\mathcal{N}_{-1}$ use the same weight. As such, the kernel corresponding to the CNN-GAP depends on both the diagonal and off-diagonal terms $\boldsymbol{t} = (\boldsymbol{t}_{uv})_{u,v \in \mathcal{N}_{-1}} \in [-1,1]^{k^L \times k^L}$, which can be written as (Novak et al., 2019b)

$$\mathcal{K}_{\text{CNN-GAP}}(\boldsymbol{t}) = \fint_{u,v \in \mathcal{N}_{-1}} \mathcal{K}_{pen}(\boldsymbol{t}_{uv}) = \frac{1}{w^2} \sum_{u,v \in \mathcal{N}_{-1}} \mathcal{K}_{pen}(\boldsymbol{t}_{uv}) \tag{157}$$

Applying Theorem 1 to $\mathcal{K}_{pen}$, we have

$$\mathcal{K}_{\text{CNN-GAP}}(\boldsymbol{t}) = \frac{1}{w^2} \sum_{\boldsymbol{r} \in \mathbb{N}^{kL}} \lambda_{\mathcal{K}_{pen}}(\boldsymbol{r}) \sum_{u,v \in \mathcal{N}_{-1}} \sum_{\boldsymbol{l} \in \boldsymbol{N}(\boldsymbol{d}_{pen},\boldsymbol{r})} \overline{\boldsymbol{Y}}_{\boldsymbol{r},\boldsymbol{l}}(\boldsymbol{\xi}_u), \overline{\boldsymbol{Y}}_{\boldsymbol{r},\boldsymbol{l}}(\boldsymbol{\eta}_v) \tag{158}$$

$$= \sum_{\boldsymbol{r} \in \mathbb{N}^{kL}} \frac{1}{w} \lambda_{\mathcal{K}_{pen}}(\boldsymbol{r}) \sum_{\boldsymbol{l} \in \boldsymbol{N}(\boldsymbol{d}_{pen},\boldsymbol{r})} \left( w^{-\frac{1}{2}} \int_{u \in \mathcal{N}_{-1}} \overline{\boldsymbol{Y}}_{\boldsymbol{r},\boldsymbol{l}}(\boldsymbol{\xi}_u) \right) \left( w^{-\frac{1}{2}} \int_{u \in \mathcal{N}_{-1}} \overline{\boldsymbol{Y}}_{\boldsymbol{r},\boldsymbol{l}}(\boldsymbol{\eta}_u) \right) \tag{159}$$

$$= \sum_{\boldsymbol{r} \in \mathbb{N}^{kL}} \frac{1}{w} \lambda_{\mathcal{K}_{pen}}(\boldsymbol{r}) \sum_{\boldsymbol{l} \in \boldsymbol{N}(\boldsymbol{d}_{pen},\boldsymbol{r})} \overline{\boldsymbol{Y}}_{\boldsymbol{r},\boldsymbol{l}}^{\text{Sym}}(\boldsymbol{\xi}) \overline{\boldsymbol{Y}}_{\boldsymbol{r},\boldsymbol{l}}^{\text{Sym}}(\boldsymbol{\eta}) \tag{160}$$

where

$$\overline{\boldsymbol{Y}}_{\boldsymbol{r},\boldsymbol{l}}^{\text{Sym}}(\boldsymbol{\xi}) \equiv w^{-\frac{1}{2}} \int_{u \in \mathcal{N}_{-1}} \overline{\boldsymbol{Y}}_{\boldsymbol{r},\boldsymbol{l}}(\boldsymbol{\xi}_u) \quad , \quad \boldsymbol{r} \in \mathbb{N}^{kL} \text{ and } \quad \boldsymbol{l} \in \boldsymbol{N}(\boldsymbol{d}_{pen},\boldsymbol{r}) \tag{161}$$

That is the eigenfunctions and eigenvalues of $\mathcal{K}_{\text{CNN-GAP}}$ are $\{\overline{\boldsymbol{Y}}_{\boldsymbol{r},\boldsymbol{l}}^{\text{Sym}}(\boldsymbol{\xi})\}_{\boldsymbol{r},\boldsymbol{l}}$ and $\{\frac{1}{w}\lambda_{\mathcal{K}_{pen}}(\boldsymbol{r})\}_{\boldsymbol{r},\boldsymbol{l}}$ resp.

In sum, the eigenvalues of $\mathcal{K}_{\text{CNN}}$ and $\mathcal{K}_{\text{CNN-GAP}}$ are the same (up to the multiplicity factor $w$). Each eigenspace of $\mathcal{K}_{\text{CNN-GAP}}$ is given by symmetric polynomials of the form Eq. (161). We can see that the GAP reduces the dimension of each eigenspace by a factor of $w$. Same arguments can also be applied to the NTKs

$$\Theta_{\text{CNN}}(\boldsymbol{t}) = \fint_{v \in \mathcal{N}_{-1}} (\mathcal{K}_{pen}(\boldsymbol{t}_{vv}) + \Theta_{pen}(\boldsymbol{t}_{vv})) \tag{162}$$

$$= \sum_{\boldsymbol{r} \in \mathbb{N}^{kL}} \left( \frac{1}{w} \lambda_{\mathcal{K}_{pen}}(\boldsymbol{r}) + \frac{1}{w} \lambda_{\Theta_{pen}}(\boldsymbol{r}) \right) \sum_{v \in \mathcal{N}_{-1}} \sum_{\boldsymbol{l} \in \boldsymbol{N}(\boldsymbol{d}_{pen},\boldsymbol{r})} \overline{\boldsymbol{Y}}_{\boldsymbol{r},\boldsymbol{l}}(\boldsymbol{\xi}_v), \overline{\boldsymbol{Y}}_{\boldsymbol{r},\boldsymbol{l}}(\boldsymbol{\eta}_v) \tag{163}$$

$$\Theta_{\text{CNN-GAP}}(\boldsymbol{t}) = \fint_{u,v \in \mathcal{N}_{-1}} (\mathcal{K}_{pen}(\boldsymbol{t}_{uv}) + \Theta_{pen}(\boldsymbol{t}_{uv})) \tag{164}$$

$$= \sum_{\boldsymbol{r} \in \mathbb{N}^{kL}} \left( \frac{1}{w} \lambda_{\mathcal{K}_{pen}}(\boldsymbol{r}) + \frac{1}{w} \lambda_{\Theta_{pen}}(\boldsymbol{r}) \right) \sum_{v \in \mathcal{N}_{-1}} \overline{\boldsymbol{Y}}_{\boldsymbol{r},\boldsymbol{l}}^{\text{Sym}}(\boldsymbol{\xi}), \overline{\boldsymbol{Y}}_{\boldsymbol{r},\boldsymbol{l}}^{\text{Sym}}(\boldsymbol{\eta}) \tag{165}$$

where $\Theta_{pen}$ is the NTK of the penultimate layer which is the same for all nodes in $\mathcal{N}_{-1}$. Since $\frac{1}{w}\lambda_{\Theta_{pen}} \sim d^{-\mathcal{L}(\boldsymbol{r})}$, we have

$$\frac{1}{w} \lambda_{\mathcal{K}_{pen}}(\boldsymbol{r}) + \frac{1}{w} \lambda_{\Theta_{pen}}(\boldsymbol{r}) \sim d^{-\mathcal{L}(\boldsymbol{r})} \tag{166}$$

### K.1 Generalization bound of CNN-GAP

We show that GAP improves the data efficiency of D-CNNs by a factor of $w \sim d^{\alpha_w}$ under a stronger assumptions on activations $\phi$.

**Assumption Poly-$\phi$:** There is a sufficiently large $J \in \mathbb{N}$ such that for all hiddens nodes $u$

$$\phi_u^{*(j)}(0) \neq 0 \quad \text{for} \quad 1 \leq j \leq J \quad \text{and} \quad \phi_u^{*(j)}(0) = 0 \quad \text{otherwise} \tag{167}$$

This assumption implies that there are $0 < J_1 < J_2 \in \mathbb{R}$ such that for all $\boldsymbol{0} \neq \boldsymbol{r}$ with $|\boldsymbol{r}| < J_1$

$$\frac{d^{\boldsymbol{r}}}{d\boldsymbol{t}} \mathcal{K}_{pen}(\boldsymbol{0}) \neq 0 \quad \text{and} \quad \frac{d^{\boldsymbol{r}}}{d\boldsymbol{t}} \Theta_{pen}(\boldsymbol{0}) \neq 0 \tag{168}$$

and for all $\boldsymbol{r}$ with $|\boldsymbol{r}| > J_2$

$$\frac{d^{\boldsymbol{r}}}{d\boldsymbol{t}} \mathcal{K}_{pen} \equiv \boldsymbol{0} \quad \text{and} \quad \frac{d^{\boldsymbol{r}}}{d\boldsymbol{t}} \Theta_{pen} \equiv \boldsymbol{0} \tag{169}$$

Moreover, $J_1 \to \infty$ as $J \to \infty$.

Let $L_{\text{Sym}}^p(\boldsymbol{\mathcal{X}}) \leq L^p(\boldsymbol{\mathcal{X}})$ be the close subspace spanned by symmetric eigenfunctions Eq. (161). Let $\mathcal{K}_{\text{Sym}} = \mathscr{K}_{\text{CNN-GAP}}$ or $\Theta_{\text{CNN-GAP}}$.

For $X \subseteq \boldsymbol{\mathcal{X}}$ and $r \notin \mathcal{L}(\mathcal{G}^{(d)})$, define the regressor and the projection operator to be

$$\boldsymbol{R}_X^{\text{Sym}}(f)(x) = \mathcal{K}_{\text{Sym}}(x, X)\mathcal{K}_{\text{Sym}}(X, X)^{-1}f(X)$$

$$\boldsymbol{P}_{>r}^{\text{Sym}}(f) = \sum_{\boldsymbol{r}:\mathcal{L}(\boldsymbol{r})>r} \sum_{\boldsymbol{l}\in \boldsymbol{N}(\boldsymbol{d}_{pen},\boldsymbol{r})} \langle f, \overline{\boldsymbol{Y}}_{\boldsymbol{r},\boldsymbol{l}}^{\text{Sym}}\rangle_{L^2(\boldsymbol{\mathcal{X}})}\overline{\boldsymbol{Y}}_{\boldsymbol{r},\boldsymbol{l}}^{\text{Sym}}.$$

**Theorem 6.** *Let $\mathcal{G} = \{\mathcal{G}^{(d)}\}_d$, where each $\mathcal{G}^{(d)}$ is a DAG associated to the D-CNN in Eq. (147) with $\alpha_k, \alpha_p, \alpha_w > 0$. Let $r \notin \mathcal{L}(\mathcal{G}^{(d)})$ be fixed and the activations satisfy **Assumption Poly-$\phi$** for $J = J(r)$ sufficiently large. Let $f \in L_{\text{Sym}}^2(\boldsymbol{\mathcal{X}})$ with $\mathbb{E}_{\boldsymbol{\sigma}} f = 0$. Then for $\epsilon > 0$,*

$$\left| \left\|\boldsymbol{R}_X^{\text{Sym}}(f) - f\right\|_{L_{\text{Sym}}^2(\boldsymbol{\mathcal{X}})}^2 - \left\|\boldsymbol{P}_{>r}^{\text{Sym}}(f)\right\|_{L_{\text{Sym}}^2(\boldsymbol{\mathcal{X}})}^2 \right| = c_{d,\epsilon}\|f\|_{L_{\text{Sym}}^2(\boldsymbol{\mathcal{X}})}^2, \tag{170}$$

*where $c_{d,\epsilon} \to 0$ in probability as $d \to \infty$ over $X \sim \boldsymbol{\sigma}^{[d^{r-\alpha_w}]}$.*

*Proof of Theorem 6.* We need the following dimension counting lemma, which follows directly from Lemma 4.

**Lemma 6.** *Let $r \in \mathcal{L}(\mathcal{G}^{(d)})$. Then*

$$\dim\left(\text{span}\left\{\overline{\boldsymbol{Y}}_{\boldsymbol{r},\boldsymbol{l}}^{\text{Sym}}: \quad \mathcal{L}(\boldsymbol{r}) = r, \ \boldsymbol{l} \in \boldsymbol{N}(d,\boldsymbol{r})\right\}\right) \sim d^{r-\alpha_w} \tag{171}$$

Recall that $\mathscr{L}(\mathcal{G}^{(d)}) = \{r_j\}$ in non-descending order. Similarly, let

$$E_i^{\text{Sym}} = \text{span}\{\overline{\boldsymbol{Y}}_{\boldsymbol{r},\boldsymbol{l}}^{\text{Sym}}: \quad \mathscr{L}(\boldsymbol{r}) = r_i\} \tag{172}$$

From the above lemma, we have

$$\dim(E_i^{\text{Sym}}) \sim d^{r_i-\alpha_w} \tag{173}$$

Since $r \notin \mathscr{L}(\mathcal{G}^{(d)})$, there is a $j$ such that $r_j < r < r_{j+1}$. Let $n(d) = d^{r-\alpha_w}$ and

$$m(d) = \dim\left(\text{span}\{\overline{\boldsymbol{Y}}_{\boldsymbol{r},\boldsymbol{l}}^{\text{Sym}}: \mathscr{L}(\boldsymbol{r}) \leq r_j\}\right) = \dim(\text{span}\bigcup_{i\leq j} E_i^{\text{Sym}}) \tag{174}$$

Clearly, $m(d) \sim d^{r_j-\alpha_w}$. We list all eigenvalues of $\mathcal{K}_{\text{Sym}}$ in non-ascending order as $\{\lambda_{d,i}\}$. In particular, we have

$$\lambda_{d,m(d)} \sim d^{-r_j} > d^{-r} > d^{-r_{j+1}} \sim \lambda_{d,m(d)+1}. \tag{175}$$

We proceed to verify Assumptions 4 and 5 in Sec. L.

**Assumptions 4 (a)** We choose $u(d)$ to be

$$u(d) = \dim\left(\text{span}\bigcup_{i:r_i\leq 2r+100} E_i^{\text{Sym}}\right). \tag{176}$$

Let $s = \inf\{\bar{r} \in \mathscr{L}(\mathcal{G}^{(d)}): \bar{r} > 2r + 100\}$. Assumption 4 (a) follows from Proposition 1.

**Assumptions 4 (b)** For $l > 1$, we have

$$\sum_{j=u(d)+1} \lambda_{d,j}^l \sim \sum_{r_i:r_i\geq s} (d^{-r_i})^l \dim(E_i^{\text{Sym}}) \sim d^{-s(l-1)-\alpha_w} \tag{177}$$

which also holds for $l = 1$ since

$$d^{\alpha_w}\sum_{j=u(d)+1} \lambda_{d,j} \sim 1 \tag{178}$$

Thus

$$\frac{(\sum_{j=u(d)+1} \lambda_{d,j}^l)^2}{\sum_{j=u(d)+1} \lambda_{d,j}^{2l}} \sim \frac{d^{-2s(l-1)-2\alpha_w}}{d^{-s(2l-1)-\alpha_w}} = d^{s-\alpha_w} > d^{2r+100-\alpha_w} > n(d)^{2+\delta} \sim d^{(2+\delta)(r-\alpha_w)}.$$
(179)

**Assumption 4 (c)** This requires some work and is verified in Sec. K.2.

**Assumption 5 (a)** Since $m(d) = \dim(\text{span} \bigcup_{i \leq j} E_i)$ and $r_{j+1} > r > r_j$, we have

$$\frac{1}{\lambda_{d,m(d)+1}} \sum_{j \geq m(d)+1} \lambda_{d,j}^l \sim \frac{1}{(d^{-r_{j+1}})^l} \sum_{i>j} (d^{-r_i})^l \dim(E_i^{\text{Sym}})$$
(180)

$$\sim \dim(E_{j+1}^{\text{Sym}}) = d^{r_{j+1}-\alpha_w}$$
(181)

$$> n(d)^{1+\delta} = d^{(r-\alpha_w)(1+\delta)}$$
(182)

as long as $\delta < (r_{j+1} - \alpha_w)/(r - \alpha_w) - 1$.

**Assumption 5 (b)** This is obvious since $m(d) \sim d^{r_j - \alpha_w}$, $n(d) \sim d^{r-\alpha_w}$ and $r > r_j$.

**Assumption 5 (c)** This follows from $r_j < r$. Indeed,

$$\frac{1}{\lambda_{d,m(d)}} \sum_{j \geq m(d)+1} \lambda_{d,j} \sim \frac{1}{(d^{-r_j})} \sum_{i>j} (d^{-r_i}) \dim(E_i^{\text{Sym}}) \sim d^{r_j-\alpha_w}$$
(183)

$$\leq n(d)^{1-\delta} = d^{(r-\alpha_w)(1-\delta)}$$
(184)

as long as $0 < \delta < 1 - (r_j - \alpha_w)/(r - \alpha_w)$. $\qquad\square$

## K.2 VERIFICATION OF ASSUMPTIONS 4(C).

We begin with proving Eq. (227). Let $X_i$ define the random variable

$$X_i \equiv \mathbb{E}_{\boldsymbol{\xi} \sim \boldsymbol{\sigma}_d} \mathcal{K}_{\text{Sym},>m(d)}(\boldsymbol{\xi}_i, \boldsymbol{\xi})^2 \quad \text{and} \quad \Delta_i \equiv X_i - \mathbb{E}X_i$$
(185)

We need to show that

$$\frac{\sup_{i \in [n(d)]} |\Delta_i|}{\mathbb{E}X_i} \xrightarrow[d \to \infty]{\text{in prob.}} 0.$$
(186)

By Markov's inequality, it suffices to show that

$$(\mathbb{E}X_i)^{-1} \mathbb{E} \sup_{i \in [n(d)]} |\Delta_i| \xrightarrow[d \to \infty]{} 0$$
(187)

By orthogonality and treating $\boldsymbol{r} \in \mathbb{N}^{k^L}$ as an element of $\mathbb{N}^{wk^L}$, we have

$$\mathbb{E}X_i = \mathbb{E}_{\boldsymbol{\xi}, \bar{\boldsymbol{\xi}} \sim \boldsymbol{\sigma}_d} \mathcal{K}_{\text{Sym},>m(d)}(\boldsymbol{\xi}, \bar{\boldsymbol{\xi}})^2$$
(188)

$$= \mathbb{E}_{\boldsymbol{\xi}, \bar{\boldsymbol{\xi}} \sim \boldsymbol{\sigma}_d} \left| \sum_{\boldsymbol{r}:\mathscr{L}(\boldsymbol{r})>r} \widehat{\mathcal{K}}_{\text{Sym}}(\boldsymbol{r}) \sum_{\boldsymbol{l} \in \boldsymbol{N}(\boldsymbol{d}_{pen},\boldsymbol{r})} \overline{\boldsymbol{Y}}_{\boldsymbol{r},\boldsymbol{l}}^{\text{Sym}}(\boldsymbol{\xi}) \overline{\boldsymbol{Y}}_{\boldsymbol{r},\boldsymbol{l}}^{\text{Sym}}(\bar{\boldsymbol{\xi}}) \right|^2$$
(189)

$$= \mathbb{E}_{\boldsymbol{\xi}, \bar{\boldsymbol{\xi}} \sim \boldsymbol{\sigma}_d} \sum_{\boldsymbol{r}:\mathscr{L}(\boldsymbol{r})>r} \widehat{\mathcal{K}}_{\text{Sym}}^2(\boldsymbol{r}) \sum_{\boldsymbol{l} \in \boldsymbol{N}(\boldsymbol{d}_{pen},\boldsymbol{r})} \left| \overline{\boldsymbol{Y}}_{\boldsymbol{r},\boldsymbol{l}}^{\text{Sym}}(\boldsymbol{\xi}) \overline{\boldsymbol{Y}}_{\boldsymbol{r},\boldsymbol{l}}^{\text{Sym}}(\bar{\boldsymbol{\xi}}) \right|^2$$
(190)

$$= \sum_{\boldsymbol{r}:\mathscr{L}(\boldsymbol{r})>r} \widehat{\mathcal{K}}_{\text{Sym}}^2(\boldsymbol{r}) \sum_{\boldsymbol{l} \in \boldsymbol{N}(\boldsymbol{d}_{pen},\boldsymbol{r})} \mathbb{E}_{\boldsymbol{\xi}, \bar{\boldsymbol{\xi}} \sim \boldsymbol{\sigma}_d} \left| \overline{\boldsymbol{Y}}_{\boldsymbol{r},\boldsymbol{l}}^{\text{Sym}}(\boldsymbol{\xi}) \overline{\boldsymbol{Y}}_{\boldsymbol{r},\boldsymbol{l}}^{\text{Sym}}(\bar{\boldsymbol{\xi}}) \right|^2$$
(191)

$$= \sum_{\boldsymbol{r}:\mathscr{L}(\boldsymbol{r})>r} \widehat{\mathcal{K}}_{\text{Sym}}^2(\boldsymbol{r}) \sum_{\boldsymbol{l} \in \boldsymbol{N}(\boldsymbol{d}_{pen},\boldsymbol{r})} 1$$
(192)

$$= \sum_{\boldsymbol{r}:\mathscr{L}(\boldsymbol{r})>r} \widehat{\mathcal{K}}_{\text{Sym}}^2(\boldsymbol{r}) |\boldsymbol{N}(\boldsymbol{d}_{pen},\boldsymbol{r})|$$
(193)

and

$$X_i = \mathbb{E}_{\boldsymbol{\xi} \sim \boldsymbol{\sigma}_d} \mathcal{K}_{\text{Sym}, > m(d)}(\boldsymbol{\xi}_i, \boldsymbol{\xi})^2 \tag{194}$$

$$= \mathbb{E}_{\boldsymbol{\xi} \sim \boldsymbol{\sigma}_d} \left| \sum_{\boldsymbol{r}: \mathscr{L}(\boldsymbol{r}) > r} \widehat{\mathcal{K}}_{\text{Sym}}(\boldsymbol{r}) \sum_{\boldsymbol{l} \in \boldsymbol{N}(\boldsymbol{d}_{pen}, \boldsymbol{r})} \overline{\boldsymbol{Y}}_{\boldsymbol{r}, \boldsymbol{l}}^{\text{Sym}}(\boldsymbol{\xi}_i) \overline{\boldsymbol{Y}}_{\boldsymbol{r}, \boldsymbol{l}}^{\text{Sym}}(\boldsymbol{\xi}) \right|^2 \tag{195}$$

$$= \mathbb{E}_{\boldsymbol{\xi} \sim \boldsymbol{\sigma}_d} \sum_{\boldsymbol{r}: \mathscr{L}(\boldsymbol{r}) > r} \widehat{\mathcal{K}}_{\text{Sym}}^2(\boldsymbol{r}) \sum_{\boldsymbol{l} \in \boldsymbol{N}(\boldsymbol{d}_{pen}, \boldsymbol{r})} \left| \overline{\boldsymbol{Y}}_{\boldsymbol{r}, \boldsymbol{l}}^{\text{Sym}}(\boldsymbol{\xi}_i) \overline{\boldsymbol{Y}}_{\boldsymbol{r}, \boldsymbol{l}}^{\text{Sym}}(\boldsymbol{\xi}) \right|^2 \tag{196}$$

$$= \sum_{\boldsymbol{r}: \mathscr{L}(\boldsymbol{r}) > r} \widehat{\mathcal{K}}_{\text{Sym}}^2(\boldsymbol{r}) \sum_{\boldsymbol{l} \in \boldsymbol{N}(\boldsymbol{d}_{pen}, \boldsymbol{r})} \mathbb{E}_{\boldsymbol{\xi} \sim \boldsymbol{\sigma}_d} \left| \overline{\boldsymbol{Y}}_{\boldsymbol{r}, \boldsymbol{l}}^{\text{Sym}}(\boldsymbol{\xi}_i) \overline{\boldsymbol{Y}}_{\boldsymbol{r}, \boldsymbol{l}}^{\text{Sym}}(\boldsymbol{\xi}) \right|^2 \tag{197}$$

$$= \sum_{\boldsymbol{r}: \mathscr{L}(\boldsymbol{r}) > r} \widehat{\mathcal{K}}_{\text{Sym}}^2(\boldsymbol{r}) \sum_{\boldsymbol{l} \in \boldsymbol{N}(\boldsymbol{d}_{pen}, \boldsymbol{r})} |\overline{\boldsymbol{Y}}_{\boldsymbol{r}, \boldsymbol{l}}^{\text{Sym}}(\boldsymbol{\xi}_i)|^2 \tag{198}$$

$$= \sum_{\boldsymbol{r}: \mathscr{L}(\boldsymbol{r}) > r} \widehat{\mathcal{K}}_{\text{Sym}}^2(\boldsymbol{r}) \sum_{\boldsymbol{l} \in \boldsymbol{N}(\boldsymbol{d}_{pen}, \boldsymbol{r})} \left( \frac{1}{w} \sum_{u} \overline{\boldsymbol{Y}}_{\boldsymbol{r}, \boldsymbol{l}}(\boldsymbol{\xi}_{i,u})^2 + \frac{1}{w} \sum_{u \neq v} \overline{\boldsymbol{Y}}_{\boldsymbol{r}, \boldsymbol{l}}(\boldsymbol{\xi}_{i,u}) \overline{\boldsymbol{Y}}_{\boldsymbol{r}, \boldsymbol{l}}(\boldsymbol{\xi}_{i,v}) \right) \tag{199}$$

$$= \mathbb{E}_{\boldsymbol{\xi}, \bar{\boldsymbol{\xi}} \sim \boldsymbol{\sigma}_d} \mathcal{K}_{\text{Sym}, > m(d)}(\boldsymbol{\xi}, \bar{\boldsymbol{\xi}})^2 + \sum_{\boldsymbol{r}: \mathscr{L}(\boldsymbol{r}) > r} \widehat{\mathcal{K}}_{\text{Sym}}^2(\boldsymbol{r}) \sum_{\boldsymbol{l} \in \boldsymbol{N}(\boldsymbol{d}_{pen}, \boldsymbol{r})} \left( \frac{1}{w} \sum_{u \neq v} \overline{\boldsymbol{Y}}_{\boldsymbol{r}, \boldsymbol{l}}(\boldsymbol{\xi}_{i,u}) \overline{\boldsymbol{Y}}_{\boldsymbol{r}, \boldsymbol{l}}(\boldsymbol{\xi}_{i,v}) \right) \tag{200}$$

Let

$$X_{\boldsymbol{r}, i} = \sum_{\boldsymbol{l} \in \boldsymbol{N}(\boldsymbol{d}_{pen}, \boldsymbol{r})} \left( \frac{1}{w} \sum_{u \neq v} \overline{\boldsymbol{Y}}_{\boldsymbol{r}, \boldsymbol{l}}(\boldsymbol{\xi}_{i,u}) \overline{\boldsymbol{Y}}_{\boldsymbol{r}, \boldsymbol{l}}(\boldsymbol{\xi}_{i,v}) \right) \tag{201}$$

then

$$\Delta_i = \sum_{\boldsymbol{r}: \mathscr{L}(\boldsymbol{r}) > r} \widehat{\mathcal{K}}_{\text{Sym}}^2(\boldsymbol{r}) X_{\boldsymbol{r}, i} \tag{202}$$

We replace the maximal function by the $l^q$-norm for $q \geq 1$,

$$\mathbb{E} \sup_{i \in [n(d)]} |\Delta_i| \leq \mathbb{E} \left( \sum_{i \in [n(d)]} |\Delta_i|^q \right)^{\frac{1}{q}} \leq \left( \mathbb{E} \sum_{i \in [n(d)]} |\Delta_i|^q \right)^{\frac{1}{q}} = n(d)^{\frac{1}{q}} (\mathbb{E} |\Delta_i|^q)^{\frac{1}{q}} \tag{203}$$

where the first three expectations are taken over $\boldsymbol{\xi}_1, \ldots, \boldsymbol{\xi}_{n(d)} \sim \boldsymbol{\sigma}_d$ and the last one is taken over $\boldsymbol{\xi}_i \sim \boldsymbol{\sigma}_d$. Then we replace the $L^q$-norm by the $L^2$-norm via Hypercontractivity, in which we used **Assumption Poly-$\phi$** which implies that $\Delta_i$ is a polynomial of bounded degree

$$\mathbb{E} \sup_{i \in [n(d)]} |\Delta_i| \leq n(d)^{\frac{1}{q}} (\mathbb{E} |\Delta_i|^q)^{\frac{1}{q}} \leq C_q n(d)^{\frac{1}{q}} (\mathbb{E} |\Delta_i|^2)^{\frac{1}{2}} = C_q n(d)^{\frac{1}{q}} (\mathbb{E} \Delta_i^2)^{\frac{1}{2}} \tag{204}$$

We expand the $L^2$-norm and use orthogonality to "erase" the off-diagonal terms twice: first for $\boldsymbol{r} \neq \bar{\boldsymbol{r}}$

$$\mathbb{E}_{\boldsymbol{\xi}_i \sim \boldsymbol{\sigma}_d} X_{\boldsymbol{r}, i} X_{\bar{\boldsymbol{r}}, i} = 0 \tag{205}$$

and second for $\boldsymbol{l} \neq \boldsymbol{l}'$ or $u \neq v$

$$\mathbb{E}_{\boldsymbol{\xi}_i \sim \boldsymbol{\sigma}_d} X_{\boldsymbol{r}, i}^2 = \frac{1}{w^2} \mathbb{E}_{\boldsymbol{\xi}_i \sim \boldsymbol{\sigma}_d} \sum_{\boldsymbol{l} \in \boldsymbol{N}(\boldsymbol{d}_{pen}, \boldsymbol{r})} \left( \sum_{u \neq v} \overline{\boldsymbol{Y}}_{\boldsymbol{r}, \boldsymbol{l}}(\boldsymbol{\xi}_{i,u}) \overline{\boldsymbol{Y}}_{\boldsymbol{r}, \boldsymbol{l}}(\boldsymbol{\xi}_{i,v}) \right) \sum_{\boldsymbol{l}'} \left( \sum_{u' \neq v'} \overline{\boldsymbol{Y}}_{\boldsymbol{r}, \boldsymbol{l}'}(\boldsymbol{\xi}_{i,u'}) \overline{\boldsymbol{Y}}_{\boldsymbol{r}, \boldsymbol{l}'}(\boldsymbol{\xi}_{i,v'}) \right) \tag{206}$$

$$= \frac{2}{w^2} \mathbb{E}_{\boldsymbol{\xi}_i \sim \boldsymbol{\sigma}_d} \sum_{\boldsymbol{l} \in \boldsymbol{N}(\boldsymbol{d}_{pen}, \boldsymbol{r})} \left( \sum_{u \neq v} \overline{\boldsymbol{Y}}_{\boldsymbol{r}, \boldsymbol{l}}(\boldsymbol{\xi}_{i,u})^2 \overline{\boldsymbol{Y}}_{\boldsymbol{r}, \boldsymbol{l}}(\boldsymbol{\xi}_{i,v})^2 \right) = \frac{2}{w^2} \sum_{\boldsymbol{l} \in \boldsymbol{N}(\boldsymbol{d}_{pen}, \boldsymbol{r})} w(w - 1) \tag{207}$$

$$= \frac{2(w - 1)}{w} \sum_{\boldsymbol{l}} 1 = \frac{2(w - 1)}{w} |\boldsymbol{N}(\boldsymbol{d}_{pen}, \boldsymbol{r})| \leq 2 |\boldsymbol{N}(\boldsymbol{d}_{pen}, \boldsymbol{r})| \tag{208}$$

Combining this estimate with Eq. (193), Eq. (202) and Eq. (204) yields

$$(\mathbb{E}X_i)^{-1}(\mathbb{E}\sup_{i\in[n(d)]}|\Delta_i|) \leq C_q n(d)^{\frac{1}{q}} \frac{(2\sum_{\boldsymbol{r}:\mathscr{L}(\boldsymbol{r})>r}\widehat{\mathcal{K}}_{\mathrm{Sym}}^4(\boldsymbol{r})|\boldsymbol{N}(\boldsymbol{d}_{pen},\boldsymbol{r})|)^{1/2}}{\sum_{\boldsymbol{r}:\mathscr{L}(\boldsymbol{r})>r}\widehat{\mathcal{K}}_{\mathrm{Sym}}^2(\boldsymbol{r})|\boldsymbol{N}(\boldsymbol{d}_{pen},\boldsymbol{r})|} \tag{209}$$

$$\leq \sqrt{2}C_q n(d)^{\frac{1}{q}} \frac{\sum_{\boldsymbol{r}:\mathscr{L}(\boldsymbol{r})>r}\widehat{\mathcal{K}}_{\mathrm{Sym}}^2(\boldsymbol{r})|\boldsymbol{N}(\boldsymbol{d}_{pen},\boldsymbol{r})|^{1/2}}{\sum_{\boldsymbol{r}:\mathscr{L}(\boldsymbol{r})>r}\widehat{\mathcal{K}}_{\mathrm{Sym}}^2(\boldsymbol{r})|\boldsymbol{N}(\boldsymbol{d}_{pen},\boldsymbol{r})|} \tag{210}$$

$$\leq \sqrt{2}C_q n(d)^{\frac{1}{q}} \sup_{\boldsymbol{r}:\mathscr{L}(\boldsymbol{r})>r} \frac{\widehat{\mathcal{K}}_{\mathrm{Sym}}^2(\boldsymbol{r})|\boldsymbol{N}(\boldsymbol{d}_{pen},\boldsymbol{r})|^{1/2}}{\widehat{\mathcal{K}}_{\mathrm{Sym}}^2(\boldsymbol{r})|\boldsymbol{N}(\boldsymbol{d}_{pen},\boldsymbol{r})|} \tag{211}$$

$$\sim 2C_q d^{\frac{r}{q}} \sup_{\boldsymbol{r}:\mathscr{L}(\boldsymbol{r})>r} |\boldsymbol{N}(\boldsymbol{d}_{pen},\boldsymbol{r})|^{-\frac{1}{2}} \tag{212}$$

$$\sim d^{\frac{r}{q}} \sup_{\boldsymbol{r}:\mathscr{L}(\boldsymbol{r})>r} d^{-|\boldsymbol{r}|\alpha_p} \xrightarrow[d\to\infty]{} 0 \tag{213}$$

by choosing $q$ (independent of $d$) sufficiently large.

The proof of Eq. (228) is similar. Let $X_i$ denote the random variable

$$X_i \equiv \mathcal{K}_{\mathrm{Sym},>m(d)}(\boldsymbol{\xi}_i,\boldsymbol{\xi}_i) \quad \text{and} \quad \Delta_i \equiv X_i - \mathbb{E}X_i \tag{214}$$

and it suffices to prove

$$\frac{\mathbb{E}\sup_{i\in[n(d)]}|\Delta_i|}{\mathbb{E}X_i} \xrightarrow[d\to\infty]{} 0 \tag{215}$$

We have

$$\mathbb{E}X_i = \mathbb{E}_{\boldsymbol{\xi}_i\sim\boldsymbol{\sigma}_d}\mathcal{K}_{\mathrm{Sym},>m(d)}(\boldsymbol{\xi}_i,\boldsymbol{\xi}_i) \tag{216}$$

$$= \mathbb{E}_{\boldsymbol{\xi}_i\sim\boldsymbol{\sigma}_d} \sum_{\boldsymbol{r}:\mathscr{L}(\boldsymbol{r})>r} \widehat{\mathcal{K}}_{\mathrm{Sym}}(\boldsymbol{r}) \sum_{\boldsymbol{l}\in\boldsymbol{N}(\boldsymbol{d}_{pen},\boldsymbol{r})} \overline{\boldsymbol{Y}}_{\boldsymbol{r},\boldsymbol{l}}^{\mathrm{Sym}}(\boldsymbol{\xi}_i)\overline{\boldsymbol{Y}}_{\boldsymbol{r},\boldsymbol{l}}^{\mathrm{Sym}}(\boldsymbol{\xi}_i) \tag{217}$$

$$= \sum_{\boldsymbol{r}:\mathscr{L}(\boldsymbol{r})>r} \widehat{\mathcal{K}}_{\mathrm{Sym}}(\boldsymbol{r})|\boldsymbol{N}(\boldsymbol{d}_{pen},\boldsymbol{r})| \tag{218}$$

and

$$\Delta_i = X_i - \mathbb{E}X_i = w^{-1}\sum_{\boldsymbol{r}:\mathscr{L}(\boldsymbol{r})>r} \widehat{\mathcal{K}}_{\mathrm{Sym}}(\boldsymbol{r}) \sum_{\boldsymbol{l}\in\boldsymbol{N}(\boldsymbol{d}_{pen},\boldsymbol{r})} \sum_{u\neq v} \overline{\boldsymbol{Y}}_{\boldsymbol{r},\boldsymbol{l}}(\boldsymbol{\xi}_{i,u})\overline{\boldsymbol{Y}}_{\boldsymbol{r},\boldsymbol{l}}(\boldsymbol{\xi}_{i,v}), . \tag{219}$$

The remaining steps (replacing the maximal function by the $l^q$-norm, and then the $L^q$-norm by the $L^2$-norm using hypercontractivity) are similar to that of the proof of Eq. (227), which are omitted here.

## L  KERNEL CONCENTRATION, HYPERCONTRACTIVITY AND GENERALIZATION OF MEI ET AL. (2021A)

For convenience, we briefly recap the analytical results regarding generalization bounds of kernel machines from Mei et al. (2021a) Sec 3.

Let $(\boldsymbol{X}_d, \boldsymbol{\sigma}_d)$ be a probability space and $\mathscr{H}_d$ be a compact self-adjoint positive definite operator from $L^2(\boldsymbol{X}_d, \boldsymbol{\sigma}_d) \to L^2(\boldsymbol{X}_d, \boldsymbol{\sigma}_d)$. We assume $\mathscr{H}_d \in L^2(\boldsymbol{X}_d \times \boldsymbol{X}_d)$. Let $\{\psi_{d,j}\}$ and $\{\lambda_{d,j}\}$ be the eigenfunctions and eigenvalues associated to $\mathscr{H}_d$, i.e.

$$\mathscr{H}_d\psi_{d,j}(x) \equiv \int_{y\in\boldsymbol{X}_d} \mathscr{H}_d(x,y)\psi_{d,j}(y)\boldsymbol{\sigma}_d(y) = \lambda_{d,j}\psi_{d,j}(x). \tag{220}$$

We assume the eigenvalues are in non-ascending order, i.e. $\lambda_{d,j+1} \geq \lambda_{d,j} \geq 0$. Note that

$$\sum_j \lambda_{d,j}^2 = \|\mathscr{H}_d\|_{L^2(\boldsymbol{X}_d\times\boldsymbol{X}_d)}^2 < \infty. \tag{221}$$

The associated reproducing kernel Hilbert space (RKHS) is defined to be functions $f \in L^2(\boldsymbol{X}_d, \boldsymbol{\sigma}_d)$ with $\|\mathscr{H}_d^{-\frac{1}{2}} f\|_{L^2(\boldsymbol{X}_d, \boldsymbol{\sigma}_d)} < \infty$. Given a finite training set $X \subseteq \boldsymbol{X}_d$ and observed labels $f(X) \in \mathbb{R}^{|X|}$, the regressor is an extension operator defined to be

$$\mathscr{R}_X f(x) = \mathscr{H}_d(x, X)\mathscr{H}_d(X, X)^{-1} f(X). \tag{222}$$

Intuitively, when "$X \to \boldsymbol{X}_d$" in some sense, we expect the following

$$\mathscr{R}_X f(x) = \mathscr{H}_d(x, X)\mathscr{H}_d(X, X)^{-1} f(X) \to \mathscr{R}_{\boldsymbol{X}_d} f(x) = \mathscr{H}_d(\mathscr{H}_d^{-1} f)(x) = f(x), \tag{223}$$

namely, "$\mathscr{R}_X \to \boldsymbol{I}_{\boldsymbol{X}_d}$" in some sense.

Using tools from the non-asymptotic analysis of random matrices (Vershynin, 2010), the work Mei et al. (2021a) provides a very nice answer to the above question in terms of the decay property of the eigenvalues $\{\lambda_{d,j}\}$ and the hypercontractivity property of the eigenfunctions $\{\psi_{d,j}\}$. They show that $\mathscr{R}_X$ is essentially a projection operator onto the low eigenspace under certain regularity assumptions on the operator $\mathscr{H}_d$. These assumptions are stated via the relationship between the number of (training) samples $n = n(d)$, the tail behavior of the eigenvalues with index $\geq m = m(d)$ and the tail behavior of the operator $\mathscr{H}_d$

$$\mathscr{H}_{d,>m(d)}(x, \bar{x}) \equiv \sum_{j>m(d)} \lambda_j \psi_j(x)\psi_j(\bar{x}) \tag{224}$$

as the "input dimension" $d$ becomes sufficiently large.

**Assumption 4.** We say that the the sequence of operator $\{\mathscr{H}_d\}_{d \geq 1}$ satiesfies the Kernel Concentration Property (KCP) with respect to the sequence $\{n(d), m(d)\}_{d \geq 1}$ if there exsts a sequence of integers $\{u(d)\}_{d \geq 1}$ with $u(d) \geq m(d)$ such that the following holds

(a) (**Hypercontractivity.**) Let $D_{u(d)} = \text{span}\{\psi_j : 1 \leq j \leq u(d)\}$. Then for any fixed $q \geq 1$, and $C = C(q)$ such that for $\boldsymbol{f} \in D_{u(d)}$

$$\|\boldsymbol{f}\|_{L^q(\boldsymbol{X}_d, \boldsymbol{\sigma}_d)} \leq C\|\boldsymbol{f}\|_{L^2(\boldsymbol{X}_d, \boldsymbol{\sigma}_d)} \tag{225}$$

(b) (**Eigen-decay.**) There exists $\delta > 0$, such that, for all $d$ large enough, for $l = 1$ and $2$,

$$n(d)^{2+\delta} \leq \frac{(\sum_{j \geq u(d)+1} \lambda_{d,j}^l)^2}{\sum_{j \geq u(d)+1} \lambda_{d,j}^{2l}} \tag{226}$$

(c) (**Concentration of Diagonals.**) For $\{x_i\}_{i \in [n(d)]} \sim \boldsymbol{\sigma}_d^{n(d)}$, we have:

$$\frac{\sup_{i \in [n(d)]} \left| \mathbb{E}_{x \sim \boldsymbol{\sigma}_d} \mathscr{H}_{d,>m(d)}(x_i, x)^2 - \mathbb{E}_{x, \bar{x} \sim \boldsymbol{\sigma}_d} \mathscr{H}_{d,>m(d)}(x, \bar{x})^2 \right|}{\mathbb{E}_{x, \bar{x} \sim \boldsymbol{\sigma}_d} \mathscr{H}_{d,>m(d)}(x, \bar{x})^2} \xrightarrow[d \to \infty]{\text{in Prob.}} 0 \tag{227}$$

$$\frac{\sup_{i \in [n(d)]} \left| \mathscr{H}_{d,>m(d)}(x_i, x_i) - \mathbb{E}_{x \sim \boldsymbol{\sigma}_d} \mathscr{H}_{d,>m(d)}(x, x) \right|}{\mathbb{E}_{x \sim \boldsymbol{\sigma}_d} \mathscr{H}_{d,>m(d)}(x, x)} \xrightarrow[d \to \infty]{\text{in Prob.}} 0 \tag{228}$$

where $c_d \to 0$ in probability as $d \to \infty$.

**Assumption 5.** Let $\mathscr{H}_d$ and $\{m(d), n(d)\}_{d \geq 1}$ be the same as above.

(a) For $l = 1$ and $2$, there exists $\delta > 0$ such that

$$n(d)^{1+\delta} \leq \frac{1}{\lambda_{d,m(d)+1}^l} \sum_{k = \lambda_{m(d)+1}} \lambda_{d,k}^l \tag{229}$$

(b) There exists $\delta > 0$ such that

$$m(d) \leq n(d)^{1-\delta} \tag{230}$$

(c) (Spectral Gap.) There exists $\delta > 0$ such that

$$n(d)^{1-\delta} \geq \frac{1}{\lambda_{d,m(d)}} \sum_{k \geq m(d)+1} \lambda_{d,k} \tag{231}$$

Let $\mathscr{P}_{>k}$ (similarly for $\mathscr{P}_k$, $\mathscr{P}_{\leq k}$, etc.) denote the projection operator

$$\mathscr{P}_{>k}f = \sum_{j>k}\langle f,\psi_j\rangle\psi_j \tag{232}$$

**Theorem 7** (Mei et al. (2021a)). *Assume $\mathscr{H}_d$ satisfy **Assumptions 4** and **5**. Let $\{f_d\}_{d\geq 1}$ be a sequence of functions and let $X \sim \boldsymbol{\sigma}_d^{n(d)}$. Then for every $\epsilon > 0$,*

$$\|\mathscr{R}_X(f_d) - f_d\|_{L^2(\boldsymbol{X}_d,\boldsymbol{\sigma}_d)}^2 = \|\mathscr{P}_{>m(d)}f_d\|_{L^2(\boldsymbol{X}_d,\boldsymbol{\sigma}_d)}^2 + c_{d,\epsilon}\|f_d\|_{L^{2+\epsilon}(\boldsymbol{X}_d,\boldsymbol{\sigma}_d)}^2 \tag{233}$$

*where $c_{d,\epsilon} \to 0$ in probability as $d \to \infty$.*

The theorem says, $\mathscr{R}_X$ is essentially the projection operator $\mathscr{P}_{\leq m(d)}$ in the sense that when restricted to $L^{2+\epsilon}(\boldsymbol{X}_d,\boldsymbol{\sigma}_d)$,

$$\mathscr{R}_X = \mathscr{P}_{\leq m(d)} + \text{Error}_{d,\epsilon} \tag{234}$$

