# OpenReview forum: "Eigenspace Restructuring:  a Principle of Space and Frequency in Neural Networks"
_ICLR.cc/2022/Conference — ICLR 2022 Submitted_

### Official Review · Reviewer_JW33 · 2021-10-30

**Correctness:** 3
**Technical Novelty And Significance:** 2
**Empirical Novelty And Significance:** 3
**Recommendation:** 5
**Confidence:** 3

**Main Review:**

The paper provides solid foundations to understand the mechanism behind neural networks, including deep neural networks. This is done thanks to recent theoretical advances of neural network Gaussian process and neural tangent kernels. This work seems very interesting. However, this work only explore the architecture of neural networks to understand their behaviors. While the conducted analysis is carried on the model of the neural network, we find that it is missing some in-depth analysis on the other two basic ingredients of machine learning methods, namely the data and the optimization algorithm. It is worth noting that different optimization algorithms are used in the paper, namely SGD+ Momentum (used for finite-width networks) and kernel regression (used for infinite-width networks), but nothing is said about the influence of the algorithms. This is also the case about missing analysis on the influence of the training dataset.
There are some spelling and grammatical errors in the paper, such as “An demonstration”, “It describe a”, “our theorem justify”, “spherial harmonics”...
-------------------------------------------------------
Rebuttal
The authors reply is off the mark, as they did not tackle our concerns, neither address comments from the other reviews. After reading all the reviews and the authors' replies, I have downgraded my score.

**Summary Of The Paper:**

This paper provides new insights to understanding the mechanism behind neural networks, thanks to a space-frequency through the so-called eigenspace restructuring.


**Summary Of The Review:**

The paper provides solid foundations to understand the mechanism behind (deep) neural networks.

---

> ### Author Response · Authors · 2021-11-16
> **Author Respond**
>
>
> Thanks a lot for the encouraging review and for pointing out the spelling/grammatical errors in the paper.
>
> We agree with the reviewer that, to understand deep learning, we need to understand the Venn diagram of data/architectures/inference, i.e.
>
> - **[Stage I]** what can be explained by architecture/data/inference alone,
> - **[Stage II]** what must be explained jointly by any two combinations and
> -  **[Stage III]** jointly by all.
>
> However, a full understanding of this Venn diagram may require decades of effort. An in-depth understanding of [Stage I] is necessary for making progress in later stages. As such, rather than scratching the surface of each component, we wrote our paper focusing exclusively on the fundamental role of architectures and trying to dig deeper into this question. We believe our results provide certain unique and novel insights into the mathematical role of architecture, particularly the role of the hierarchical locality.  Moreover, among the three basic ingredients, architectures (CNN, RNN, GNN, Transformers, etc.) are the most critical innovation in deep learning that sets it apart from other ML methods.
>
> Finally, we would like to point out that although data/inference is not the focus, the paper provides several clues for future investigations.
>
> - **[Data]** We know (a) precisely why CNNs are much better than MLPs in terms of learnability (CNNs can model a wide range of space-frequency combinations and MLPs can only model low-frequency interactions), and (b) empirically CNN kernels perform much better than MLP kernels. (a) and (b) suggest: several types of space-frequency combinations (e.g., median-range-median-frequency) must exist in the data/task that is hard for MLPs to capture. As such, a natural next step is to investigate and identify such types of eigenfunctions on practical datasets, e.g., CIFAR10.
>
> - **[Inference algorithm]** On page 9, bullet point 3, we mentioned that: although $Y_5$ (long-range-low-frequency) and $Y_5^*$ (short-range-high-frequency) have the same learning index, in the kernel setting $Y_5$ learns faster than $Y_5^*$ but slower in the SGD setting. This difference seems to be an indication of the existence of certain “implicit" SGD effects.
>
> Overall, a comprehensive understanding of the fundamental roles of architectures can provide novel insight into the roles of other components.

---

### Official Review · Reviewer_V5xU · 2021-11-02

**Correctness:** 3
**Technical Novelty And Significance:** 3
**Empirical Novelty And Significance:** 2
**Recommendation:** 5
**Confidence:** 2

**Main Review:**

Strengths: The paper sounds very interesting! As far as I know, it proposes a unique perspective to capture architectural biases finely.

Weaknesses: The paper is very hard to follow partially because it is non-standard. Some parts lack rigor. Figures need a much better explanation.

In my opinion, the biggest issue is the interpretation of the main results (Section 5). I cannot follow how the authors created the DAG of the s-CNN network for example and from there arrived at the equation (30).

Experiments are only made for a particular and somehow peculiar type of synthetic data.

I would also appreciate some explanation on what Conv(p^2)^2 (and so on) corresponds to in terms of standard ConvNet architectures.

Detailed feedback:
- In figure 1, it was not clear to me what the black line represents. Can you study what kind of architectural changes make the family of learnable functions bigger (for example making the network wider or deeper in the case of MLPs)?
- "The amount of time needed to learn the j-th eigenfunction is about 1/K(j)" (page 3) -- can the authors elaborate on this point?
- Figure 2 is not comprehensible as it stands. Why not include a nice & explanatory caption?
- Page 5, at the beginning of the main results: "the goal is ... non-asymptotic characterization as the input dimension $d \to \infty$": here I'm confused about what does non-asymptotic refer to.
- Why is the spatial index defined twice both in Def 1 and Def 2?

Minor feedback/typos:
- Although I liked the color code in general, I found the color code for finer interpolations confusing. (In Figure 1 caption and bullet point 3 the color codes do not match).
- At the end of page 2, where the MSE is defined, it seems that the index for the summation should be (x, y) \in (X, Y)
- Page 2, .... the gradient dynamics is describe'd' ('d' is missing)
- It is not clear what "concrete" input spaces refer to on page 3.
- Page 6, It describe's' a connection .... ('s' is missing)

**Summary Of The Paper:**

The paper categorizes architectural biases from a frequency point of view through the lens of NTKs and NNGPs. The two main quantities are the frequency index (FI) and the spatial index (SI) which are calculated from the DAGs. The authors claim that these two quantities capture what type of functions (in terms of frequency and space) can be learned by specific architectures: i.e., MLPs learn LRLF functions, s-CNNs learn SRHF functions, and so on. The main theoretical contribution is Theorem 1: the authors give an architecture-dependent spectral decomposition formula for the NTK and NNGP Kernels using DAGs.

**Summary Of The Review:**

Although the paper looks very interesting, I found it very hard to follow. I think there is substantial room for making the paper more accessible (see the detailed comments above). But also, my lack of familiarity with the tools used in the paper (i.e., graph computations) may have made it difficult to read for me. I could not follow how the authors arrived at the LRLF-SRHF- and similar interpretations. Therefore, I can not recommend acceptance.

---

> ### Author Response · Authors · 2021-11-16
> **Author Respond Part 1**
>
>
> Thank you for your thoughtful feedback! We are glad you found our paper interesting. We address your comments / questions below:
>
> - **In my opinion, the biggest issue is the interpretation of the main results (Section 5). I cannot follow how the authors created the DAG of the s-CNN network for example and from there arrived at the equation (30).**
>
> We will add a new section after the intro to provide a toy example to explain LRLF-SRHF, etc. The toy example is about learning the functions
> \begin{align}
>     f_1(x) = x_9,
>     f_2(x) = x_0x_1,
>     f_3(x) = x_0x_8,
>     f_4(x) = x_6x_7(x_6^2 - x_7^2),
>     f_5(x) = x_2x_3x_5
> \end{align}
> in $x\in\mathbb R^{10}$ and the motivation for a `meaningful` definition of an architecture-and-task-dependent complexity measure.
>
> The first function is a linear function which is not very interesting here. The remaining corresponds to functions with various space-frequency combinations. By `frequency` (here and in the paper), we mean the order(=degree) of the functions.
> $f_2$ and $f_3$ are low frequency functions (deg=2)  but they have totally different "spatial distance" (i.e., the `range` used in the paper).
> The two interacting terms $x_0$ and $x_1$ in $f_2$ are spatially "close" to each other, while $x_0$ and $x_8$ are "far" from each other. In our terminology, $f_2$ and $f_3$ are of types Short-Range-Low-Frequency and Long-Range-Low-Frequency.  Similarly, $f_4$ (deg($f_4$)=4, depends only on $x_6$ and $x_7$) and $f_5$ (deg$(f_5)$=3 and depends on $x_2, x_3, x_5$) are of types Short-Range-High-Frequency and Median-Range-Median-Frequency, reps. One contribution of this paper is about how to formally defines an architecture-and-task dependent complexity measure (=learning index in the paper) so that it can (1) capture both  “frequency complex” (=frequency index) and “spatial complexity”(=spatial index) of the functions (2) mathematically characterize training/learning dynamics and generalization of the infinite-width networks. The frequency index is easier to understand as it is the weighted sum of the order of the eigenfunction. The spatial index is more abstract and harder to understand as it seems to be new to the best of my knowledge. Here is one intuitive way to understand it. It measures the quantity: how much distance do all interacting terms (pixels of an image) of a function need to travel along the “architecture” of the network to the output (i.e. logits) when *shared* paths only count for once. This is why we need “graph traversal” (computational graph) and compute the length of minimum spanning trees. Back to the above example, if we use, e.g., deep convnets with filter size and stride equal to 2 to learn the functions, the spatial index of $f_2$ is small as the interacting terms $x_0$ and $x_1$ “meet” each other in the first convolutional layer. The spatial index of $f_5$ is larger, as $x_2, x_3$ and $x_5$ meet each other in the third convolutional layer. $f_3$ has the largest spatial index, as $x_0$ and $x_8$ may not see each other until the last convolutional layer.
>
> The main results (Section 5) are about making such intuition rigorous, sharp and general in math. Hope the above explanation is helpful and let us know if you have any questions.
>
> - **Experiments are only made for a particular and somehow peculiar type of synthetic data.**
>
> One implication of our theoretical results is, in terms of performance, MLPs < Shallow CNNs < Deep CNNs < High Resolution CNNs. These empirical results are well-known in the community; e.g. in [1] table3 Myrtle5 < Myrtle7 < Myrle10 (5/7/10 layer Myrtle networks) for both kernels and finite width networks and in [2] EfficientNets improves performance. We thought it would be more interesting to see the preciseness of Theorem 1&2 in the more controlled synthetic setting.
>
> We agree that having experimental results on practical models and datasets could add values to the paper. As such, we have run extra experiments on ImageNets using ResNets (34/50/101). These experiments are aimed to support another theoretical statement (not in the submitted version): when the dataset size is sufficiently large, CNNs with and without GAP (global average pooling) perform essentially the same. As such, we compare the test performance of the original ResNets and the modified version (with GAP replaced by flattening) by continuously (a factor of 2**0.5) increasing the training set sizes. We see the performance difference between with/without GAP is dramatically reduced:
>
> | Architectures \ Training set sizes | 113240 | 160145 | 226480 | 320291 | 452960 | 640583 | 905921 | 1281167 |
> |------------------------------------|--------|--------|--------|--------|--------|--------|--------|---------|
> | ResNets50 (original)               | 0.492  | 0.551  | 0.605  | 0.653  | 0.689  | 0.72   | 0.748  | 0.765   |
> | ResNets50-Flatten                  | 0.292  | 0.36   | 0.468  | 0.569  | 0.645  | 0.693  | 0.722  | 0.745   |

---

> > ### Author Response · Authors · 2021-11-16
> > **Author Respond Part 2**
> >
> >
> > - **I would also appreciate some explanation on what Conv(p^2)^2 (and so on) corresponds to in terms of standard ConvNet architectures.**
> >
> > Conv(p^2)^2 corresponds to a 3-hidden layer convnet: the first/second hidden layer is a convolutional layer (+ activation) with filter size and stride equal to p^2, the last hidden layer is a Dense+activation layer. Conv(p)^4 has five-hidden layers, the first/second/third/fourth hidden layer is a convolutional layer (+activation) with filter size and stride equal to p and the last hidden layer is Dense+activation layer. The number of channels is 512 for all convnet experiments. We will add a code block (written by NeuralTangents/JAX)  to the appendix to specify the architectures used in the paper.
> >
> >
> >
> > - **In figure 1, it was not clear to me what the black line represents. Can you study what kind of architectural changes make the family of learnable functions bigger (for example making the network wider or deeper in the case of MLPs)?**
> >
> > Apologize for the confusion. The solid lines are the ones used by the DAGs to compute the spatial index via the minimum spanning tree. The dashed ones are the ones concerning the frequency index and they are not used in computing the spatial index. We have remade the graphs, added captions. In addition, we added a new section in the appendix to explain the DAGs and how the spatial/frequency/learning index is computed through several examples.
> >
> > - **"The amount of time needed to learn the j-th eigenfunction is about 1/K(j)" (page 3) -- can the authors elaborate on this point?**
> >
> > We will elaborate this in the paper. Roughly speaking, the residual MSE of the $j$-th eigenfunction at time equal to $t$ is given by
> > $\sim e^{-t K_j} $ and to make this number small, e.g. $e^{-t K_j} < \epsilon$ for some small $\epsilon>0$,  we need $t> (-\log {\epsilon} ) / K(j)$. That is to say, to learn the $j$-th eigenfunction, we roughly need about $1/K(j)$ time. (note that $K(0)$ is usually ~ 1)
> >
> > - **Figure 2 is not comprehensible as it stands. Why not include a nice & explanatory caption?**
> >
> > Thanks for the great suggestion. We have updated the caption. See the reply above.
> >
> > - **Page 5, at the beginning of the main results: "the goal is ... non-asymptotic characterization as the input dimension ": here I'm confused about what does non-asymptotic refer to.**
> >
> > By saying *Asymptotic*, people are interested in the limiting behavior of certain systems. In contrast,  by saying *Non-asymptotic*, they are interested in the *rate* of convergence. Specific to our case, we are interested in (1) the rate of convergence (to zero) of the eigenvalues (2) the growth rate of the number of required examples to learn certain functions as the input dimension $d\to\infty$.
> >
> >
> > -**Why is the spatial index defined twice both in Def 1 and Def 2?**
> >
> > Def 2 depends on Def 1. In the first definition, the spatial index is defined for a collection of nodes in a graph and it is not about functions at the moment. In Def 2, the spatial index is defined for (eigen)functions: we first need to find out the interaction terms and then use the nodes associated with them together with Def 1 to define the spatial index of a function.
> >
> > -**Minor feedback/typos**. Many thanks! Will improve the paper accordingly.
> >
> >
> > Please let us know if you have any further questions, particularly the interpretation of the main results. We would appreciate it if you could consider reevaluating our work.
> >
> > [1] Shankar et al. Neural Kernels without tangents,  https://arxiv.org/abs/2003.02237
> >
> > [2] Tan&Quoc, EfficientNet: Rethinking Model Scaling for Convolutional Neural Networks https://arxiv.org/abs/1905.11946

---

> > > ### Comment · Reviewer_V5xU · 2021-11-29
> > > **Thanks for the response**
> > >
> > > I thank the authors for the modifications, and for the toy example added in the appendix. It helps clarify the frequency and space concepts for those who are not familiar with the related literature.

---

> > > > ### Author Response · Authors · 2021-11-30
> > > > **Author reply**
> > > >
> > > > Thanks again for the detailed comments. In particular, the comments about frequency and space, clarities of LRLF etc, which inspired us to construct a toy example.

---

### Official Review · Reviewer_aE2B · 2021-11-02

**Correctness:** 3
**Technical Novelty And Significance:** 3
**Empirical Novelty And Significance:** 3
**Recommendation:** 5
**Confidence:** 3

**Main Review:**

Strength:
The theorem statements are simple and clear.

Weakness:
It is hard to decipher what is happening with just the theorem statements alone.

Clarity: The paper is clear till Section 3, then most part of Section 4 is confusing. The theorem statements are simple and clear. However, it is hard to decipher what is happening with just the theorem statements alone.


The following are not clear and need to be discussed explicitly:

* How is the cardinality of $\Lambda_{\mathcal{G}}$ related to the architecture?

* Does condition (1) on Assumption-G place a restriction?

* Is the weight $w_{uv}$associated to the edge $uv\in\varepsilon^{(d)}$ a constant, since it is said that $w_{uv}\equiv \alpha_u$. Looks like $w_{uv}= \alpha_u$ is used in Definition 1. However, it is confusing because $w_{uv}$ is also being used in Section 3.1 equations (14), (17-20). Please clarify, i.e, are the $w_{uv}$ in Section 3.1 and Section 4 the same, if not is it possible to use a different notation?

* Is $\sum_{v\in\mathcal{N}_0} d_v =d$?

* What is the intuition behind spatial distance?

* What are the activations that satisfy Assumption-$\phi$?

* $\mathbf{t}^{\mathbf{r}}$ is defined, but it is not clear where it is used.

* Is ${n(\mathbf{r})}$ equal to the set of input nodes for all $\mathbf{r}$ such that $\mathbf{r}>0$? Or put differently what happens if one of the $r_v$ is equal to zero and one of the input nodes get left out?

* Is there an example of $n(\mathbf{r})$ without a common ancestor $u$ such that $\phi_u$ is admissible?

* Is there an example of an $\mathbf{r}\notin\mathcal{A}(\mathcal{G}^{(d)})$, i.e., an $\mathbf{r}$ that is not learnable?

* In Section 5, (S-CNNs and SRHF Interactions),

   + it is said that since the activation function of the last layer is identity we have $\mathcal{A}(\mathcal{G}^{(d)})=\{\mathbf{r}\in \mathbb{N}^{|\mathcal{N}_0^{(d)}|}, n(\mathbf{r}) = 0 \text{ or } 1\}$. However, it seems that the same argument applies to MLPs as well (page 4, last paragraph mentions "the activations of the input/output nodes be the identity function"), yet, in equation (29) $\mathcal{A}(\mathcal{G}^{(d)})=\mathbb{N}^d$. It is not clear why there is a difference.
   + How can $|n(\mathbf{r})|$ be $1$ or $0$? Given that there are $w$ input nodes, which node gets chosen for $|n(\mathbf{r})|=1$?

* How is $|\mathbf{r}|$ defined?


* What is the meaning of short (long) range interaction?

* It is not clear what the dotted lines in Figure 2 mean. Especially, what is the point of highlighting the left most branches alone. Same goes with the Figure 4 (Figure Zoo) in the appendix.

Overall:
1. It is not clear what is meant by space and the space frequency trade off?

2. Given that there are too many constants, $\alpha_u$, $\alpha_v$, $\alpha_w$, $\alpha_p$, it might be great to give some template networks to illustrate. While the paper indeed talks about, MLP, S-CNN, D-CNN etc, the discussion about how the specific constant were arrived at in equations (29-31) is very rushed and is the major cause of confusion.



**Summary Of The Paper:**

The paper shows how the topological of convolutional neural network restructures the eigenspace of the neural kernel.

**Summary Of The Review:**

The current score is mainly due to the issues related to clarity.

---

> ### Author Response · Authors · 2021-11-17
> **Author Reply Part 1.**
>
> Thank you for your thoughtful feedback! Your comments could help us improve the clarify of the paper a lot. We addressed your comments / questions below.
>
> - There are several questions related to the concepts space (short/long range), frequency,  space-frequency trade-off and the interpretation of the main theorem(s). We think it is better to explain them by a toy example, which will be added to the paper.
>
> The toy example is about learning the functions
> \begin{align}
>     f_1(x) = x_9,
>     f_2(x) = x_0x_1,
>     f_3(x) = x_0x_8,
>     f_4(x) = x_6x_7(x_6^2 - x_7^2),
>     f_5(x) = x_2x_3x_5
> \end{align}
> in $x\in\mathbb R^{10}$ and the motivation for a `meaningful` definition of an architecture-and-task-dependent complexity measure.
>
> The first function is a linear function which is not very interesting here. The remaining corresponds to functions with various space-frequency combinations. By `frequency` (here and in the paper), we mean the order(=degree) of the functions, which is a standard terminology used in Fourier/harmonic  analysis (e.g. the $n$-th Fourier mode $e^{2\pi i nx} = (e^{2\pi i x})^n$, which is a homogeneous polynomial of degree $n$ in $\cos(2\pi x)$ and $\sin(2\pi x)$. )
> $f_2$ and $f_3$ are low frequency functions (deg=2)  but they have totally different "spatial distance" (i.e., the `range` used in the paper).
> The two interacting terms $x_0$ and $x_1$ in $f_2$ are spatially "close" to each other, while $x_0$ and $x_8$ are "far" from each other. In our terminology, $f_2$ and $f_3$ are of types Short-Range-Low-Frequency and Long-Range-Low-Frequency, resp.  Similarly, $f_4$ (deg($f_4$)=4, depends only on $x_6$ and $x_7$) and $f_5$ (deg$(f_5)$=3 and depends on $x_2, x_3, x_5$) are of types Short-Range-High-Frequency and Median-Range-Median-Frequency, reps. One contribution of this paper is about how to formally define an architecture-and-task dependent complexity measure (=learning index in the paper) so that it can (1) capture both  “frequency complex” (=frequency index) and “spatial complexity”(=spatial index) of the functions (2) mathematically characterize training/learning dynamics and generalization of the infinite-width networks. The frequency index is easier to understand as it is the weighted sum of the order of the eigenfunction. The spatial index is more abstract and harder to understand as it seems to be new, to the best of my knowledge. Here is one intuitive way to understand it. It measures the quantity: how much distance do all interacting terms (pixels of an image) of a function need to travel along the “architecture” of the network to the output (i.e. logits) when *shared* paths only count for once. This is why we need “graph traversal” (computational graph) and compute the length of minimum spanning trees. Back to the above example, if we use, e.g., deep convnets with filter size and stride equal to 2 to learn the functions, the spatial index of $f_2$ is small as the interacting terms $x_0$ and $x_1$ “meet” each other in the first convolutional layer and "share" the same path to the logit layer. The spatial index of $f_5$ is larger, as $x_2, x_3$ and $x_5$ meet each other in the third convolutional layer. $f_3$ has the largest spatial index, as $x_0$ and $x_8$ may see each other in the last convolutional layer.
>
> By saying frequency-space trade-off, we mean for a given "learning" budget (e.g., the total number of training points/the amount of gradient flow time), the set of learnable functions are the ones with "the spatial complexity (spatial index) + frequency complexity (frequency index)" below the budget. That is to say, one cannot model Long-Range-High-Frequency interactions (relative to the "learning" budget.)  Back to the above example, the function $g(x)=x_0 x_3 x_6 x_9$ has high complexity in both space and frequency (Long-Range-High-Frequency). The motivation that we use "a principle of frequency and space" and "frequency-space trade-off" in the paper is due to the uncertainty principle in Fourier/harmonic analysis (see https://en.wikipedia.org/wiki/Uncertainty_principle#Harmonic_
> analysis): a function cannot simultaneously localize in time and frequency space. In our setting, learnable functions cannot simultaneously have high complexity in space and frequency.
>
>  Back to the interpretation of the main results, Theorem 1 in the paper is about the precise dependence of the complexity measure (learning index) on the topology of the networks, and Theorem 2 is about the precise dependence of almost sharp generalization bound on the topology of networks. Both statements are in a high-dimensional setting. As such, given a "learning" budget and an architecture (in the setting of the paper),  the class of learnable functions can be precisely characterized by the learning index.

---

> > ### Author Response · Authors · 2021-11-17
> > **Author Reply Part 2.**
> >
> > - **How is the cardinality of $\Lambda(G)$  related to the architecture?**
> >
> > Roughly speaking, the cardinality of $\lambda_G$ is equal to the cardinality of the set of filter sizes used in convnet(in a high-dimensional setting). Consider the case: the inputs are in $\mathbb R^d$ and the network is a $L$ layer ConvNets. There are two extreme cases: (1) $|\lambda_G|=1$ if the filter size and stride  in all convolutional layers is equal to $d^{1/L}$; (2) $|\lambda_G|=L$ where the filter size  and the stride of layer $j$ is $d^{\alpha_j}$ and $\alpha_j$ are all different ($\sum_{\alpha_j}=1$.
> > Does condition (1) on Assumption-G place a restriction?
> >
> > - **Is the weight…**
> >
> > They are different. We will modify the notation! Thanks a lot for catching the notation bug!
> >
> > - **$\sum_v {d_v} = 1**
> >
> > Yes! Will mention this in the paper.
> >
> > - **What is the intuition behind spatial distance?**
> >
> > See the toy example above.
> >
> > - **What are the activations that satisfy Assumption**
> >
> > The function $ \phi(x) = \sqrt(2)\sin(\sqrt(2) x + \pi/4)$ (used in the paper, whose NNGP kernel is the Gaussian kernel) is semi-admissible. If we centralize it to have mean zero, then it should be admissible. However, if $\phi^*$ has vanishing lower order derivatives at zero, it has a certain implication to the learning index, e.g., the network might deprioritize the learning of certain lower order interactions.
> >
> > - **${\bf t}^{\bf r}$ is defined …**
> >
> > It is mainly used to connect definition 2 to definition 1 (and will be heavily used in the appendix.)
> > **Is {{\bf n}}({\bf r})... equal to the set of all input nodes for all ${\bf r}_v$ that ${\bf r}_v>0$… **
> >
> > Do you mean for all ${\bf r}$ that ${\bf r}_v>0$? If so, yes.
> >
> > - **Is there an example of $n({\bf r})$ without a common ancestor …**
> >
> > Yes. For example, S-CNN in our paper, where the network architecture is “convolution-activation-flatten-dense”. In this case, there isn’t nonlinear interaction between pixels in (non-overlapping) patches.
> >  Is there an example of an ${\bf r}\notin \mathcal A(\mathcal G)$
> > Yes. Let’s consider the S-CNN example above. Suppose the inputs are in $\mathbb R^{d}$  and the filter size and stride of the convolutional layer are equal to $k$.  Then functions $x_ix_{j}$ is not learnable if $|i-j| \geq  k$
> >
> > - **n Section 5, (S-CNNs and SRHF Interactions),**
> >
> > -- last paragraph mentions "the activations of the input/output nodes be the identity function"), yet, in equation (29) $ \mathcal A(\mathcal G(d)) = \mathbb N^{d}$ It is not clear why there is a difference.
> > Apologize for the serious typo here!  For MLP, it should be $ \mathcal A(\mathcal G(d)) =\mathbb N\backslash{0}$. I.e. , ${\bf r}$ for MLP is a non-negative integer, not a multi-index.
> >
> > --  **How can $|n({\bf r})| ….$ be 1 and 0?  Given that there are $w$ input nodes, which node gets chosen for $|n({\bf r})|=1$.**
> >
> > Good question. For a S-CNN (architecture ==  “convolution-activation-flatten-dense”), there is only one activation layer that “mixes” all terms  *within* the same patch and there isn’t any *mixing* (non-linear interaction) between any two patches. So the learnable functions are of the form $\sum_{i} f_i(x)$, where $f_i(x)$ is some function defined on the $i$-th patch. As such, the non-zero eigenfunctions must have $|n({\bf r})|=1$, i.e. they are defined on only one patch.
> >
> > - **It is not clear what the dotted lines in Figure 2 mean. Especially, what is the point of highlighting the left most branches alone. Same goes with the Figure 4 (Figure Zoo) in the appendix.**
> >
> > Sorry for the confusion. The highlighting branch represents the minimum spanning tree that connects all $n({\bf r})$ to the output node for a given eigenfunction $Y_i$. The solid lines are the one used to compute the spatial index. We have remade the graphs, added captions. In addition, we added a new section in the appendix to explain the DAGs and how the spatial/frequency/learning index is computed through several examples used in the experiments. Hope this will also resolve the question “the discussion about how the specific constant were arrived at in equations (29-31) is very rushed and is the major cause of confusion.”
> >
> >
> > Please let us know if you have any further questions. We would also appreciate it if you could consider reevaluating our work.

---

> > > ### Comment · Reviewer_aE2B · 2021-11-29
> > > **Thanks for the response**
> > >
> > > The author response and the revision (the toy example in the appendix certainly helped) has taken care of most of the concerned raised in my review. I thank the authors for the same.

---

> > > > ### Author Response · Authors · 2021-11-30
> > > > **Reply**
> > > >
> > > > Thanks again for the detailed comments, which is super helpful for us to improve the paper. We are glad that the toy example is helpful.

---

### Official Review · Reviewer_nazU · 2021-11-09

**Correctness:** 2
**Technical Novelty And Significance:** 2
**Empirical Novelty And Significance:** 3
**Recommendation:** 3
**Confidence:** 3

**Main Review:**

Strengths:

This paper evaluates the success of deep neural networks by running extensive experiments that show the tradeoff between space and frequency.

Weakness:

The meaning of the word frequency is unclear in the context of this paper.

The main idea behind theorem "eigenspace restructuring" which replaces one eigenvector/value computation with a set of part based eigenvalue/eigenvectors  has limited novelty.  The novelty is in its use in neural network architecture analysis.

The authors might want to consider evaluating the performance of a system based on the amount of data employed and based on how data was acquired (observational vs experimental studies).  Systems that employ data from observational studies are known to be susceptible to selection bias and spurious correlations, as opposed to data from experimental studies that are employed in objective causal inference.


Missing reference:
As the authors might recall, Vasilescu etal. in their ICPR 2020 paper have advocated replacing the SVD computation with a set of part based SVDs for which they provided a closed form mathematical derivation.  Based on this mathematical derivation, they developed  the Incremental Hierarchical M-mode Block SVD which was demonstrated experimentally.

@inproceedings{Vasilescu20,

author={Vasilescu, M. Alex O. and Kim, Eric and Zeng, Xiao S.}, booktitle={2020 25th International Conference of Pattern Recognition (ICPR 2020)}, title={Causal{X}: {C}ausal e{X}planations and {B}lock {M}ultilinear {F}actor {A}nalysis}, year={2021}, location={Milan, Italy}, month={Jan}, pages={10736--10743} }





**Summary Of The Paper:**

The authors show that deep convolutional networks restructure the eigenspaces of the inducing kernels,
which empowers them to learn a dramatically broader class of functions, covering a wide range
of space-frequency combinations.

**Summary Of The Review:**

The authors evaluate deep convolutional networks through the lens of hierarchical locality. The paper evaluates the performance of deep neural networks by performing extensive experiments that show the tradeoff between space versus frequency.  The eigenspace restructuring theorem has a well-chosen name.  However, eigenspace restructuring is a well-known concept and the novelty of the theorem is limited.

Clarity of the writing and its overall organization needs improvement.  The word frequency needs to be better defined.  It is not clear if there are any novel insights.


----
Update:  My original score was too generous and it was in anticipation that any mathematical inconsistencies or ambiguities would be addressed.

---

> ### Author Response · Authors · 2021-11-17
> **Author Reply**
>
> We thank the reviewer for the insightful comments and for pointing out a new reference, which is both interesting and highly insightful.
>
> However, we kindly disagree with the reviewer's point regarding the novelty of the paper and the eigenspace restructuring theorem. We are more than happy to have an in-depth discussion and debate about this point. It is possible that the reviewer may misinterpret the goals and results of the paper. If so, we apologize for this and will improve the presentation accordingly.
>
> First of all, this is a pure theory paper with the goal of establishing a *mathematical* characterization of the architectural inductive biases, in particular, explaining the role of architectures (e.g., CNNs) in breaking the curse of dimensionality. In terms of theoretical contribution, we believe this is novel. In addition,  we did not propose any new algorithms to solve existing practical learning problems nor propose new approaches or experiments to understand neural networks empirically. We appreciate the reviewer for praising our experimental contribution. Although they are an important part of the paper that empirically verifies the main theorems, they are not its core contribution.
>
> - **The meaning of the word frequency is unclear in the context of this paper.**  It is briefly mentioned in the abstract, "... frequency, which measures the order of interactions". We will elaborate on it in the main text. *Frequency* is a standard terminology used in Fourier analysis/signal processing, etc., to measure the "complexity" of a function/signal. Because the $n$-th Fourier mode $e^{2\pi nx}$ ($n$-th frequency) is a degree $n$ homogenous polynomials in $\cos(2\pi x)$ and $\sin(2\pi x)$, *Frequency*= degree/order of polynomials in this setting.
>
> - **The main idea behind theorem "eigenspace restructuring" which replaces one eigenvector/value computation with a set of part based eigenvalue/eigenvectors has limited novelty. The novelty is in its use in neural network architecture analysis.**
>
> We think there is some misunderstanding of the paper here.  Could you elaborate on the meaning of "replaces one eigenvector/value computation with a set of part based eigenvalue/eigenvectors".
>
> As far as we know, we did not "replace" any "eigenvector/value computation" in the paper. In the paper, we treat the NTK/NNGP kernel as an integral operator between two Hilbert spaces, diagonalizing it and *uncovering* its intrinsic eigenstructures. To the best of our knowledge, our paper is the first one to finely establish the relation between the topological structure of a network, the eigenstructure of the inducing kernel and learning dynamics, which Reviewer V5xU also mentions: "As far as I know, it proposes a unique perspective to capture architectural biases finely."
>
> - **Missing reference: As the authors might recall, Vasilescu etal. in their ICPR 2020 paper have advocated replacing the SVD computation with a set of part based SVDs ...**
>
> Thanks a lot for the reference, which is both interesting and insightful. We will add that to the reference. Nevertheless, we would like to mention that, although there is some similarity between this reference and the current paper, e.g., both investigate compositional locality, they are of different species and orthogonal to each other. We want to elaborate on this point because this may be the source of the misinterpreting of our results. The goals, methods, and results of the two are completely different.
> 1. **Goals**. The *current* paper aims for a mathematical characterization of the architectural inductive biases. The *reference* paper: "This paper proposes a unified multilinear model of wholes and parts that define a data tensor in terms of a hierarchical data tensor" (from the paper).
>
> 2. **Methods**. The current paper use tools from infinite-width networks (NTK/NNGP kernel), analysis on spherical harmonics, high-dimensional probability theory, etc. The reference paper: tensor-based factorization, SVD, etc.  In particular, the former works on infinite-dimensional functions space of type $\prod L^2(\mathbb S_{p-1})$ and the latter works on the space of tensor product, i.e. a multilinear space of type  $ \mathbb R^{I_1\times I_2 \cdots  \times I_k}$, which is finite-dimensional.
>
> 3. **Results**. Current paper: a mathematical characterization of architectural inductive biases in a high-dimensional setting. The reference paper:  "This paper deepens the definition of causality in a multilinear (tensor) framework by addressing the distinctions between intrinsic versus extrinsic causality, and local versus global causality." (from the paper)
>
> Overall, we believe the ideas from the paper are mostly orthogonal to each other. Combining them together is a very promising direction in incorporating causality into neural networks in a principal way that we would like to investigate. However, the current paper is mainly about the mathematical characterization of inductive biases.

---

> > ### Author Response · Authors · 2021-11-17
> > **Author reply part 2**
> >
> > - **The authors might want to consider evaluating the performance of a system based on the amount of data employed and based on how data was acquired (observational vs experimental studies). Systems that employ data from observational studies are known to be susceptible to selection bias and spurious correlations, as opposed to data from experimental studies that are employed in objective causal inference.**
> >
> > Thanks for the suggestion. This is a great idea, and we agree there is a huge potential for connecting neural networks and causality.
> >
> > - **However, eigenspace restructuring is a well-known concept and the novelty of the theorem is limited.**
> >
> > We kindly disagree with this one as discussed above, at least in the field of theory of deep learning. It would be great if the reviewer could provide more references and content to elaborate on the claim. We also think it may not be fair to compare the novelty of two similar/related ideas in practice v.s. in theory. E.g., one can also argue that the "space-frequency" principle is similar to the "time-frequency" analysis in signal processing/ Fourier analysis, which has already existed in the literature for nearly one century. Discovering and mathematically defining the intrinsic structure of various systems, making connections/analogs (formally) between unknown and known systems, and among different fields are indeed central themes in science and mathematics. Moreover, the gap between an idea that is "practically/heuristically" sound to that is theoretically grounded can be arbitrarily large. Finally, our theorems provide a fine and sharp characterization between the topology of a network and its generalization performance, which is new. To our best of knowledge, the next best results in this direction are  MLPs [1]  and one-layer CNNs [2, 3]. Our results work for arbitrarily deep CNNs.
> >
> > - **Clarity of the writing and its overall organization needs improvement. The word frequency needs to be better defined. It is not clear if there are any novel insights.**
> >
> > Please see the discussion above regarding *frequency*. We will add a section to explain the motivations of the paper together with a toy example to help parse the technical ideas; (see reply to Reviewer aE2B).
> >
> > Please let us know if you have any further questions. We would also appreciate it if you could reevaluate our work.
> >
> > [1] Linearized two-layers neural networks in high dimension. (https://arxiv.org/abs/1904.12191)
> >
> > [2] Locality defeats the curse of dimensionality in convolutional teacher-student scenarios (https://openreview.net/forum?id=sBBnfOFtPc)
> >
> > [3] What Breaks the Curse of Dimensionality in Deep Learning? https://openreview.net/forum?id=xBEZnN9Qts4

---

### Author Response · Authors · 2021-11-23
**Revision Uploaded**

We want to thank the reviewers again for their many insightful comments. We make some revisions accordingly.

- We add a section (Sec. B in the appendix) containing a toy example and motivation to help readers understand several concepts: space, frequency, space-frequency trade-off, short-/long-range interactions, etc.

- A new section (in the appendix) illustrates how the space/frequency/learning indices are computed. We also remake the graphs for the DAGs; see Sec. D.

- Simple sample code for $\text{CNN}(p^2)^{\otimes 2}$, etc.; Sec. F.

- New theoretical and experimental results comparing global average pooling and flattening; Sec. C.

---

### Decision · Program_Chairs · 2022-01-20

**Decision:**

Reject

**Comment:**

The paper shows that deep convolutional neural networks in the kernel regime restructure the eigenspaces of the inducing kernels, which leads to some insights regarding the range of space-frequency combinations learned by such networks.

The reviewers identified a number of problems with the current submission. For instance, they found that the paper is hard to follow, it lacks clarity and the theorem statements are hard to understand. The authors also use a somewhat non-standard experimental setup.

Despite an extensive discussion with the authors which cleared out a few minor problems, the bulk of the concerns of the reviewers were not successfully adressed. I am therefore not able to recommend acceptance. The authors need to improve the clarity of the paper and provide more discussions of the theorems in a resubmission, as well as potentially reconsider their experimental setup.